# The Importance of Being Lazy: Scaling Limits of Continual Learning

Jacopo Graldi [* 1]   Alessandro Breccia [* 2]   Giulia Lanzillotta [3 4]   Thomas Hofmann [3]   Lorenzo Noci [3]

## Abstract

Despite recent efforts, neural networks still struggle to learn in non-stationary environments, and our understanding of catastrophic forgetting (CF) is far from complete. In this work, we perform a systematic study on the impact of model scale and the degree of feature learning in continual learning. We reconcile existing contradictory observations on scale in the literature, by differentiating between *lazy* and *rich* training regimes through a variable parameterization of the architecture. We show that increasing model width is only beneficial when it reduces the amount of *feature learning*, yielding more laziness. Using the framework of dynamical mean field theory, we then study the infinite width dynamics of the model in the feature learning regime and characterize CF, extending prior theoretical results limited to the lazy regime. We study the intricate relationship between feature learning, task non-stationarity, and forgetting, finding that high feature learning is only beneficial with highly similar tasks. We identify a transition modulated by task similarity where the model exits an effectively lazy regime with low forgetting to enter a rich regime with significant forgetting. Finally, our findings reveal that neural networks achieve optimal performance at a critical level of feature learning, which depends on task non-stationarity and *transfers across model scales*. This work provides a unified perspective on the role of scale and feature learning in continual learning.

## 1. Introduction

Modern neural networks (NNs) have achieved impressive results on many hard benchmarks, recently beating all expectations with the introduction of large autoregressive language models (OpenAI, 2023). However, the learning algorithms in use are only applicable to stationary data distributions. In particular, these algorithms fail to retain knowledge when learning new skills, a phenomenon termed *Catastrophic Forgetting* (CF) (McCloskey & Cohen, 1989) in the literature. Arguably, the design of algorithms which can adapt to changes in the environment is a crucial step towards the development of more widely applicable, trustworthy, and scalable AI systems (Wang et al., 2024). Continual learning (CL) (e.g. Ring, 1994; Thrun & Mitchell, 1995; Silver et al., 2013; Parisi et al., 2019; Hadsell et al., 2020; Lesort et al., 2020) directly aims at devising algorithms which allow the model to learn under distribution shifts. This entails two requirements: knowledge adaptation – also called *plasticity* – and knowledge retention – *stability*. For NNs these objectives are inherently conflicting, creating an unavoidable tradeoff, often called the *stability-plasticity dilemma*. Various studies have investigated the causes of CF in artificial neural networks. However, theoretical models of CF have only been established in simplified settings, e.g., by *assuming fixed features* during training (Bennani et al., 2020; Doan et al., 2021; Evron et al., 2022; Goldfarb & Hand, 2023; Lin et al., 2023; Goldfarb et al., 2024). At the same time, empirical studies have significantly contributed to the understanding of CF in modern deep networks used in practice (Mirzadeh et al., 2020; 2021; 2022a; Ramasesh et al., 2020; 2022). Among these, multiple recent works have focused on the *role of scale and overparameterization* in continual learning, producing different if not contradictory answers (Ramasesh et al., 2022; Mirzadeh et al., 2022a;b; Goldfarb & Hand, 2023; Lin et al., 2023; Wenger et al., 2023) – as elaborated below in Section 2. In particular, the question of whether *scaling alone can ameliorate forgetting* is still open.

A separate thread of literature has studied the so-called *scaling limits* of neural networks, where the network dimensions (i.e., width and depth) are taken to infinity (Neal, 1996). Depending on the *parameterization* used – see formal definition in Sec. 3 – the network exhibits fundamentally different training dynamics. More specifically, in one ex-

---

[*]Equal contribution  [1]Dept. of Information Technology and Electrical Engineering, ETH Zurich, Switzerland [2]Dept. of Physics and Astronomy, University of Padua, Italy [3]Dept. of Computer Science, ETH Zurich [4]ETH AI Center. Correspondence to: <jacopograldi@gmail.com>.

*Proceedings of the 42nd International Conference on Machine Learning*, Vancouver, Canada. PMLR 267, 2025. Copyright 2025 by the author(s).

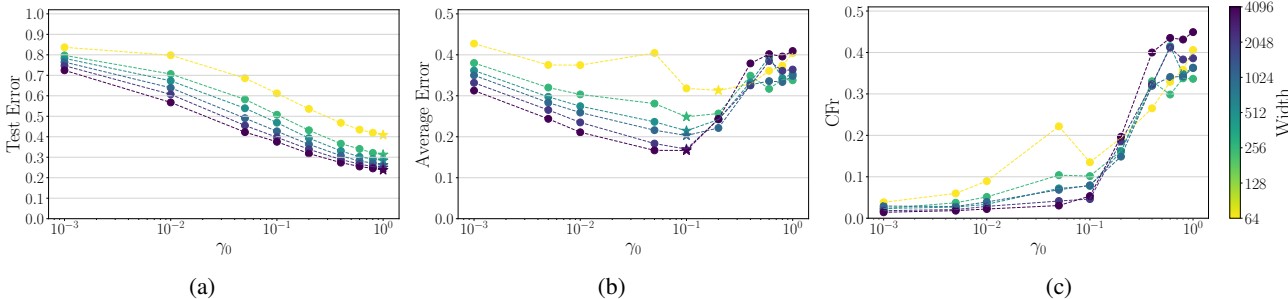

*Figure 1.* (a) Test error of stationary training (joint CIFAR10). (b) Final average error and (c) Catastrophic Forgetting rate (CFr) of non-stationary sequential training (Split-CIFAR10). The factor $\gamma_0$ interpolates between lazy ($\gamma_0 \to 0$) and rich ($\gamma_0 = 1$) regimes. (a) In stationary training, more feature learning and width scaling are strictly beneficial. (b) In non-stationary training the optimal performance is achieved at a critical level of feature learning $\gamma_0^\star \approx 0.1$ above which (c) forgetting explodes worsening the final performance, and the increased scale of the network is wasted without any benefits. The region of $\gamma_0$ at which forgetting transitions from low to high defines a change of training regime of the network from being *effectively* lazy to having *rich* dynamics. All figures report the average over 5 random seeds.

treme – the Neural Tangent Parameterization (NTP) – the dynamics are *lazy* (Chizat et al., 2019), meaning that the network activations (a.k.a features) evolve with vanishingly small magnitude during training, effectively keeping the network close to its initialization (Jacot et al., 2018; Arora et al., 2019; Lee et al., 2019); while in the other extreme – the Maximal Update Parameterization ($\mu$P) – the dynamics are *rich*, and feature learning is maintained at any scale (Yang & Hu, 2020; Bordelon & Pehlevan, 2022).

Harnessing the power of exactly characterizing the network training dynamics, in this work we study the *scaling limits of catastrophic forgetting*, aiming at a comprehensive understanding of the role of scale in continual learning. In particular, we interpolate between the lazy and rich training regimes, varying a parameter $\gamma_0$ of the network parameterization that measures the *degree of feature learning*. In the analysis of scale, $\gamma_0$ reveals to be the key to explaining the aforementioned inconsistencies in the literature.

We find that in non-stationary training the effect of scaling is essentially modulated by the degree of feature learning of the model, finding that *past a certain degree of feature learning scaling the network does not benefit performance* (Fig. 1b, center). Curiously, we find that the optimal plasticity-stability tradeoff is achieved at a fixed data-dependent degree of feature learning $\gamma_0^\star < 1$, which *transfers across widths*. Thus, in practice, scaling the network beyond this threshold effectively *wastes the network capacity*. In order to appreciate the subtlety of our results, it is important to consider the standard setting of stationary training: increasing the degree of feature learning or the model scale (in particular width) decreases the error (see Fig. 1a, left). On the other hand, an optimal $\gamma_0^\star < 1$ in the non-stationary case underlies the destructive nature of high feature learning on knowledge retention. We find that a large $\gamma_0$ sharply increases catastrophic forgetting (Fig. 1c, right), reflecting a transition of the network dynamics from being

*effectively* lazy to being *rich* and uncover its intertwined nature with the non-stationarity of data.

**Contributions and Paper Structure**

- In Section 4, we demonstrate that the effect of width scaling on catastrophic forgetting depends on the network parameterization: while NTP leads to reduced forgetting with scale, $\mu$P does not. This indicates the entangled relationship between scale and training regime. We then extend the Mean Field formalism to task-sequential training, modeling infinite-width rich training dynamics. This model aligns with finite-size network predictions, making it a useful framework for studying NNs under task non-stationarities.

- In Section 5, we find that there is a non-linear relationship between the degree of feature learning $\gamma_0$ and forgetting characterized by a sharp *low-to-high* forgetting transition. We also characterize the $\gamma_0^\star$ that optimizes the stability-plasticity tradeoff, showing that it *transfers* across model scales.

- In Section 6 we study the role of the degree of non-stationarity by controlling the *task similarity* in both synthetic and natural tasks. We show that lower task similarity causes a shift of the *low-to-high* forgetting transition towards larger values of $\gamma_0$, which translates to a larger $\gamma_0^\star$. This suggests that laziness is key in highly non-stationary scenarios.

## 2. Related Work

**Catastrophic Forgetting and the Question of Scaling** Catastrophic Forgetting is a known phenomenon in the deep learning literature since the early nineties (McCloskey & Cohen, 1989; French, 1999), and to date it has been consistently observed in distinct benchmarks and network architectures (Wang et al., 2024; Mai et al., 2022; De Lange

et al., 2021; Ke & Liu, 2022; Khetarpal et al., 2022). Even large language models (LLM) are vulnerable to CF when trained on highly non-stationary data streams (Luo et al., 2023; Wu et al., 2024).

Goldfarb & Hand (2023); Lin et al. (2023) study the effect of overparameterization in linear regression models, proving that in this setting scaling reduces CF. For neural networks used in practice, the existing empirical evidence depicts a more nuanced picture, and – due to the complexity of the question – there is currently no theory on it. Ramasesh et al. (2022) find that scaling benefits CL *only for pretrained models*, and not for models trained from scratch. By contrast, Mirzadeh et al. (2022a;b) observe that increasing the width of a network – but not the depth – reduces CF even when training from scratch. In response, Wenger et al. (2023) show that the observation of Mirzadeh et al. (2022a) is dependent on the training time: increasing the training time eliminates the positive effect of scaling on CF. Our work contributes to this line of research, offering a solid hypothesis – based on theoretical and empirical grounds – as to why and when scaling helps CF.

**Scaling Limits and Continual Learning** The research on scaling limits aims at theoretically characterizing the behavior of neural networks in the limit of infinite width and/or depth. Early works describe the network function at initialization (Lee et al., 2017; Yang, 2020; Hayou et al., 2021; Noci et al., 2021), which has been useful to prescribe optimal initialization conditions for stable training (Schoenholz et al., 2016; Hanin, 2018; Hanin & Rolnick, 2018; Martens et al., 2021). Later, the field advanced to study the network while training, either in the so-called *rich* or *lazy* regime (Yang & Hu, 2020; Bordelon & Pehlevan, 2022). Although rich regimes are often credited as superior in performance (Fort et al., 2020), this premise has been partially reconsidered: Petrini et al. (2022) demonstrate that rich mean-field training can drive fully connected networks to overfit to sparse features, reducing generalization. These results have been derived under the assumption of a stationary input distribution. In this work, we *extend the theory on scaling limits to the multi-task setting*.

We are not the first to apply scaling limits to the study of continual learning. Doan et al. (2021); Bennani et al. (2020) provide a theory of catastrophic forgetting in the lazy regime. Crucially, their results do not extend to the rich feature learning regime. Our results show that the presence of feature learning is a key factor in the behavior of CF with scale, marking the relevance of our contribution to the existing literature. Also, recent work has applied tools from statistical mechanics to continual learning (Ingrosso et al., 2024; Mori et al., 2024; Shan et al., 2024). In Shan et al. (2024), the authors study the large data, width asymptotics of continual learning by analyzing the posterior Gibbs ensemble. Compared to their work, here we focus on the gradient descent training dynamics in the fixed data, large width limit.

## 3. Methodology

Let us introduce the notation for residual networks of fixed width $N$ and with $L$ residual layers (i.e. the depth), where the $\ell$-th layer's parameters are initialized as $W_{ij}^\ell \sim \mathcal{N}(0, \sigma_\ell^2)$. For an input $\mathbf{x} \in \mathbb{R}^D$, the preactivations of the first block are defined as $\mathbf{h}^1(\mathbf{x}) = \beta_0 \mathbf{W}^0 \mathbf{x}$. The $N$-dimensional preactivations of the $\ell$-th block have a residual branch scaled by $\beta_\ell$, and the outputs $f(\mathbf{x}) \in \mathbb{R}$ are additionally inversely scaled by $\gamma$:

$$\mathbf{h}^{\ell+1}(\mathbf{x}) = \alpha \mathbf{h}^\ell(\mathbf{x}) + \beta_\ell \mathbf{W}^\ell \phi(\mathbf{h}^\ell(\mathbf{x})),$$
$$f(\mathbf{x}) = \frac{\beta_L}{\gamma} \underbrace{\mathbf{w}^L \cdot \phi(\mathbf{h}^L(\mathbf{x}))}_{h^{L+1}}, \tag{1}$$

where we use the scalar $\alpha \in \{0, 1\}$ to turn on and off skip connections. The choice of how to scale the factors $\beta_\ell$ and $\gamma$, the weights initialization variance $\sigma_\ell^2$, and the (possibly time-varying) learning rate $\eta(t)$ as the width increases, differentiates between the two parameterizations considered in this work: the *Neural Tangent Parameterization* (NTP) and the *Maximal Update Parameterization* ($\mu$P). We summarize the scaling of these parameters in Tab. 1. Importantly, by varying the $\gamma_0$ parameter it is possible to smoothly interpolate between lazy ($\gamma_0 \to 0$) and rich ($\gamma_0 = 1$) regimes. To

*Table 1.* Branch and output scales, learning rate, and weight variance in the three parameterizations: NTP (PyTorch default) (Yang & Hu, 2020) and Mean Field / $\mu$P (Bordelon & Pehlevan, 2022).

| | NTP | Mean Field / $\mu$P |
|---|---|---|
| Branch Scale $\beta_\ell$ | $\begin{cases} N^{-1/2}, & \ell > 0 \\ D^{-1/2}, & \ell = 0 \end{cases}$ | $\begin{cases} N^{-1/2}, & \ell > 0 \\ D^{-1/2}, & \ell = 0 \end{cases}$ |
| Output Scale $\gamma$ | $1$ | $\gamma_0 N^{1/2}$ |
| LR Schedule $\eta(t)$ | $\eta_0(t)$ | $\eta_0(t)\gamma_0^2 N$ |
| Weight Variance $\sigma_\ell^2$ | $1$ | $1$ |

have comparable starting points of the scaling behavior, we normalize the parameterizations of the $\mu$P to be equivalent to the NTP at a base width of $N = 64$. Further details are in App. A.3.

### 3.1. Experimental Setup

In most of our experiments, we utilize a ResNet architecture (with base width $N = 64$ and depth $L = 6$), trained with Stochastic Gradient Descent (SGD) and evaluated on MNIST, CIFAR10, and TinyImagenet and their suitable adaptions to the continual learning setting. We also evaluate a Convolutional Neural Network (by setting $\alpha = 0$ in

Eq. 1) in App. B.7 finding no significant deviations from the ResNet case. For the experiments involving the infinite-width simulations of the network dynamics, instead, we will use a simpler non-linear two-layer perceptron (MLP henceforth), on a small subset of MNIST with 30 samples, suitably modified as a 2-tasks CL benchmark. The choice of this simplified setting is imposed by the numerical complexity (cubic in both the number of samples and the number of tasks) of the infinite-width simulations. Further details on the experimental setup are reported in App. A.4.

### 3.1.1. METRICS

For a benchmark with $T$ tasks, we define the test accuracy on the task $\mathcal{T}_i$ after training on $\mathcal{T}_j$, with $i, j \in \{1, ..., T\}$, as $a_{j,i}$. The *learning accuracy* (LA) is a measure of plasticity, and it is defined simply as the average $\langle a_{i,i} \rangle_i$ over the task index $i$ (Mirzadeh et al., 2022a). We will also consider the *learning error* (LE), namely the complement of the LA $(1 - \text{LA})$. The *average accuracy* (AA) measures the plasticity-stability tradeoff, and is the average $\langle a_{T,i} \rangle_i$. We will also use its complement – the *average error* (AE), $(1 - \text{AA})$.

**Catastrophic Forgetting rate (CFr)** To evaluate the models' capacity to retain the knowledge of the past tasks, prior work (e.g. Mirzadeh et al. (2022a)) defines *catastrophic forgetting* (CF) (Def. A.3), as the average drop in accuracy after training on all later tasks. However, it implicitly depends on the raw learning accuracy, making it challenging to fairly compare models with different performance levels—such as in this work—and has led to misleading conclusions in the literature (as argued by Wenger et al. (2023)). For this reason, we introduce a novel metric – the *Catastrophic Forgetting rate* (CFr) – as the average *relative* drop in accuracy after training on all other tasks (Def. A.4).

All experiments involving the MLP and the infinite-width limit are evaluated in terms of training loss performance since the proposed theory (introduced in Sec. 4.1) will predict those dynamics. Concretely, if we define the training loss on the task $\mathcal{T}_i$ after training on $\mathcal{T}_j$, with $i, j \in \{1, ..., T\}$, as $\mathcal{L}_{j,i}$, then we can substitute $a$ with $\mathcal{L}$ in the definitions of the LE, AE and CFr above, obtaining the *Learning Loss*, *Average Loss*, and *CFr (loss)*. Formal definitions of these metrics are found in App. A.1.

## 4. The Effect of Width Scaling Depends on the Parameterization

We begin our investigation of the role of width scaling in non-stationary training on the CL benchmark Split-CIFAR10 and with a ResNet model (more details in App. A). We reckon that this setup is similar to previous work on the role of width scaling in CL. However – crucially – in contrast to previous studies (Mirzadeh et al., 2022a; Wenger

et al., 2023), we consider two different parameterizations of the architecture when scaling: NTP and $\mu$P.

In Fig. 2 we show the CFr for varying width and parameterization. We identify two separate behaviors in width, depending on the parameterization. Thus *the parameterization shapes the relationship between width and forgetting*, where only NTP endows a diminishing forgetting with scale. When the network is parameterized following the $\mu$P, width does not offer reductions in forgetting. Moreover, the forgetting curves of wider $\mu$P models almost coincide, a width consistency reminiscent of the observations of Vyas et al. (2024) in the stationary online setting.

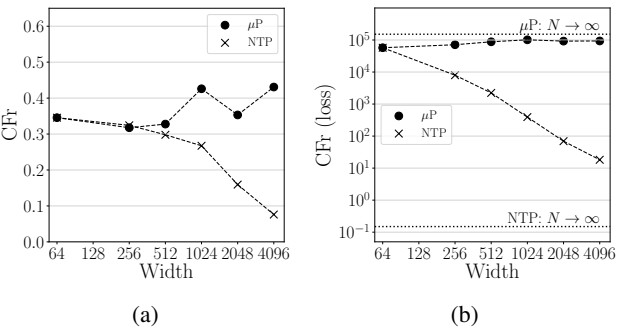

(a)                 (b)

*Figure 2.* The effect of width scaling on forgetting with models parameterized with the NTP and muP ($\gamma_0 = 1$). (a) Split-CIFAR10 with ResNet model; (b) Permuted-MNIST with MLP and infinite-width asymptotic behavior.

**Width Scaling Dilemma** Our finding provides a new key to interpret the observations of (Mirzadeh et al., 2022a; 2020; Wenger et al., 2023; Ramasesh et al., 2020) regarding the effect of scale on CF. In particular, we postulate that the conclusion that width-scaling reduces CF depends on lazy training dynamics (recall that the SP is equivalent to our NTP). On the other hand, Wenger et al. (2023)'s longer training time implicitly induces rich training dynamics. To confirm this, we revisit Wenger et al. (2023)'s experiment in App. B.1. We show that longer training time and reasonably large learning rates induce more evolution of the NTK and thus more deviations from the lazy regime (as shown in (Fort et al., 2020)).

In summary, this first result highlights at least two new observations. First, scaling *per se* is neither positive nor negative, as its effect is intrinsically modulated by the network parameterization. Second, the difference between rich and lazy training regimes, i.e. feature learning, is a crucial factor in CF. To further study the large-width behavior of forgetting, we proceed to characterize the infinite-width dynamics of a model trained under non-stationarities in the feature learning regime.

### 4.1. Infinite Width Dynamics in Non-Stationary Training

We derive the dynamics under mean field scaling (see Tab. 1) of a continually learned two-layer non-linear network in the infinite-width limit using the framework of dynamical mean field theory (DMFT) of Bordelon & Pehlevan (2022). DMFT has been successfully employed to describe the network dynamics in both infinite-width (Bordelon & Pehlevan, 2022; Bordelon et al., 2024) and -depth limits (Bordelon et al., 2023), but never in the non-stationary setting. Due to space limitations, we defer more details, including the proofs, to Appendix C.

We are interested in tracking the evolution of the model dynamics in function space under gradient descent and Mean Squared Error (MSE) loss. As it will be clearer later, an important quantity that governs the dynamics and catastrophic forgetting is the *Neural Tangent Kernel* (NTK) *across tasks*, defined as:

$$K_{\alpha_i \beta_j}(t) = \sum_{\ell=1}^{L} \left\langle \nabla_{\mathbf{W}_\ell} h_{L+1}(\mathbf{x}_{\alpha_i}, t), \nabla_{\mathbf{W}_\ell} h_{L+1}(\mathbf{x}_{\beta_j}, t) \right\rangle,$$
(2)

where $\alpha_i \in \mathcal{T}_i$ and $\beta_j \in \mathcal{T}_j$ and $\langle \cdot, \cdot \rangle$ is the standard inner product in Euclidean space. In the presence of training with multiple tasks $\mathcal{T}_1, \ldots, \mathcal{T}_T$, the equations governing the dynamics at task $\mathcal{T}_i$ are:

$$\frac{d}{dt} f(\mathbf{x}_{\mu_i}) = \sum_{j=1}^{T} \tilde{U}_j(t) \sum_{\alpha_j \in \mathcal{T}_j} K_{\mu_i \alpha_j}(t) \Delta_{\alpha_j}(t). \quad (3)$$

We have introduced $\tilde{U}_j(t) = U(t - t_{j-1})U(t_j - t)$, where $U(t-t_{j-1}) = \mathbb{1}_{t > t_{j-1}}$ is the Heaviside step function, which we use to model the transitions from the $(j-1)$-th to the $j$-th task. $\Delta_{\alpha_j} = y_j^\alpha - f(\mathbf{x}_j^\alpha)$ is the negative residual and $K_{\mu_i \alpha_j}(t)$ is the NTK across tasks. We also introduce *gradient neurons* $\mathbf{g}_\alpha(t) \in \mathbb{R}^N$, defined as $\mathbf{g}_\alpha^\ell(t) = \sqrt{N} \frac{\partial h^{L+1}}{\partial \mathbf{h}^\ell}$, as well as the forward and backward kernels (also called feature and gradient kernels):

$$\Phi_{\alpha_i \beta_j}^\ell(t) = \frac{1}{N} \left\langle \phi(\mathbf{h}_{\alpha_i}^\ell(t)), \phi(\mathbf{h}_{\beta_j}^\ell(t)) \right\rangle \quad (4)$$

$$G_{\alpha_i \beta_j}^\ell(t) = \frac{1}{N} \left\langle \mathbf{g}_{\alpha_i}^\ell(t), \mathbf{g}_{\beta_j}^\ell(t) \right\rangle. \quad (5)$$

Intuitively, the feature (resp. gradient) kernel controls the effect of the weights on the geometry of the feature space in the forward (resp. backward) pass. Often, it will be more convenient to manipulate the *pre-gradient* variables $\mathbf{z}_{\mu_i}^\ell = \frac{1}{N} \mathbf{W}^{\ell\top} \mathbf{g}_{\mu_i}^{\ell+1}$. Then we have that $\mathbf{g}_{\mu_i}^\ell(t) = \dot\phi(\mathbf{h}_{\mu_i}^\ell) \odot z_{\mu_i}^\ell$. These quantities allow us to re-write the NTK across tasks as:

$$K_{\alpha_i \beta_j}(t) = \sum_{\ell=1}^{L} \Phi_{\alpha_i \beta_j}^{\ell-1}(t) G_{\alpha_i \beta_j}^\ell(t), \quad (6)$$

which indicates that the NTK is fully determined by the kernels $\Phi_{\alpha_i \beta_j}^\ell(t), G_{\alpha_i \beta_j}^\ell(t)$. This results in the following finite width dynamics, where the scaling factor $\gamma$ and its dependency to $\gamma_0$ and $N$ crucially differentiates NTP from $\mu$P:

$$\begin{aligned}
\boldsymbol{h}_{\mu_i}^1(t) &= \boldsymbol{\chi}_{\mu_i}^1(t) \\
&+ \frac{\gamma}{\sqrt{N}} \int_0^t \sum_{j=1}^{T} \tilde{U}_j(t) \sum_{\alpha_j} \Delta_{\alpha_j}(s) \boldsymbol{g}_{\alpha_j}^1(s) K_{\mu_i \alpha_j}^x \\
\boldsymbol{z}^1(t) &= \boldsymbol{\xi}^1(t) \\
&+ \frac{\gamma}{\sqrt{N}} \int_0^t \sum_{j=1}^{T} \tilde{U}_j(s) \sum_{\alpha_j} \Delta_{\alpha_j}(s) \phi(\boldsymbol{h}_{\alpha_j}^1(s))
\end{aligned} \quad (7)$$

In Proposition 4.1, we show that in the infinite width limit for $\mu$P there exists a set of variables including $\{h_{\alpha_i}^1, g_{\alpha_i}^1\}_{i,\alpha}$, $\{\Phi_{\alpha_i}^\ell, G_{\alpha_i}^\ell\}_{i,\alpha}$ that forms a close system of self-consistent (i.e. implicit) equations that fully describe the output $f$ in the large $N$ limit:

**Proposition 4.1** (Informal). *Let $f$ be the output of a neural network with a single hidden layer in Eq. 1 with $\alpha = 0$ (i.e. without skip connections) trained for $T$ tasks $\mathcal{T}_1, \ldots, \mathcal{T}_T$ for a finite number of steps on each task, and for $\mu$P. As $N \to \infty$, the pre-activations and pre-gradient variables $\{\mathbf{h}_{\mu_i}^1\}_{\mu,i}, z^1$ can be described as i.i.d. draws from marginal distributions described by the following stochastic processes:*

$$\begin{aligned}
h_{\mu_i}^1(t) &= \chi_{\mu_i}^1 \\
&+ \gamma_0 \int_0^t \sum_{j=1}^{T} \tilde{U}_j(s) \sum_{\alpha_j} \Delta_{\alpha_j}(s) g_{\alpha_j}^1(s) K_{\mu_i \alpha_j}^x ds \\
z^1(t) &= \xi^1 \\
&+ \gamma_0 \int_0^t \sum_{j=1}^{T} \tilde{U}_j(s) \sum_{\alpha_j} \Delta_{\alpha_j}(s) \phi(h_{\alpha_j}^1(s)) ds
\end{aligned} \quad (8)$$

*where $[\chi_{\mu_i}, \chi_{\beta_j}] \sim \mathcal{N}(0, [K_{\mu_i}^x, K_{\beta_j}^x])$ and $\xi \sim \mathcal{N}(0, I)$ are the Gaussian Processes characterizing the initialization. Furthermore, the dynamics of the output $f$, as well as the kernels $\Phi^1, G^1$ concentrate around their expectation:*

$$\lim_{N \to \infty} \Phi_{\alpha_i \beta_j}^1(t) = \mathbb{E}\left[ \phi(h_{\alpha_i}^1(t)) \phi(h_{\beta_j}^1(t)) \right] \quad (9)$$

$$\lim_{N \to \infty} G_{\alpha_i \beta_j}^1(t) = \mathbb{E}\left[ g_{\alpha_i}^1(t) g_{\beta_j}^1(t) \right]. \quad (10)$$

*where the expectation is taken over the marginals for $\{h_{\mu_i}^1\}_{\mu,i}, g^1$.*

**Remarks.** Notice that the preactivations are one-dimensional variables, and not vectors, as every unit can be seen as an i.i.d. draw from this marginal, which corresponds to the single site process in the infinite width limit.

Finally, the term in $\gamma_0$ in Equation (8) is the feature learning correction characterizing the time evolution from the initial process. If $\gamma_0 \rightarrow 0$, we recover the lazy regime (Jacot et al., 2018; Chizat et al., 2019), where it is clear from the equations that there is no time evolution of the hidden layer.

Proposition 4.1 provides an exact description of the network dynamics at infinite width. The self-consistent system of equations can be simulated to reproduce the features and function evolution during training. Here, we apply this result to characterize the dynamics of CF. For simplicity, take the definition of CF to be the following[1]:

$$CF(t) := \sum_{\alpha_1} \frac{1}{2}\Delta_{\alpha_1}^2(t) - \frac{1}{2}\Delta_{\alpha_1}^2(t_1) \qquad (11)$$

where $t > t_1$ and $\{\mathbf{x}_{\alpha_1}, y_{\alpha_1}\}_{\alpha_1}$ is the data of the first task. In the case of two tasks, it can be shown that the dynamics of forgetting are determined by the errors $\Delta$ and the NTK across tasks:

$$\frac{d}{dt}CF(t) = -\sum_{\mu_1\alpha_2}\Delta_{\mu_1}(t)K_{\mu_1\alpha_2}(t)\Delta_{\alpha_2}(t) \qquad (12)$$

This equation is very general, and it is valid for all degrees of feature learning $\gamma_0$ and all input distributions. In Appendix E, we use perturbation theory to isolate the effect of small $\gamma_0$ corrections to the lazy limit, up to second order.

In Fig. 2b, we report the asymptotic behavior of catastrophic forgetting on the loss under $\mu$P and NTP, at both finite and infinite-widths. For the infinite-width limit, we use the dynamics of Proposition 4.1 with $\gamma_0 = 1$ for $\mu$P and $\gamma_0 = 0$ for NTP. As expected, as the width is increased the CFr at finite sizes approaches the theoretical limit. More importantly, we again observe two separate behaviors depending on the parameterization, substantiating the claim that the *effect of scaling on CF depends on the training regime*.

## 5. Feature Learning and Catastrophic Forgetting in Non-Stationary Training

We now investigate the relationship between *feature learning* (in particular its presence or absence) and catastrophic forgetting. In particular, we ask whether *the presence of feature learning during training is conducive to CF*.

### 5.1. *Lazy-Rich* and *Low-High Forgetting* Transitions

We leverage the $\gamma_0$ parameter in the $\mu$P to smoothly interpolate between the two lazy ($\gamma_0 \rightarrow 0$) and rich ($\gamma_0 = 1$) training regimes. This methodology has already been employed to study phenomena tied to network training dynamics (Kumar et al., 2024; Atanasov et al., 2024).

[1]This quantity is also known as *backward transfer*.

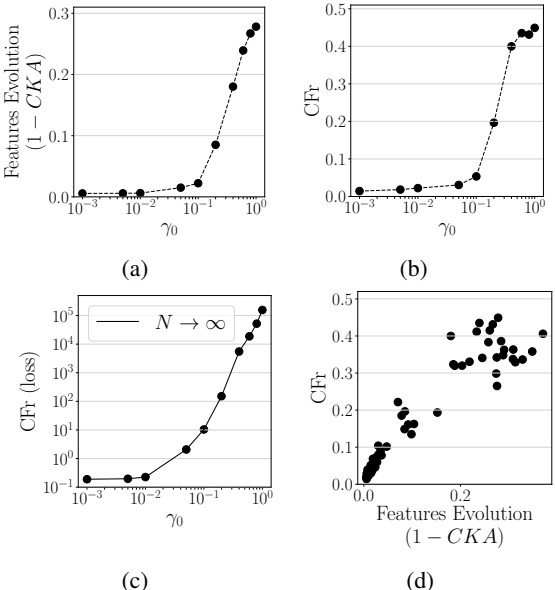

*Figure 3.* Entanglement between $\gamma_0$, feature evolution, and forgetting. (a) Non-linear relationship between $\gamma_0$ and feature evolution $(1 - CKA)$ in ResNet model ($N = 4096$, Split-CIFAR10); a transition happens at $\gamma_0 \approx \gamma_0^{LRT}$. (b) $\gamma_0$ and CFr (ResNet, $N = 4096$). (c) $\gamma_0$ and CFr at the infinite-width limit (MLP, permuted-MNIST). (d) The amount of feature evolution correlates with forgetting across various widths and values of $\gamma_0$ (ResNet, Split-CIFAR10).

We first measure the amount by which the features evolve during training by monitoring the internal representation of a task's data. More specifically, after training on a task, we look at the activation vectors at every residual block of the model of the same data while training on the remaining tasks. We compare activations in a permutation-invariant manner by measuring the cosine-alignment of the respective kernels, a measure known as *Centered Kernel Alignment* (CKA) (Kornblith et al., 2019). We plot the average over training and all tasks.

We uncover a surprisingly non-linear relationship between $\gamma_0$ and the evolution of features (Fig. 3a). In particular, we identify a critical *Lazy-Rich* Transition (LRT) region $\gamma_0 \approx \gamma_0^{LRT}$ that non-linearly separates two different behaviors: for $\gamma_0 < \gamma_0^{LRT}$ the relationship is mostly flat and the features change by very small amounts, signifying that the network is in an *effectively lazy* regime. Conversely, for $\gamma_0 > \gamma_0^{LRT}$, the features' evolution sharply increases with $\gamma_0$, thus the network is in a *rich* regime. We hypothesize that this transition could be related to our choice of LR scaling (Tab. 1, the LR scales quadratically with $\gamma_0$). Indeed, the concurrent work of Atanasov et al. (2024) observes that this scaling is optimal for the lazy regime, but not for the rich and ultra-rich regimes (their Fig. 1b), where instead sub-quadratic LR scaling is optimal. Hence, as we increase $\gamma_0$, our LR shifts from the optimal LR towards a *larger-than-optimal* LR, potentially triggering the sharp rise we observe.

We leave for future study the exploration of this interesting hypothesis.

The relationship between $\gamma_0$ and CFr is strikingly similar (Fig 3b): for $\gamma_0 < \gamma_0^{LRT}$ CFr is very low regardless of $\gamma_0$, while it grows significantly for $\gamma_0 > \gamma_0^{LRT}$. This result is confirmed consistently across different finite widths (Fig. 1b), as well as at the infinite-width limit (Fig. 3c). By analyzing the relationship between features evolution and CFr across scales and $\gamma_0$, we recover consistently that feature learning negatively impacts CF (Fig. 3d), with a strong (p-value $< 10^{-30}$) positive correlation between the amount of evolution in the features and CF. Taken together, these results convincingly point to the fact that *the presence of feature learning is indeed conducive to CF, and higher degrees of feature learning lead to higher levels of forgetting in non-stationary learning.*

**Remark on the *Lazy-Rich* Transition**   Curiously, this transition also appears to mark a shift in the behavior of width-scaling in forgetting. As visible in Fig. 1c, whereby for $\gamma_0 < \gamma_0^{LRT}$ width scaling improves CFr, while for $\gamma_0 > \gamma_0^{LRT}$ it does not. This appears to happen not only on Split-CIFAR10 and the ResNet, but also on the MLP setting of the infinite-width simulations (Fig. 11). Nevertheless, this *inversion* is not observed in the feature evolution but only in forgetting (cf. Fig. 13a). Our last remark hints at a fundamental shift in the interaction between width, $\gamma_0$, and training dynamics happening at the *lazy-rich* transition. This is a topic that deserves further investigation and it lies beyond the scope of this work. However, in Appendix B.5 we present preliminary results, looking at the changes in the loss landscape geometry as we vary $\gamma_0$ and width jointly.

## 5.2. Intermediate Feature Learning Optimizes the Stability-Plasticity Tradeoff

When considering the overall final performance, a trade-off between LA and CF naturally arises, commonly referred to as the *plasticity-stability dilemma*. Consolidated evidence in Deep Learning proves that higher degrees of feature learning help fit a task better (Fig. 1a), thus lifting the LA. However, our results so far have shown that increased feature learning also causes increased forgetting. Therefore we ask, *which degree of feature learning achieves the optimal trade-off between plasticity and stability?*

In Figures 1b and 4a we plot the average performance as a function of $\gamma_0$ in our Split-CIFAR (ResNet) and Permuted-MNIST (MLP with finite and infinite widths) experiments, respectively. The optimal performance is achieved at intermediate and relatively low $\gamma_0$-values: $\gamma_0^{\star} \approx 0.1$ for both benchmarks. Surprisingly, $\gamma_0^{\star}$ *transfers across widths*, which means that one can determine $\gamma_0^{\star}$ by tuning at low widths to then scale up without additional costs. A similar transfer of other hyperparameters has been observed in the scaling lim-

its literature (Yang et al., 2022a; 2023; Bordelon et al., 2023; Noci et al., 2024). The infinite-width simulation confirms our empirical findings with striking precision. Furthermore, we notice also that for $\gamma_0 > \gamma_0^{\star}$ the increased scale of the network is *wasted* as the larger width does not bring any benefits in performance.

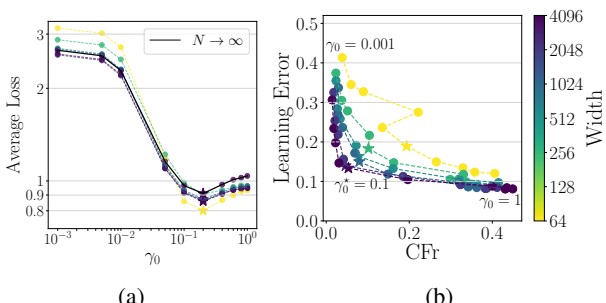

*Figure 4.* (a) Average Loss for varying $\gamma_0$ at finite and infinite widths (MLP, permuted-MNIST, 30 samples). The optimal $\gamma_0^{\star}$ transfers to all finite and infinite widths. (b) The plasticity-stability tradeoff is controlled by the amount of feature learning (ResNet, Split-CIFAR10).

Finally, we disentangle the effect of plasticity and stability from the compound metric of the average performance, by measuring them through the Learning Error and CFr, respectively. In Fig. 4b we observe that varying $\gamma_0$ allows us to navigate the stability-plasticity tradeoff, which generates a Pareto front. In particular, we see that at small $\gamma_0$ we have a low CFr, but also a high LE. Increasing $\gamma_0$, the LE first undergoes a rapid decrease without significantly impacting the CFr, striking the optimal tradeoff for $\gamma_0^{\star} \approx 0.1$. If we further increase $\gamma_0$, the CFr starts to sharply increase in return for a diminishing benefit of LE. We reproduce this tradeoff also with the MLP and the infinite-width simulation in Fig. 12. All in all, this experiment recovers the stability-plasticity dilemma and highlights that neither scaling nor feature learning allow the circumvention of the fundamental constraints imposed by this tradeoff.

## 5.3. Discussion on Depth Scaling

Mirzadeh et al. (2022a;b) show experimentally that depth, unlike width, worsens the performance of the model in CL. We can interpret this result in light of our findings on the relationship of feature learning and CF. In fact, Bordelon & Pehlevan (2022) proved that in NTP and $\mu$P depth scaling induces evolution of the NTK, therefore increasing the amount of feature learning. According to the results presented in this Section, increased feature learning comes at the cost of higher forgetting. Bordelon et al. (2023) and Yang et al. (2023) recently introduced a modified $\mu$P specifically for depth scaling, which allows scaling the network depth boundlessly while maintaining a depth-independent level of feature learning. In App. B.2, we perform scaling

experiments in depth with this modified parameterization. Our evidence, although of a preliminary nature, indicates that once again scaling in this rich regime does not reduce (nor increase) CF. Nevertheless, we leave a thorough characterization of this aspect for future study.

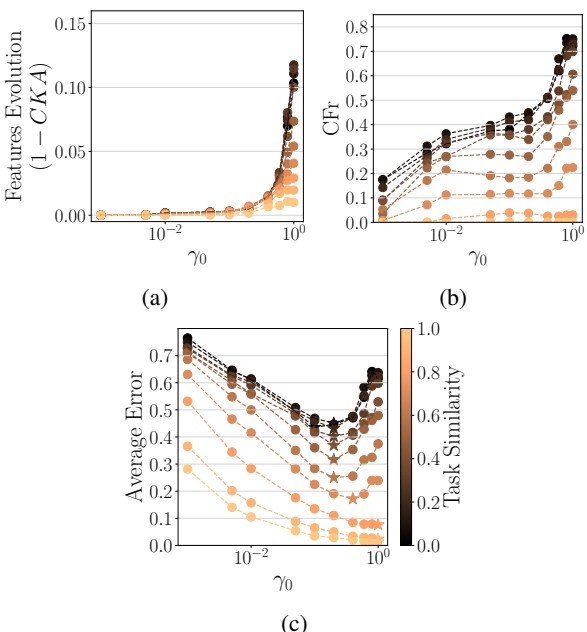

(a)

(b)

(c)

*Figure 5.* Results on the synthetic dataset Permuted-MNIST with varying levels of task similarity; a ResNet with a width of 4096 is used. (a) The evolution of features is modulated by the task similarity for fixed $\gamma_0$; $\gamma_0^{LRT}$ shifts towards 1 for higher task similarity. (b) Following the shift of the *lazy-rich* transition, forgetting sharply increases at a later $\gamma_0$ for higher task similarity levels. (c) The higher the task similarity, the more the optimal $\gamma_0^\star$ shifts towards 1.

## 6. Interpolating Between Stationary and Non-Stationary Data

In the previous sections, we have shown that what holds true for the stationary setting (that feature learning is beneficial for performance) does not hold true for the non-stationary setting. In this section we consider these two separate settings as two extremes of a continuous spectrum, directly linking our findings with consolidated knowledge. In particular, we interpolate between stationary and (fully) non-stationary scenarios by manipulating the data distribution.

### 6.1. Task Similarity, *Lazy-Rich* Transition and $\gamma_0^\star$

First, we control the non-stationarity in Permuted MNIST by varying the permutation size (see App. A.4). We measure the task similarity $\rho \in [0, 1]$ as a proxy for stationarity: the lower $\rho$ the greater the non-stationarity.

Analyzing the levels of features' evolution for varying degrees of task similarity (Fig. 5a) we observe that the amount

of non-stationarity directly impacts the amount of feature evolution: *higher task similarity induces lower feature evolution across all $\gamma_0$ values* [2]. In other words, the *lazy-rich* transition happens at a higher $\gamma_0$ for more stationary data streams, and thus the degree of feature learning $\gamma_0^\star$ which trades off optimally LA and CF moves closer to 1 (Fig. 5c). Thus in the stationary limit, the maximal performance is reached for $\gamma_0^\star = 1$.

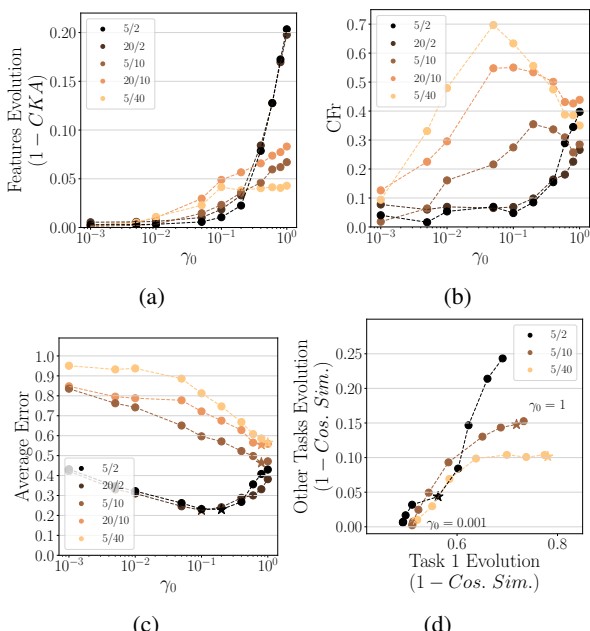

(a)

(b)

(c)

(d)

*Figure 6.* Results on Split-TinyImagenet with varying number of tasks and classes per task (e.g. 5/2 signifies 5 tasks of 2 classes each); a ResNet with a width of 1024 is used. (a) The task similarity shifts the *lazy-rich* transition. (b) CFr for varying $\gamma_0$; for many classes per task (5/40, 20/10), the CFr follows a non-monotonic relationship with $\gamma_0$. (c) The optimal level of feature learning $\gamma_0^\star$ shifts towards 1 with the task similarity, controlled by the number of classes per task. (d) Features evolution during the first, and later tasks with varying number of classes per task, on the Split-TinyImagenet dataset. When the tasks have many classes, the evolution of features on later tasks is constant in $\gamma_0$ suggesting features reuse and a *pretraining effect*.

### 6.2. The *Pretraining* Effect

Secondly, we control the non-stationarity in the TinyImagenet dataset by varying the number of classes per task (and the number of tasks) in the benchmark. We use the notation $t/c$ for a setting with $t$ tasks of $c$ classes each. Intuitively, increasing the number of classes per task enhances data diversity within each task, leading to greater overlap between task distributions and thus higher similarity between tasks.

Consistently with the evidence from Permuted MNIST, we observe higher feature evolution when the number of classes

---

[2]As a consequence, the assumption of lazy training dynamics in CL does not hold equally for all datasets and benchmarks.

per task is low – and so is the task similarity (Fig. 6a). This confirms that varying the number of classes per task controls the amount of non-stationarity, validating our methodology. Similarly to before, the increased task similarity allows a $\gamma_0^\star \approx 1$ to be optimal.

Intriguingly, in this setting alone we observe a non-monotonic relationship between the degree of feature learning and CF at the higher end of the range of non-stationarity tested (Fig. 6b). In fact, for these high-similarity tasks, increasing feature learning beyond a certain threshold appears to *decrease* the CFr. We hypothesize that this effect occurs when the features learned on the first task are useful for later tasks as well, thereby attenuating the amount of feature evolution in later tasks. We call this effect the *pretraining effect*, implying that the first task training acts similarly to pretraining, by finding features that transfer to later tasks.

To verify this hypothesis we compare the evolution of features before and after training on a certain task (with cosine similarity). We differentiate the evolution during the first task, from that of later tasks (Fig. 6d). We observe that the feature trajectories overlap at lower values of $\gamma_0$ for the different combinations of classes per task. However, they separate for higher values of $\gamma_0$: in the 5/2 curve the evolution during the first task is proportional to that of later tasks, whereas at the other extreme of the task similarity, the later tasks exhibit no further feature learning beyond a certain $\gamma_0$. In other words, the network switches to an effectively lazy regime after the first task. Additionally, we note that width is strictly beneficial both in terms of forgetting and final performance when there is the pretraining effect (App. B.6), reproducing the results of Ramasesh et al. (2022) on the entangled relationship between scale and pretraining.

## 7. Conclusions

Our work takes a decisive step towards understanding the roles of feature learning and scale in continual learning, as well as their interplay with task similarity. We show that there is no intrinsic benefit of scale and that the degree of feature learning is ultimately responsible for forgetting. We observe a non-linear transition in the relationship between feature learning and the parameterization factor $\gamma_0$. We extend DMFT in infinitely wide NNs to the treatment of multiple tasks. Our theoretical simulations confirm that the phenomena observed at finite widths are still valid at infinite widths. Finally, our findings add to the evolving perspective on feature learning in modern NNs, challenging the common "more feature learning is better" narrative, and underscoring the importance of modeling non-stationarity.

We foresee that our findings could be of practical guidance: the optimal degree of feature learning $\gamma_0^\star$ can be tuned only for small models and then transferred to large scales. In this context, our results can be inserted in the recent line of work of zero-shot hyperparameter transfer (Yang et al., 2022a; 2023; Bordelon et al., 2023; Lingle, 2024; Bjorck et al., 2024). Future work might also investigate mitigation methods (like experience replay (Chaudhry et al., 2019) and EWC (Kirkpatrick et al., 2017a)) and their scaling properties. In this setting, devising parameterizations that achieve a well-defined feature learning limit can lead to NNs that better navigate the plasticity-stability frontier, with the end goal of achieving maximal feature learning without forgetting at scale.

## Acknowledgements

AB would like to thank Marco Baiesi for helpful feedback and comments. LN would like to acknowledge the support of a Google PhD research fellowship. GL would like to acknowledge the support of the AI Center PhD fellowship.

## Impact Statement

This paper presents work whose goal is to advance the field of Machine Learning. There are many potential societal consequences of our work, none of which we feel must be specifically highlighted here.

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

# A. Experimental Details

## A.1. Evaluation Metrics

We report here the formal definitions of the evaluation metrics introduced in Sec. 3.1.1.

For a benchmark with $T$ tasks, we define the test accuracy on the task $\mathcal{T}_i$ after training on $\mathcal{T}_j$, with $i, j \in \{1, ..., T\}$, as $a_{j,i}$. We use the Learning Accuracy as measure of plasticity.

**Definition A.1** (Learning Accuracy (LA)). The *Learning Accuracy* (LA) is the average over all tasks of the accuracy on the task it has just been trained on:

$$\text{LA} = \frac{1}{T} \sum_{i=1}^{T} a_{i,i}. \tag{13}$$

Similarly, the complement of the LA is the Learning Error.

**Definition A.2** (Learning Error (LE)). The *Learning Error* (LE) is the complement of the LA:

$$LE = 1 - LA \tag{14}$$

Forgetting is traditionally measured as the average drop in accuracy.

**Definition A.3** (Catastrophic Forgetting (CF)). The Catastrophic Forgetting is the average drop in accuracy after training on all other tasks:

$$\text{CF} = \frac{1}{T-1} \sum_{i=1}^{T-1} \max_{t \in \{i,...,T-1\}} (a_{t,i}) - a_{T,i}. \tag{15}$$

The limitations of the CF are more easily comprehended with a simple toy example. Let us consider two models trained on a benchmark with only two tasks: the first model reaches 100% accuracy on the first task, and after training on the second task it drops to 80%, i.e. it has a CF of 20%. The second model, instead, reaches 40% accuracy on the first task, and after training on the second task it drops to 30%, i.e. it has a CF of 10%. Looking only at the CF, the second model would be favored but, in reality, the relative drop in accuracy is higher for the second model. Therefore, we introduce a novel metric that measures the Catastrophic Forgetting rate.

**Definition A.4** (Catastrophic Forgetting Rate (CFr)). The Catastrophic Forgetting Rate is the average *relative* drop in accuracy after training on all other tasks:

$$\text{CFr} = \frac{1}{T-1} \sum_{i=1}^{T-1} \frac{\max_{t \in \{i,...,T-1\}} (a_{t,i}) - a_{T,i}}{\max_{t \in \{i,...,T-1\}} (a_{t,i})}. \tag{16}$$

If we consider the CFr for the two models in the example above, we would see that the first model has a CFr of 20%, while the second model has a CFr of 25%, rightly favoring the first model.

**Definition A.5** (Average Accuracy (AA)). The *Average Accuracy* is the average accuracy on all tasks after all of them have been trained sequentially:

$$\text{AA} = \frac{1}{T} \sum_{i=0}^{T-1} a_{T-1,i}. \tag{17}$$

**Definition A.6** (Average Error (AE)). The *Average Error* is the complement of the AA.

$$AE = 1 - AA \tag{18}$$

We define also the same metrics in terms of the training loss, needed for the experiments with the infinite-width simulations. Concretely, for a benchmark with $T$ tasks, we define the training loss on the task $\mathcal{T}_i$ after training on $\mathcal{T}_j$, with $i, j \in \{1, ..., T\}$, as $\mathcal{L}_{j,i}$.

**Definition A.7** (Learning Loss (LL)). The *Learning Loss* (LL) is the average over all tasks of the loss on the task it has just been trained on:

$$\text{LL} = \frac{1}{T}\sum_{i=1}^{T} \mathcal{L}_{i,i}. \tag{19}$$

**Definition A.8** (Catastrophic Forgetting (CF) (loss)). The Catastrophic Forgetting in terms of loss is the average drop in loss after training on all other tasks:

$$\text{CF (loss)} = \frac{1}{T-1}\sum_{i=1}^{T-1} \max_{t\in\{i,\dots,T-1\}}(\mathcal{L}_{t,i}) - \mathcal{L}_{T,i}. \tag{20}$$

**Definition A.9** (Catastrophic Forgetting Rate (CFr) (loss)). The Catastrophic Forgetting Rate in terms of loss is the average *relative* drop in loss after training on all other tasks:

$$\text{CFr (loss)} = \frac{1}{T-1}\sum_{i=1}^{T-1} \frac{\max_{t\in\{i,\dots,T-1\}}(\mathcal{L}_{t,i}) - \mathcal{L}_{T,i}}{\max_{t\in\{i,\dots,T-1\}}(\mathcal{L}_{t,i})}. \tag{21}$$

**Definition A.10** (Average Loss (AL)). The *Average Loss* is the average loss on all tasks after all of them have been trained sequentially:

$$\text{AL} = \frac{1}{T}\sum_{i=0}^{T-1} \mathcal{L}_{T-1,i}. \tag{22}$$

## A.2. Model Architecture

We use an architecture of the ResNet family (He et al., 2016). In particular, we first apply a convolutional layer with kernel size 7x7 and $N$ filters with stride 2, followed by Batch Normalization (BN), ReLU, and max pooling. Then, we repeat 3 times the following configuration: $L/3$ residual blocks (each with 1 convolutional layer with N channels, BN, and ReLU), with the first block having a stride of 2. Finally, we apply a global average pooling, flatten the output, and apply a linear layer. We will simply call $N$ the *width*, and $L$ the *depth* of the model. The base configuration uses $N = 64$ and $L = 6$. For some of the CL datasets considered, a separate classification head is used for each task (see Sec. A.4).

Experiments involving the infinite-width simulation are executed on a 2-layer non-linear (ReLU) perceptron.

## A.3. Parameterization Details

Note that many equivalent parameterizations can achieve the same functional behavior (Yang & Hu, 2020; Bordelon & Pehlevan, 2022; Yang et al., 2023; Bordelon et al., 2023). In Tab. 1 we report the notation of (Bordelon et al., 2023, Tab. 1), but with the NTP from (Yang & Hu, 2020, Tab. 1) for implementational simplicity. Also, note that the standard parameterization (SP) of PyTorch (i.e. the SP column in (Bordelon et al., 2023, Tab. 1)) is equivalent to the NTP used here.

## A.4. Datasets and Training Details

**ResNet Experiments**  In all experiments – if not otherwise specified – we use Stochastic Gradient Descent (SGD) optimization without momentum nor weight decay. To find the learning rate $\eta_0(0)$, we do a hyperparameter search on the model at base width and depth (i.e. where all parameterizations are equivalent) on the full (i.e. non-CL) dataset taking the optimal test accuracy as a metric. We use a cosine learning rate schedule without a warmup, restarting the learning rate at the beginning of each task. For all datasets, we use a batch size of 128.

**Infinite-Width Experiments on 2-layer non-linear MLP**  In the infinite width limit simulations and finite-width comparisons, due to limited computational resources, was used a small set of 30 samples of the MNIST dataset (see below), three per class. This is because simulations involving the theoretical limit scale with $P^2T^2$ in terms of memory and of $P^3T^3$ in terms of time (Bordelon & Pehlevan, 2022), since kernels are of order $P^2T^2$ and undergo a matrix-vector product in

dimension $P$ and an integration over $T$ time steps. The simulations rely on drawing the random initial conditions of the kernels, namely the random initial fields, from a sampled distribution of size $S = 3000$ of the relative Gaussian Process describing their infinite width limit. From those, we implemented an Euler-based, discrete-time ODE solver to obtain the dynamics of all the fundamental quantities describing the network, including kernels and residuals. The MLP is optimized with full batch Gradient Descent on MSE loss with a LR tuned with a grid search on the real network on the non-CL data. To avoid the explosion of the initialization output at low values of $\gamma_0$, we initialize the last layer of the MLP to 0 to have a well-defined output of 0 at $t = 0$. This practice is advised in (Yang et al., 2022b, App. D.2).

All training runs and experiments are executed either on a single NVIDIA GeForce RTX 4090 or on an NVIDIA RTX A6000.

### A.4.1. SPLIT-CIFAR10

The Split-CIFAR10 (Zenke et al., 2017) (TIL type of benchmark) dataset has 5 tasks of 2 classes each (i.e. the 10 classes of CIFAR10 are split into 5 tasks with non-overlapping classes), and as common for TIL benchmarks, the model uses a separate head for each task. We train for 5 epochs on each task with a learning rate of $\eta_0(0) = 30.0$. Note that the LR is larger than usual values; this is due to the particular choice of parameterization. If not otherwise specified, all experiments on Split-CIFAR10 are repeated 5 times with different random seeds, and the average is reported.

### A.4.2. PERMUTED MNIST

Inspired by previous works (Goldfarb et al., 2024; Kirkpatrick et al., 2017b), we use the permuted input MNIST dataset with 5 tasks to investigate the impact of task similarity on CF. Specifically, each task of this benchmark consists of the MNIST dataset with a random but fixed permutation of the pixels.

In particular, we consider task similarity as the fraction of pixels that are not permuted between tasks, and the inner square of the image is permuted first. As an example, a task similarity of 1.0 corresponds to the original MNIST dataset; a task similarity of 0.0 corresponds to a dataset where each task permutes all pixels of the images; a task similarity of 0.5 corresponds to tasks where the middle square containing 50% of the pixels is permuted. These tasks are synthetic, and except for the untouched pixels, the tasks are not related to each other: this allows us to artificially cause CF by design.

**ResNet Experiments**   Each task is trained for 5 epochs with $\eta_0(0) = 2.0$. We do a slight modification of the architecture presented above, by omitting batch normalization layers (following the observation of Mirzadeh et al. (2022b)). This is a DIL type of benchmark, where the model uses a single head for all tasks.

**Infinite-Width Experiments on 2-layer non-linear MLP**   For this simplified setup, instead, we use a fixed task similarity of $\rho = 0$ (all pixels permuted) and 2 tasks. We optimize utilizing a small subset of training data comprising 30 samples (3 samples per label). Each task is optimized for 1000 epochs and a fixed LR of $\eta_0 = 0.25$.

### A.4.3. SPLIT-TINYIMAGENET

Similar to Split-CIFAR10, the classes of TinyImagenet are split into tasks with non-overlapping classes (Gong et al., 2022) (i.e. it is a TIL benchmark and we use separate classification heads for each task). Throughout the experiments, we consider varying the number of tasks and classes-per-task: 5 tasks of 2 classes each (we denote it 5/2), 5/10, 5/40, as well as 20 tasks of 2 classes each (20/2), and 20/10. We optimize each task for 10 epochs with $\eta_0(0) = 15.0$.

# B. Additional Experiments

## B.1. Duration of Training and Feature Learning

Motivated by the observations of Wenger et al. (2023), in Fig. 7 we investigate the effect of the training duration on feature evolution, forgetting, and performance. As shown by (Fort et al., 2020), we observe that longer training time increases the deviation from the lazy regime as features evolve more (Fig. 7a). As thoroughly presented above, more feature evolution reflects also on increased forgetting (Fig. 7b). As observed in Wenger et al. (2023), the benefit of width-scaling and thereof-induced laziness is reduced for longer training when the deviations from the NTK regime are significant. As done by $\gamma_0$ in the $\mu$P, the training duration also impacts the balance between plasticity and stability (Fig. 7c), inducing a critical and width-dependent training-duration that optimizes the Average Error (Fig. 7d).

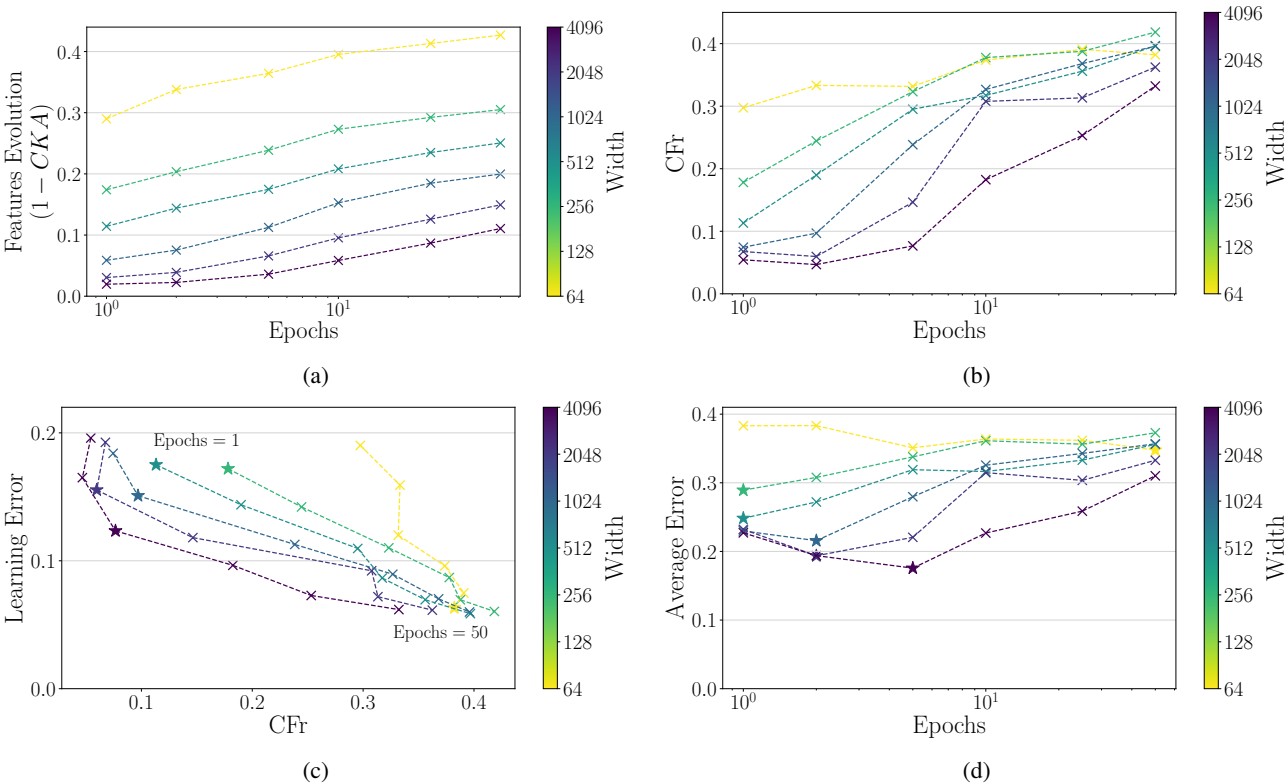

(a)  (b)

(c)  (d)

*Figure 7.* The effect of width and training duration with the NTP on Split-CIFAR10. The longer the training, the higher the feature evolution (a). Consequently, forgetting increases (b). Due to the plasticity-stability tradeoff (c), the Average Error shows a width-dependent optimal training duration (c).

## B.2. Depth Scaling

The role of depth was analyzed in (Mirzadeh et al., 2022a;b), where it was found that – in their setting – depth would not provide any benefit in terms of CF. With the precepts of scaling limits, and the observations done so far, we can interpret the reasons for this result: if not suitably parameterized, the depth augments feature learning by stimulating the evolution of the NTK (Bordelon & Pehlevan, 2022).

This effect of depth has been recently and concurrently addressed by Bordelon et al. (2023) and Yang et al. (2023); for notational simplicity, we will hereby consider the $\mu$P$+1/\sqrt{L}$ parameterization introduced in (Bordelon et al., 2023). This parameterization guarantees that the feature learning magnitude is decoupled from both width and depth. We have therefore a tool to disentangle CFr and the impact of depth, namely without the confounding effect of varying degrees of feature learning and laziness. Concretely, and following the notation in Tab. 1, $\mu$P$+1/\sqrt{L}$ differs from $\mu$P in the branch scale $\beta_l$, which depends not only on the width $N$, but also on the depth $L$ as follows:

$$\beta_l = \begin{cases} N^{-1/2}, & l = L \\ (LN)^{-1/2}, & 0 < l < L \\ D^{-1/2}, & l = 0 \end{cases}.$$ (23)

We empirically verify this hypothesis, fixing the width at 64 and varying the depth of the model in the three parameterizations. Here we use $\gamma_0 = 1.0$ for $\mu$P and $\mu$P$+1/\sqrt{L}$. Firstly we can see in Fig. 8 that the $\mu$P and NTP are equivalent since they are at base-width and share the same depth-scaling properties. We observe that initially the CFr increases, and then the stability of the network decays, with the learning error increasing until random performance. This yields an average error that significantly degrades with depth in the $\mu$P and NTP. Instead, the behavior of $\mu$P$+1/\sqrt{L}$ is different: the CFr is stable with depth (similarly as width for the $\mu$P), and the learning error is constant at a good performance. This yields an average error that is stable across depths.

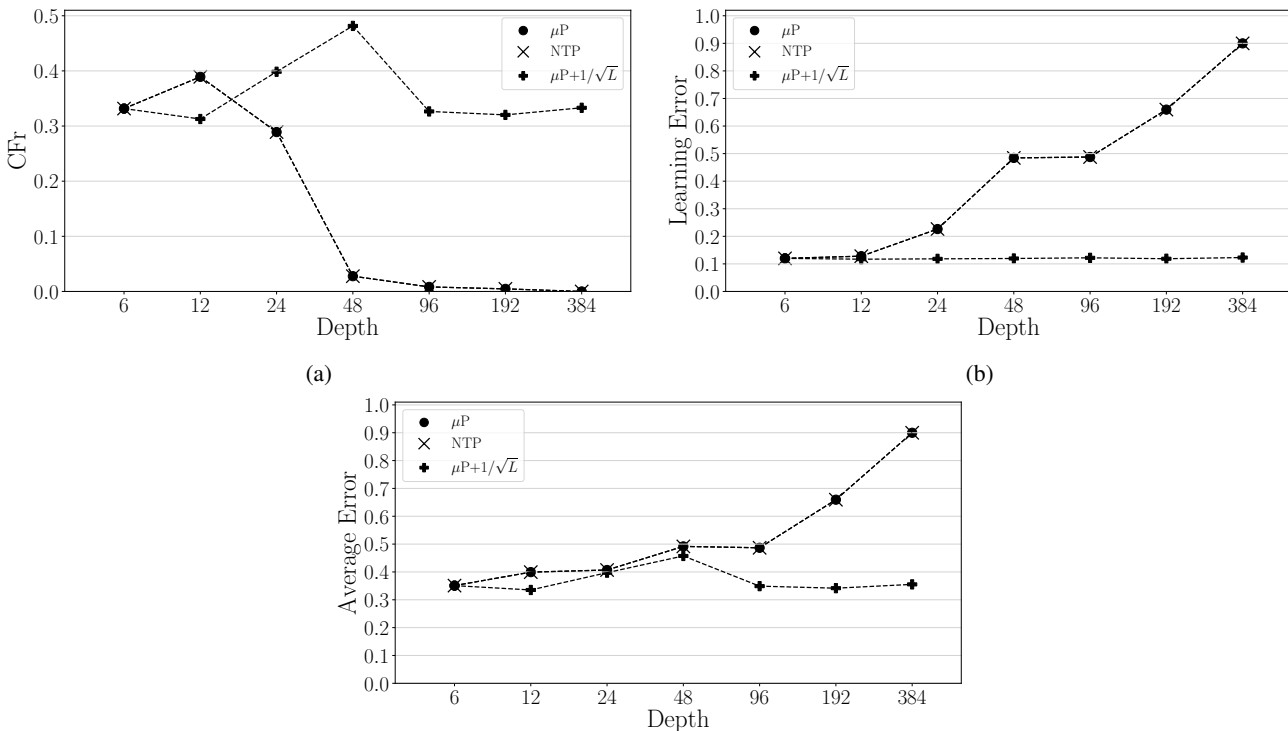

*Figure 8.* Performance of the three parameterizations with varying depth and a fixed width of 64. The $\mu$P and NTP are equivalent since the experiment is at base-width and they share the same parameterization w.r.t. the depth; $\gamma_0 = 1$. (a) CFr; (b) learning error; (c) average error.

Finally, we repeat the experiment varying the degree of feature learning at different depths; we fix the width at 512 and vary the depth of the model for various $\gamma_0$ values. Firstly, we notice that feature learning negatively impacts CFr for all depths in both $\mu$P and $\mu$P$+1/\sqrt{L}$ (Fig. 9 and Fig. 10, respectively). Furthermore, the curves for $\mu$P highly vary for different depths, namely we find a depth-dependent $\gamma_0^\star$ (Fig. 9). The curves for $\mu$P$+1/\sqrt{L}$, instead, are extremely stable for all depths, and in particular, the $\gamma_0^\star$ is constant across depths (Fig. 10): the optimal $\gamma_0^\star = 0.1$ transfers not only across widths (as found in Sec. 5) but – for $\mu$P$+1/\sqrt{L}$ – also across depths (as shown by previous works for non-CL hyperparameters (Yang et al., 2023; Bordelon et al., 2023)).

Curiously, on the Split-CIFAR10 considered here, depth does not provide any plasticity improvement (i.e. learning error). We hypothesize that this is due to the simplicity of the CIFAR10 dataset, where the model can easily learn the task at hand with a shallow architecture, and where the additional depth does not provide any benefits. Evaluations on more challenging datasets might provide further insights. However, a thorough investigation of the role of depth is out of the scope of this work, and we leave further experiments to future work.

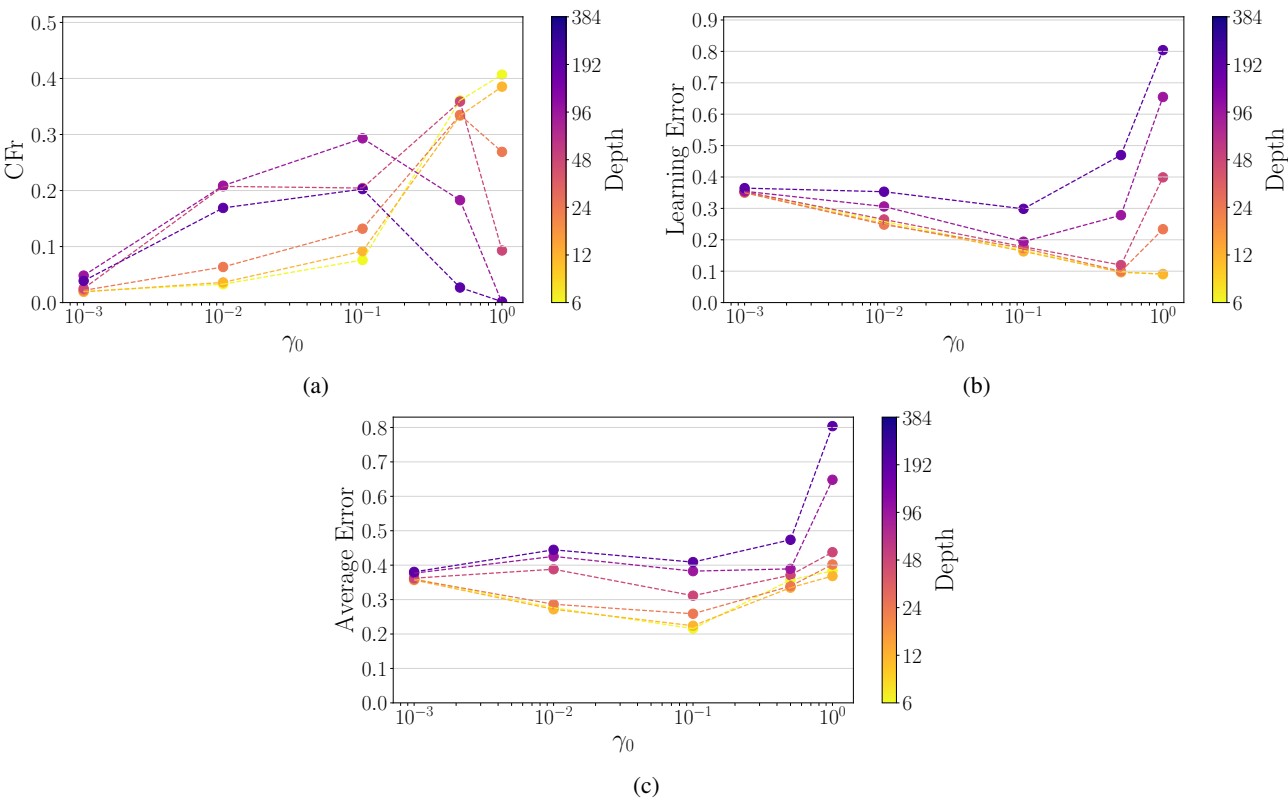

*Figure 9.* Depth scaling of the $\mu$P, with a fixed width of 512 and various degrees of feature learning. (a) CFr; (b) learning error; (c) average error.

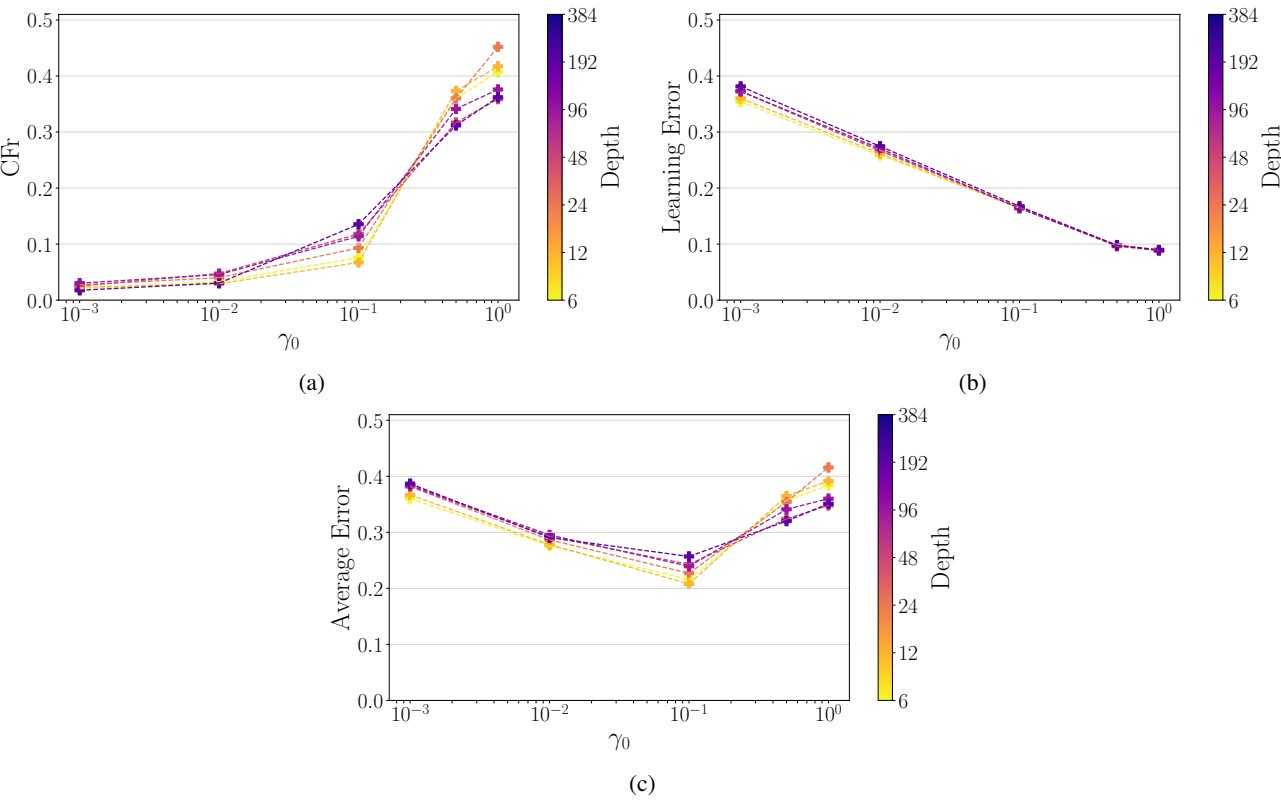

*Figure 10.* Depth scaling of the $\mu$P+$1/\sqrt{L}$, with a fixed width of 512 and various degrees of feature learning. (a) CFr; (b) learning error; (c) average error.

## B.3. Additional Experiments with the Infinite-Width Limit

Here we report additional experiments on the MLP with the infinite-width limit simulation.

In Fig. 11 we report the curves of CFr with varying $\gamma_0$ for finite and infinite width. We recover also with this setup, the characteristic relationship observed in Fig. 1c and analyzed in Sec. 5.1. Furthermore, we note that in this case too, the *lazy-rich* transition seems to differentiate two different width-scaling regimes, where for $\gamma_0 < \gamma_0^{LRT}$ the infinite-width limit is approached from above and thus width scaling reduces forgetting (Fig. 11b), while for $\gamma_0 > \gamma_0^{LRT}$ the infinite-width limit is approached from below and thus width scaling increases forgetting (Fig. 11c). We leave the characterization of the finite-width convergence to the infinite limit for future work.

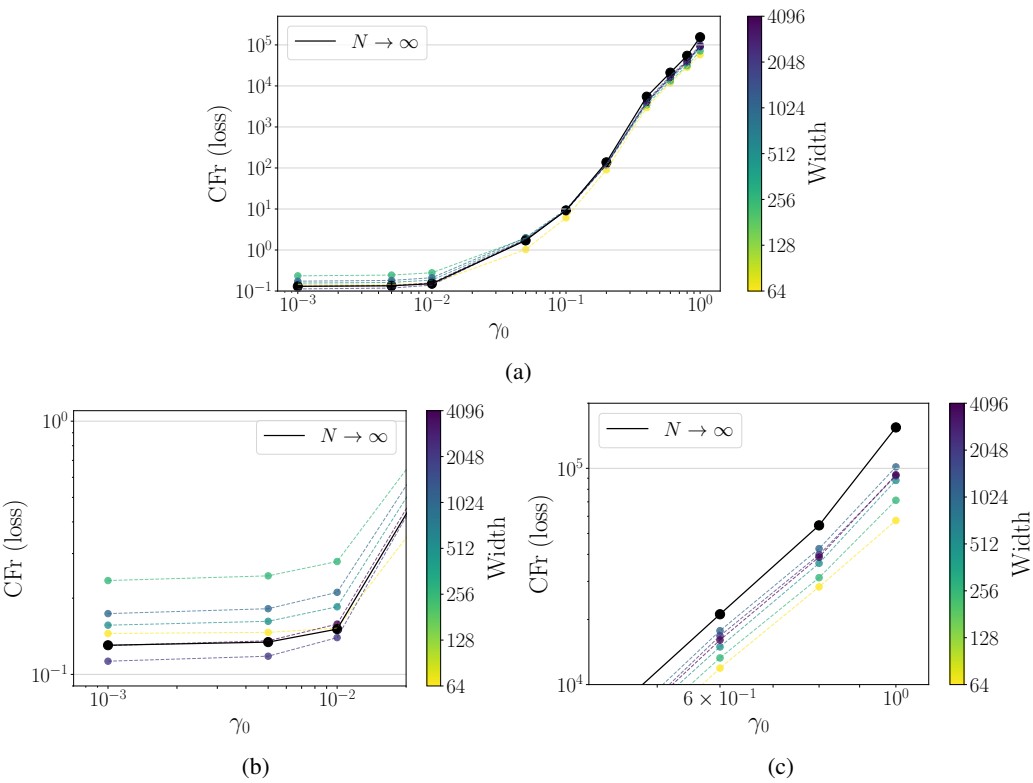

*Figure 11.* (a) Finite and infinite-widths relationship between $\gamma_0$ and CFr (loss). Zoomed in versions for (b) low $\gamma_0$ and (c) high $\gamma_0$.

In Fig. 12 we recover even for the MLP and infinite-width simulation the plasticity-stability tradeoff (Fig. 4b). This further suggests that even at the infinite-width limit NNs (trained with SGD) suffer from the plasticity-stability tradeoff.

## B.4. Features Evolution at finite widths

In Fig. 13 we report the counterpart of Fig 3a for varying finite widths.

## B.5. Loss Landscape Analysis

Motivated by the theoretical and empirical verifications of Mirzadeh et al. (2020), uncovering the crucial role of the geometrical properties of the landscape for forgetting, we investigate its peculiarities across parameterizations and scaling the width. This approach has also proven to be an important tool to uncover networks' behavior in multiple studies (Sagun et al., 2017; Martens, 2020). In particular, Noci et al. (2024) has recently uncovered crucial differences in the loss landscape of the NTP and $\mu$P.

The *sharpness* of the landscape, i.e. the maximum eigenvalue of the Hessian, is a crucial property of the loss landscape. Recently, Cohen et al. (2021) have observed an intriguing mechanism when optimizing deep networks with (full-batch) gradient descent: the models exhibit a rapid increase in sharpness towards a critical point. This increase is termed *progressive*

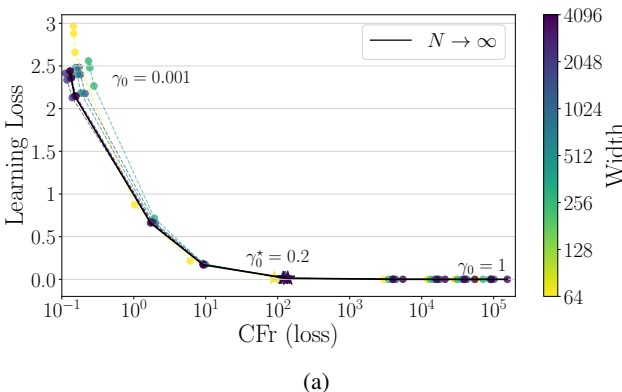

(a)

*Figure 12.* Plasticity (Learning Loss) and stability (CFR (loss)) tradeoff with finite and infinite-width simulation.

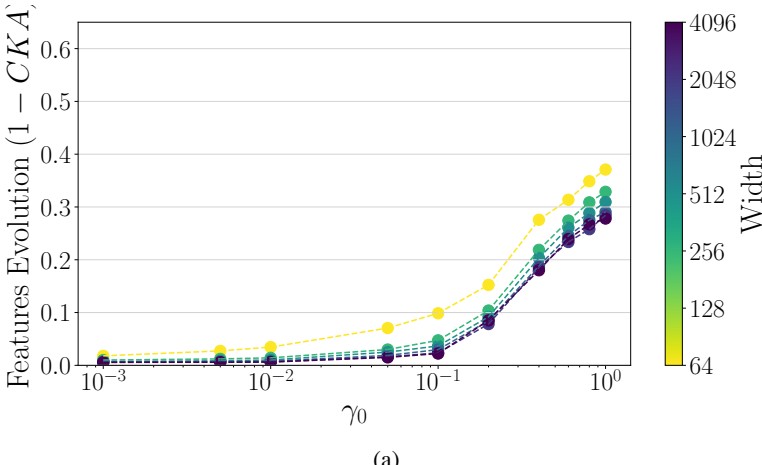

(a)

*Figure 13.* The *lazy-rich* transition: above a certain threshold of $\gamma_0$ the evolution sharply increases consistently for all widths (ResNet model on Split-CIFAR10).

*sharpening*, and the critical point is the *Edge of Stability* (EoS). Interestingly, the dynamics of the model do not diverge even beyond the EoS, thanks to a mechanism of Self-Stabilization (Damian et al., 2022). These observations partly apply also to our case of mini-batch stochastic optimization (Cohen et al., 2021; Noci et al., 2024).

We define formally the sharpness as follows:

**Definition B.1** (Sharpness). The sharpness of the loss landscape at a point $\mathbf{W}$ is defined as the maximum eigenvalue of the Hessian of the loss function at $\mathbf{W}$:

$$\text{Sharpness}(\mathbf{W}) = \lambda_{\max}(\nabla^2 \mathcal{L}(\mathbf{W})) = \lambda_{\max}(\mathbf{H}(\mathbf{W})). \tag{24}$$

We will abbreviate the sharpness as $\lambda_{max}$.

The EoS threshold arises from the convergence requirement for the gradient descent optimization with learning rate $\eta$ of a quadratic function, namely that the sharpness of the Hessian should not exceed $2/\eta$.

As in $\mu$P the learning rate $\eta$ scales with the width, the standard sharpness of Def. B.1 is hardly representative of the actual model behavior. In particular, as the LR increases – by the EoS – the sharpness of the landscape would decrease proportionally. However, from a practical perspective and from the point of view of optimization, the lower sharpness and the proportionally higher learning rate would compensate for each other. Intuitively and qualitatively, the valley of the optimization becomes flatter, but the velocity of the optimization increases, yielding a constant *effective sharpness*. To compensate for this effect, we introduce the effective sharpness as a scaled sharpness.

**Definition B.2** (effective Sharpness). The effective sharpness for a model parameterized with the $\mu$P as in Tab. 1, is defined as

$$\text{effective Sharpness}(\mathbf{W}) = \lambda_{\max}(\gamma^2 \cdot H(\mathbf{W})). \tag{25}$$

We will abbreviate the effective sharpness as $\tilde{\lambda}_{max}$. For $\mu$P, we will often refer to the effective sharpness as the sharpness for simplicity.

Crucially for our analysis, Noci et al. (2024) have shown that the NTP and $\mu$P have different (effective) sharpness properties when scaling the width: $\mu$P exhibits consistent effective sharpness at all widths, while for the NTP the sharpness dynamics are width-dependent, and are characterized by the inversely proportional relationship between sharpness and width (for short enough training). With these resulting differences, we study the landscape in the context of CL for the two parameterizations.

The trace of the Hessian provides another meaningful perspective on the curvature of the loss landscape. However, comparing the trace across models of varying dimensions (e.g., as we scale their widths) is inherently challenging, as it represents a sum over an increasing number of terms. One possible approach is to normalize the trace by the number of model parameters. However, this metric can become unreliable when the number of zero eigenvalues in the Hessian scales with the parameter number, causing the normalized trace to decrease regardless of the behavior of the non-zero eigenvalues. To date, the literature on experimental analysis of loss landscapes has not resolved these challenges, and our results should therefore be considered preliminary.

**Definition B.3** (Normalized Trace and effective Normalized Trace). For $\mathbf{W} \in \mathbb{R}^{|W|}$, the normalized trace of the Hessian is defined as

$$\text{Normalized Trace}(\mathbf{W}) = \frac{1}{|W|} \cdot \text{tr}(\nabla^2 \mathcal{L}(\mathbf{W})) = \frac{1}{|W|} \cdot \text{tr}(\mathbf{H}(\mathbf{W})). \tag{26}$$

The effective normalized trace for a model in the $\mu$P (Tab. 1) is defined as

$$\text{effective Normalized Trace}(\mathbf{W}) = \frac{1}{|W|} \cdot \text{tr}(\gamma^2 \cdot \mathbf{H}(\mathbf{W})). \tag{27}$$

We analyze multiple compound metrics of the landscape based on these fundamental quantities on Split-CIFAR10. Firstly, we observe the average effective sharpness and normalized trace at convergence, i.e. the average of these quantities at the end of training on each task and with the data of that same task. Lastly, we want to gain an insight into the multi-task loss landscape, i.e. of the compound loss of all the tasks. To do so, we measure the average effective sharpness and trace at the end of training on each task, but using data from all previous tasks.

The sharpness and normalized trace of the Hessian are calculated using one batch of training data with the PyHessian library (Yao et al., 2020).

### B.5.1. $\gamma_0$ MODULATES THE LANDSCAPE

Firstly, we observe that the average sharpness of the converged models is modulated by $\gamma_0$ but, in contrast to the findings of Noci et al. (2024), the width has a larger impact on the converged sharpness hinting at the complex influence that learning a chain of tasks has on the optimization properties (Fig. 14a). A change of regime is clearly distinguishable at $\gamma_0 \approx 0.1$. Similar behaviors are also observed for the average sharpness of the multi-task optimization (Fig. 14b), shedding an important insight for our CF problem: a higher curvature of the landscape is observed for higher feature learning, meaning that the multi-task performance is more sensitive to changes in the network parameters.

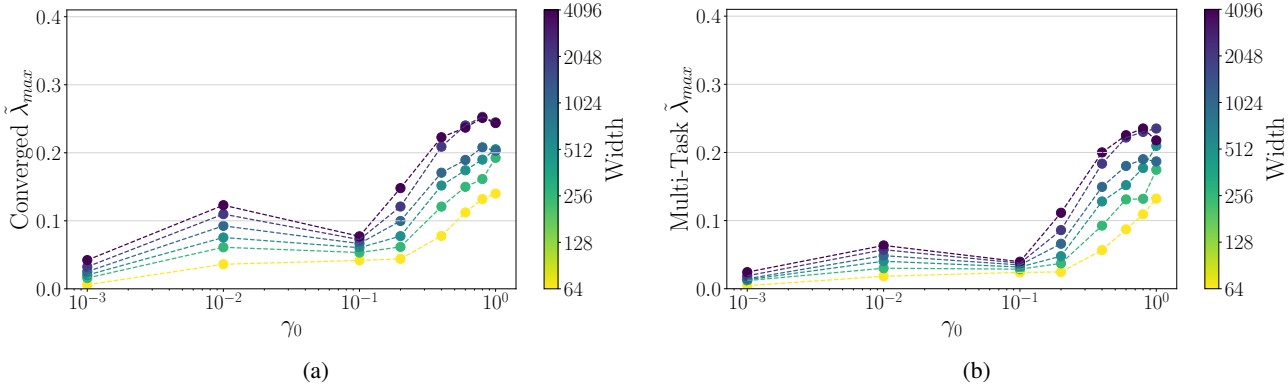

Figure 14. Average effective sharpness in the $\mu$P for varying $\gamma_0$ and width. (a) Converged; (b) multi-task.

A second insight is gained by analyzing the effective normalized trace of the Hessian, finding that it is modulated by both $\gamma_0$ and width in a non-trivial fashion (Fig. 15) and with a clear phase transition at $\gamma_0 \approx 0.1$. This is particularly pronounced for the multi-tasks trace: for low $\gamma_0$ the trace is positive and small (signaling a convex and wide valley), and at $\gamma_0 \approx 0.2$ a breakdown occurs, with the trace becoming negative and large the wider the model. This means that the landscape becomes non-convex and with high curvature, i.e. for the multi-task objective the optimization reaches a non-minimum point in the landscape (a saddle point or even a maximum). This is yet another hint at the ill-conditioning of the landscape for higher feature learning in CL and a potential cause for the increased forgetting observed for higher $\gamma_0$. Furthermore, this seems to indicate that higher widths worsen the bad conditioning of the landscape, possibly hinting at a reason for the negative effect of width on CFr observed in Sec. 5.1 (Fig. 1c) for $\gamma_0 > \gamma_0^{LRT}$.

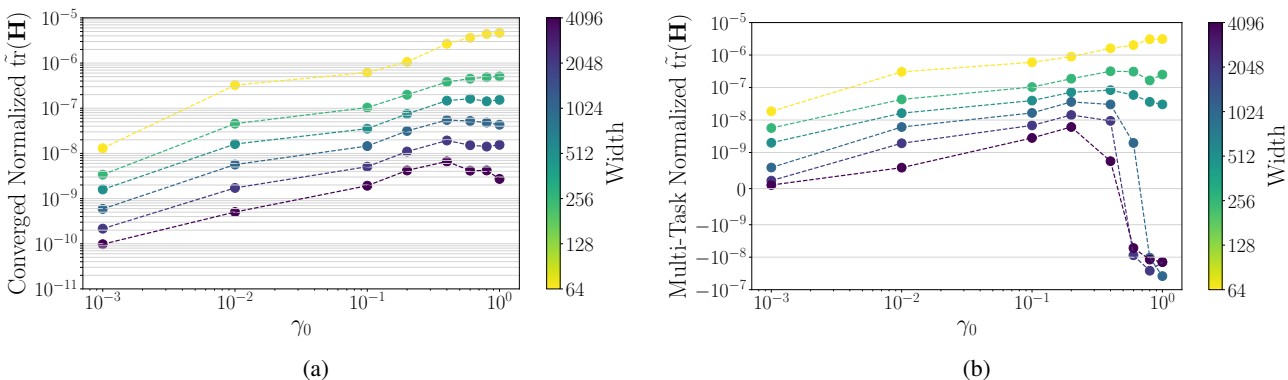

Figure 15. Average effective normalized trace in the $\mu$P for varying $\gamma_0$ and width. (a) Converged; (b) multi-task.

Lastly, we want to stress the notable consistency of the dependency between $\gamma_0$ and the landscape properties across widths, meaning that the landscape properties vary similarly with $\gamma_0$ for all widths. Since this super-consistency is at the root of the LR transfer properties of $\mu$P (Noci et al., 2024), we hypothesize that this might also be a reason for the transfer of the optimal $\gamma_0^\star$ across widths and depths observed above. Nevertheless, we leave a thorough investigation of the transfer properties of $\gamma_0$ for future work.

### B.5.2. FORGETTING CORRELATES WITH THE LANDSCAPE SHARPNESS

Mirzadeh et al. (2020) has first hypothesized the crucial role of the curvature of the loss landscape for the CF problem. Relying on a simple second-order Taylor expansion of the loss function at the minimum of the optimization, they derived a bound on the CF proportional to the curvature of the loss landscape. Although this bound relies on assumptions on the well-behavedness of the loss function, it was validated empirically in a variety of settings, showing a clear correlation between the curvature of the loss landscape and CF. Furthermore, Mirzadeh et al. (2021) have further analyzed this correlation in the context of linear mode connectivity, finding a causal relationship between the curvature of the loss landscape and the CF within a region where the approximation holds. However, their bound is vacuous if there is arbitrary displacement of the parameters, namely when the second-order local approximation is not valid. This is indeed the case for the $\mu$P, and thus it is a priori unclear if their hypothesis regarding the curvature of the landscape holds for the $\mu$P. We empirically investigate how the landscape curvature correlates with the CF in the $\mu$P and we find a clear correlation between landscape curvature and forgetting (Fig. 16). We stress that it is noteworthy that we find this correlation even in the case of varying scales, and training regimes, thus without any assumptions on the locality of the optimization or the linear connectivity of the landscape.

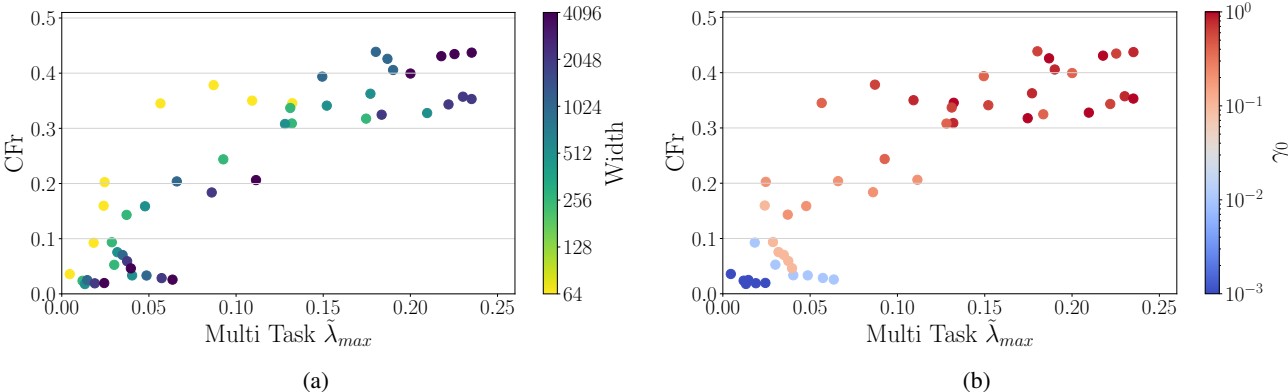

(a)    (b)

*Figure 16.* Multi Task sharpness and its correlation with CFr for $\mu$Ps.

### B.6. Width Dependency on Split-TinyImagenet

In this section, we report the width-dependency of the Split-TinyImagenet results presented in Fig. 6. In particular, in Fig. 17 we observe the width dependency for the 5/2 (5 tasks of 2 classes each), finding similar results to the ones on Split-CIFAR10: width does not mitigate forgetting, and the optimal $\gamma_0^\star$ for the AE is at an intermediate level which transfers across widths. In Fig. 18, instead, we see the effect of width on the 5/40 case where the *pretraining effect* occurs (Sec. 6.2). In particular, all widths exhibit the non-monotonic relationship of CFr with $\gamma_0$ (characteristic of the *pretraining effect*), and an optimal $\gamma_0^\star = 1$ for the AE. Interestingly, we also find that the width is beneficial not only in terms of Average Error but also in terms of forgetting. This recovers a property of pretraining already observed by Ramasesh et al. (2022).

### B.7. CNN Model

Here, were report additional experiments executed without skip connections of the ResNet model, namely including a simpler Convolutional Neural Network (CNN) Model. Comparing these results (Fig. 19) with those on the ResNet model (Fig. 1) we observe that the results are on-par.

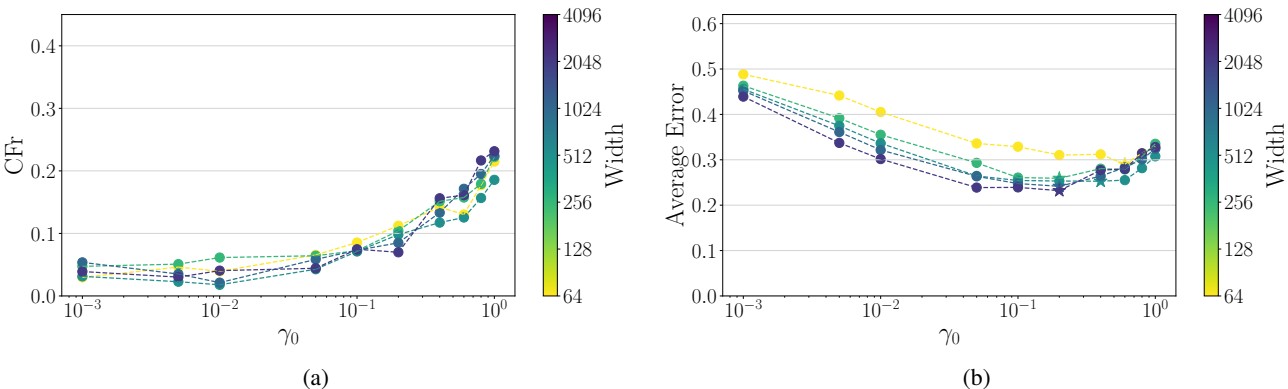

*Figure 17.* The role of width and $\gamma_0$ on Split-TinyImagenet with 5 tasks of 2 classes each. (a) CFr; (b) average error.

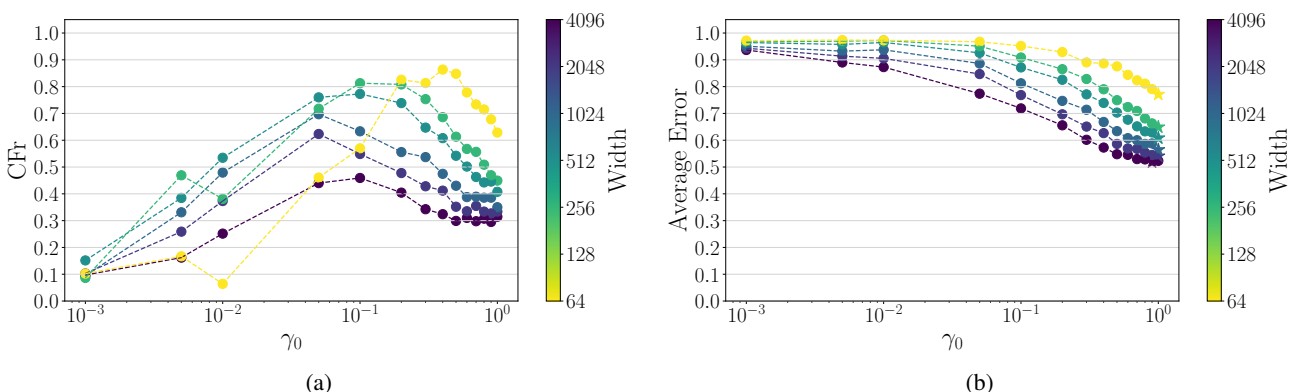

*Figure 18.* The role of width and $\gamma_0$ on Split-TinyImagenet with 5 tasks of 40 classes each. (a) CFr; (b) average error.

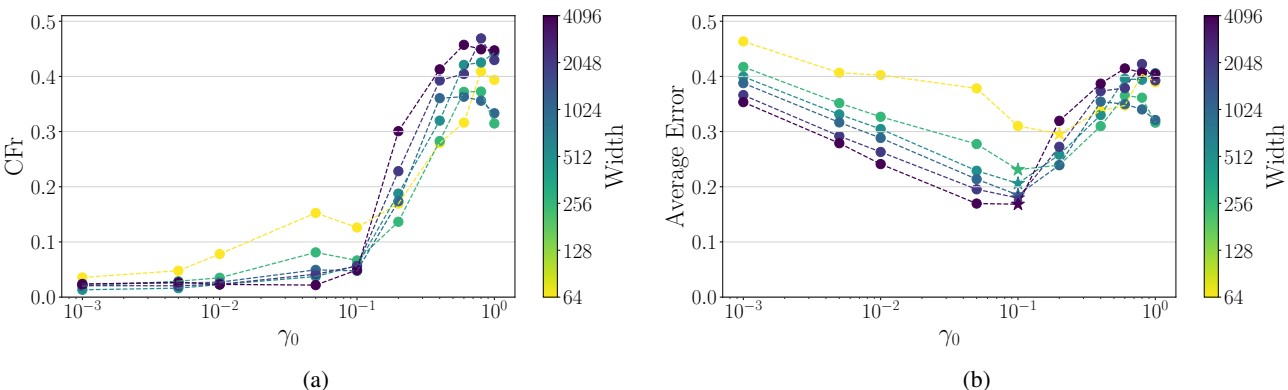

*Figure 19.* Results on CNN model (i.e. architecture as in App. A.2 but without skip connections): (a) CFr, (b) AE.

## C. Network's Function Evolution in Non-Stationary Learning Scenarios

### C.1. Non-Stationary Learning Under $\mu$P: Fields and Dynamics

In order to simplify the notation and the theoretical derivations, consider two different tasks $\mathcal{T}_1$, $\mathcal{T}_2$ represented by the datasets $(X_1, Y_1), (X_2, Y_2)$, each with respectively $P_1, P_2$ number of samples of dimension $D$. In this scenario, we train sequentially $T_1$ first and $T_2$ after. While training the latter, we have no access to the previous task. This can be easily scaled up in terms of the number of tasks, training multiple tasks sequentially, and having access only to the one we are training on. As discussed in the main paper, consider the following architecture, with weights represented by $\boldsymbol{\theta} = \text{Vec}\{\boldsymbol{W}^0, \ldots, \boldsymbol{w}^L\}$, and forward pass defined as:

$$f_\mu = \frac{1}{\gamma} h_\mu^{L+1}, \quad h_\mu^{L+1} = \frac{1}{\sqrt{N}} \boldsymbol{w}_L \cdot \phi(\boldsymbol{h}_\mu^L)$$

$$\boldsymbol{h}_\mu^{\ell+1} = \frac{1}{\sqrt{N}} \boldsymbol{W}^\ell \phi(\boldsymbol{h}_\mu^\ell), \quad \boldsymbol{h}_\mu^1 = \frac{1}{\sqrt{D}} \boldsymbol{W}^0 \boldsymbol{x}_\mu \tag{28}$$

The loss is the Mean Squared Error (MSE) $\mathcal{L} = \sum_{\mu_i} \ell_{\mu_i} = \sum_{\mu_i} \frac{1}{2}(y_{\mu_i} - f_{\mu_i})^2$. Key quantities worth tracking are the model performances on tasks $\mathcal{T}_1$ when we train the task $\mathcal{T}_2$. This involves obtaining the dynamics of the network outputs $f_{\mu_1}$, that are derived assuming gradient flow dynamics, i.e. $\frac{d\boldsymbol{\theta}}{dt} = -\eta \frac{\partial \mathcal{L}}{\partial \boldsymbol{\theta}}$. We write the evolution of the network's output on task $\mathcal{T}_1$, which can be considered as an inference step on that task after each step of the training process on the current training task $\mathcal{T}_2$, as following:

$$\begin{aligned}
\frac{\partial f_{\mu_1}}{\partial t} &= \frac{1}{\gamma} \frac{\partial h_{\mu_1}^{L+1}}{\partial t} \\
&= \frac{1}{\gamma} \frac{\partial h_{\mu_1}^{L+1}}{\partial \boldsymbol{\theta}} \frac{\partial \boldsymbol{\theta}}{\partial t}^\top \\
&= -\frac{\eta}{\gamma} \sum_{\alpha_2} \frac{\partial h_{\mu_1}^{L+1}}{d\boldsymbol{\theta}} \frac{\partial \mathcal{L}}{\partial \boldsymbol{\theta}}^\top \\
&= -\frac{\eta}{\gamma} \sum_{\alpha_2} \frac{\partial h_{\mu_1}^{L+1}}{d\boldsymbol{\theta}} \frac{\partial f_{\alpha_2}}{\partial \boldsymbol{\theta}}^\top \frac{\partial \mathcal{L}}{\partial f_{\alpha_2}} \\
&= \frac{\eta}{\gamma^2} \sum_{\alpha_2} \frac{\partial h_{\mu_1}^{L+1}}{d\boldsymbol{\theta}} \frac{\partial h_{\alpha_2}^{L+1}}{\partial \boldsymbol{\theta}}^\top \Delta_{\alpha_2}
\end{aligned} \tag{29}$$

where we made use of the gradient flow dynamics on task $\mathcal{T}_2$ and then made explicit the dependence of the Loss on the network output. In this way, calling $\Delta_{\alpha_2} = -\frac{\partial \mathcal{L}}{\partial f_{\alpha_2}} = (y_{\alpha_2} - f_{\alpha_2})$, equation above can be written as:

$$\frac{\partial f_{\mu_1}}{\partial t} = \frac{\eta}{\gamma^2} \sum_{\alpha_2} K_{\mu_1 \alpha_2} \Delta_{\alpha_2}, \quad K_{\mu_1 \alpha_2} = \frac{dh_{\mu_1}^{L+1}}{d\boldsymbol{\theta}} \frac{dh_{\alpha_2}^{L+1}}{d\boldsymbol{\theta}}^\top = \frac{dh_{\mu_1}^{L+1}}{d\boldsymbol{\theta}} \cdot \frac{dh_{\alpha_2}^{L+1}}{d\boldsymbol{\theta}} \tag{30}$$

with $K_{\mu_1 \alpha_2}$ the Neural Tangent Kernel across tasks $\mathcal{T}_1$ and $\mathcal{T}_2$. Since at initialization we have $K_{\mu_1 \alpha_2}$ is of order $\mathcal{O}(1)$, we choose the setting $\eta = \mathcal{O}(\gamma^2)$ to have order $\mathcal{O}(1)$ evolution.
We express all quantities involved in terms of features $\boldsymbol{h}_{\mu_1}, \boldsymbol{h}_{\mu_2}$ and gradients $\boldsymbol{g}_{\mu_1}, \boldsymbol{g}_{\mu_2}$ defined as:

$$\boldsymbol{g}_{\mu_i}^\ell = \sqrt{N} \frac{\partial h_{\mu_i}^{L+1}}{\partial \boldsymbol{h}_{\mu_i}^\ell} = \dot{\phi}\left(\boldsymbol{h}_{\mu_i}^\ell\right) \odot \boldsymbol{z}_{\mu_i}^\ell, \quad \boldsymbol{z}_{\mu_i}^l = \frac{1}{\sqrt{N}} \boldsymbol{W}^{\ell\top} \boldsymbol{g}_{\mu_i}^{\ell+1}$$

$$\boldsymbol{g}_\mu^L = \dot{\phi}\left(\boldsymbol{h}_\mu^L\right) \odot \boldsymbol{w}^L \tag{31}$$

where $\boldsymbol{z}_{\mu_i}$ are the *pre-gradients* of task $\mathcal{T}_i$. This allows us to explicit the gradient updates to the parameters in the backward pass as a function of the fields $\boldsymbol{h}_{\mu_i}, \boldsymbol{g}_{\mu_i}$:

$$\frac{\partial h_{\mu_i}^{L+1}}{\partial \boldsymbol{W}^\ell} = \sum_{n=1}^N \frac{\partial h_{\mu_i}^{L+1}}{\partial h_{\mu_i,n}^{\ell+1}} \frac{\partial h_{\mu_i,n}^{\ell+1}}{\partial \boldsymbol{W}^\ell} = \frac{1}{N} \boldsymbol{g}_{\mu_i}^{\ell+1} \phi\left(\boldsymbol{h}_{\mu_i}^\ell\right)^\top \tag{32}$$

enabling us to write the NTK across tasks as:

$$
\begin{aligned}
K_{\mu_1\alpha_2} &= \frac{\partial h_{\mu_1}^{L+1}}{\partial \boldsymbol{\theta}} \cdot \frac{\partial h_{\alpha_2}^{L+1}}{\partial \boldsymbol{\theta}} \\
&= \sum_{\ell=0}^{L} \frac{\partial h_{\mu_1}^{L+1}}{\partial \boldsymbol{W}^\ell} \cdot \frac{\partial h_{\alpha_2}^{L+1}}{\partial \boldsymbol{W}^\ell} \\
&= \frac{1}{N}\phi\left(\boldsymbol{h}_{\mu_1}^L\right)\cdot\phi\left(\boldsymbol{h}_{\alpha_2}^L\right) + \sum_{\ell=1}^{L-1}\left(\frac{\boldsymbol{g}_{\mu_1}^{\ell+1}\cdot\boldsymbol{g}_{\alpha_2}^{\ell+1}}{N}\right)\left(\frac{\phi\left(\boldsymbol{h}_{\mu_1}^\ell\right)\cdot\phi\left(\boldsymbol{h}_{\alpha_2}^\ell\right)}{N}\right) + \frac{\boldsymbol{g}_{\mu_1}^1\cdot\boldsymbol{g}_{\alpha_2}^1}{N}K_{\mu_1\alpha_2}^x
\end{aligned}
\tag{33}
$$

introducing the forward and backward across tasks kernels $\Phi_{\mu_1\alpha_2}, G_{\mu_1\alpha_2}$, written as the inner product along the hidden dimension:

$$
\begin{aligned}
\Phi_{\mu_1\alpha_2}^\ell &= \frac{1}{N}\phi\left(\boldsymbol{h}_{\mu_1}^\ell\right)\phi\left(\boldsymbol{h}_{\alpha_2}^\ell\right)^\top = \frac{1}{N}\phi\left(\boldsymbol{h}_{\mu_1}^\ell\right)\cdot\phi\left(\boldsymbol{h}_{\alpha_2}^\ell\right) \\
G_{\mu_1\alpha_2}^\ell &= \frac{1}{N}\boldsymbol{g}_{\mu_1}^\ell \boldsymbol{g}_{\mu_2}^\ell{}^\top = \frac{1}{N}\boldsymbol{g}_{\mu_1}^\ell \cdot \boldsymbol{g}_{\mu_2}^\ell
\end{aligned}
\tag{34}
$$

together with the input covariance matrix across tasks $K_{\mu_1\alpha_2}^x = \frac{1}{\sqrt{D}}\boldsymbol{x}_{\mu_1}\boldsymbol{x}_{\alpha_2}^\top$. We can express the weights dynamics, namely $\boldsymbol{W}^\ell(t)\,\forall\ell\in[0,L], \forall t > 0$, in terms of the aforementioned forward and backward fields:

$$
\begin{aligned}
\frac{\partial \boldsymbol{W}^\ell(t)}{\partial t} &= -\sum_{\mu_i}\eta\frac{\partial\mathcal{L}}{\partial\boldsymbol{W}^\ell} \\
&= \gamma\sum_{\mu_i}\Delta_{\mu_i}\frac{\partial h_{\mu_i}^{L+1}}{\partial\boldsymbol{W}^\ell} \\
\boldsymbol{W}^\ell(t) &= \boldsymbol{W}^\ell(0) + \frac{\gamma}{N}\int_0^t ds\sum_{\mu_i}\Delta_{\mu_i}(s)\boldsymbol{g}_{\mu_i}^{\ell+1}(s)\phi\left(\boldsymbol{h}_{\mu_i}^\ell(s)\right)^\top
\end{aligned}
\tag{35}
$$

where $\mathcal{T}_i$ is the task we are training on. As a first step, we derive the evolutions of the forward and backward fields of the training, making use of the recursive relation for $\boldsymbol{h}_i$ and $\boldsymbol{z}_i$ respectively in equations 28 and 31:

$$
\begin{aligned}
\boldsymbol{h}_{\mu_i}^{\ell+1}(t) &= \boldsymbol{\chi}_{\mu_i}^{\ell+1}(t) + \frac{\gamma}{\sqrt{N}}\int_0^t ds\sum_{\nu_i}\Delta_{\nu_i}\boldsymbol{g}_{\nu_i}^{\ell+1}(s)\Phi_{\nu_i\mu_i}^\ell(s,t) \\
\boldsymbol{z}_{\mu_i}^\ell(t) &= \boldsymbol{\xi}_{\mu_i}^\ell(t) + \frac{\gamma}{\sqrt{N}}\int_0^t ds\sum_{\nu_i}\Delta_{\nu_i}(s)\phi\left(\boldsymbol{h}_{\nu_i}^\ell(s)\right)G_{\nu_i\mu_i}^{\ell+1}(s,t)
\end{aligned}
\tag{36}
$$

where we have introduced the stochastic fields $\boldsymbol{\chi}_{\mu_i}^{\ell+1}(t), \boldsymbol{\xi}_{\mu_i}^\ell(t)$, arising from the randomness of initialisation conditions:

$$
\begin{aligned}
\boldsymbol{\chi}_{\mu_i}^{\ell+1}(t) &= \frac{1}{\sqrt{N}}\boldsymbol{W}^\ell(0)\phi(\boldsymbol{h}^\ell(t)) \\
\boldsymbol{\xi}_{\mu_i}^\ell(t) &= \frac{1}{\sqrt{N}}\boldsymbol{W}^\ell(0)\boldsymbol{g}^{\ell+1}(t)
\end{aligned}
\tag{37}
$$

Let us now consider a non-stationary learning setting, with two tasks $\mathcal{T}_1$ and $\mathcal{T}_2$, trained sequentially on a single hidden layer network (L=1) in this order. Once we have trained the first tasks, we are interested in tracking the evolution of features and gradients, in order to compute the NTK across the two tasks. To obtain the fields evolution equation, we again utilize the recursive relations definitions of $\boldsymbol{h}_1$ and $\boldsymbol{g}_1$, where this time the evolution of the weights is due to the training in tasks $\mathcal{T}_2$:

$$
\begin{aligned}
\boldsymbol{h}_{\mu_1}^1(t) &= \frac{1}{\sqrt{N}}\boldsymbol{W}^0(t)\boldsymbol{x}_{\mu_1} = \boldsymbol{\chi}_{\mu_1}^1 + \gamma_0\int_{t_1}^t\sum_{\alpha_2}\Delta_{\alpha_2}(s)\boldsymbol{g}_{\alpha_2}^1(s)K_{\mu_1\alpha_2}^x ds \\
\boldsymbol{z}^1(t) &= \boldsymbol{w}^1(t) = \boldsymbol{\xi}^1 + \gamma_0\int_{t_1}^t\sum_{\alpha_2}\Delta_{\alpha_2}(s)\phi(\boldsymbol{h}_{\alpha_2}^1(s))ds
\end{aligned}
\tag{38}
$$

with $\chi^1_{\mu_1}$ and $\xi^1$ stochastic fields. In the general case of T tasks, these equations can be adapted, making use of the Heaviside function $U(t)$, to split the Gradient Descent evolution on the total set of tasks:

$$
\boldsymbol{h}^1_{\mu_1}(t) = \frac{1}{\sqrt{N}} \boldsymbol{W}^0(t) \boldsymbol{x}_{\mu_1} = \boldsymbol{\chi}^1_{\mu_1} + \gamma_0 \int_0^t \sum_{i=1}^T \tilde{U}_i(s) \sum_{\alpha_2} \Delta_{\alpha_2}(s) \boldsymbol{g}^1_{\alpha_2}(s) K^x_{\mu_1 \alpha_2} ds
$$
$$
\boldsymbol{z}^1(t) = \boldsymbol{w}^1(t) = \boldsymbol{\xi}^1 + \gamma_0 \int_0^t \sum_{i=1}^T \tilde{U}_i(s) \sum_{\alpha_2} \Delta_{\alpha_2}(s) \phi(\boldsymbol{h}^1_{\alpha_2}(s)) ds \tag{39}
$$

with $\tilde{U}_i(t) = U(t - t_{i-1}) U(t_i - t)$, splitting the integral in multiple ones with the correct time extrema, inside which task $T_i$ is under training. Once we have the fields $\{\boldsymbol{h}^1, \boldsymbol{g}^1\}$ we can construct the primitive kernels $\Phi^1_{\mu_1 \alpha_2}, G^1_{\mu_1 \alpha_2}$, and then the NTK across tasks. The latter is then used to compute the residual evolution as:

$$
\frac{\partial}{\partial t} \boldsymbol{\Delta}_{\mu_1}(t) = -[\boldsymbol{\Phi}^1_{\mu_1 \alpha_2}(t) + \boldsymbol{G}^1_{\mu_1 \alpha_2}(t) \boldsymbol{K}^x_{\mu_1 \alpha_2}] \boldsymbol{\Delta}_{\alpha_2}(t) \tag{40}
$$

we thus can predict the evolution of residuals of different tasks with respect to the one we are training. Moreover, we have the equations governing the internal representation evolution during other tasks.

## C.2. DMFT for One Hidden Layer NN in CL Scenario (Proof of Proposition 4.1)

The derivation is carried out using the Martin-Siggia-Rose-Janssen-De Dominicis formalism (Martin et al., 1973), which allows us to prove that at $N \to \infty$ the kernels concentrate, making the evolution of residuals deterministic. Since kernels depend on the stochastic fields $\chi_{\mu_1}, \chi_{\alpha_2}$, we are interested in characterizing the distributions of them at infinite width under the non-stationary scenario. Specifically, we aim to obtain the expectation of the inner product $\langle \chi_{\mu_1} \cdot \chi_{\alpha_2} \rangle$ (in this section we write $\langle \cdot \rangle$ as the expectation $\mathbb{E}[\cdot]$), since it represents the initial condition for the forward kernel $\Phi^1_{\mu_1 \alpha_2}$. Furthermore, we want to prove that in the infinite limit the evolutions of residuals of tasks $\mathcal{T}_1$ while training task $\mathcal{T}_2$ are determinist, since kernels $\Phi^1_{\mu_1 \alpha_2}, G^1_{\mu_1 \alpha_2}$ concentrate in the $N \to \infty$ limit. To do that, we look at the Moment Generating Function (MGF) of fields $\{\boldsymbol{\chi}_{\mu_1}\}_{\mu_1 \in [P_1]}, \{\boldsymbol{\chi}_{\mu_2}\}_{\mu_2 \in [P_2]}$ and $\boldsymbol{\xi}$, defined as follows:

$$
Z\left[\{\boldsymbol{j}_{\mu_1}\}_{\mu_1 \in [P_1]}, \{\boldsymbol{j}_{\mu_1}\}_{\mu_1 \in [P_1]}, \boldsymbol{v}\right] = \left\langle \exp\left( \sum_{\mu_1} \boldsymbol{j}_{\mu_1} \cdot \boldsymbol{\chi}_{\mu_1} + \sum_{\mu_2} \boldsymbol{j}_{\mu_2} \cdot \boldsymbol{\chi}_{\mu_2} + \boldsymbol{v} \cdot \boldsymbol{\xi} \right) \right\rangle_{\theta_0} \tag{41}
$$

As a next step, we enforce the definition of the random fields through Dirac deltas, making explicit the source of randomness (weights at initialization), introducing new integration variables, and thus enabling us to compute the average.

$$
1 = \int \int \frac{d\boldsymbol{\chi}_{\mu_1} d\hat{\boldsymbol{\chi}}_{\mu_1}}{(2\pi)^N} \exp\left( i\hat{\boldsymbol{\chi}}_{\mu_1} \cdot \left( \boldsymbol{\chi}_{\mu_1} - \frac{1}{\sqrt{D}} \boldsymbol{W}(0) \boldsymbol{x}_{\mu_1} \right) \right)
$$
$$
1 = \int \int \frac{d\boldsymbol{\chi}_{\mu_2} d\hat{\boldsymbol{\chi}}_{\mu_2}}{(2\pi)^N} \exp\left( i\hat{\boldsymbol{\chi}}_{\mu_2} \cdot \left( \boldsymbol{\chi}_{\mu_2} - \frac{1}{\sqrt{D}} \boldsymbol{W}(0) \boldsymbol{x}_{\mu_2} \right) \right) \tag{42}
$$
$$
1 = \int \int \frac{d\boldsymbol{\xi} d\hat{\boldsymbol{\xi}}}{(2\pi)^N} \exp\left( i\hat{\boldsymbol{\xi}} \cdot (\boldsymbol{\xi} - \boldsymbol{w}(0)) \right)
$$

We then insert these definitions in the MGF:

$$
Z\left[\{\boldsymbol{j}_{\mu_1}\}_{\mu_1 \in [P_1]}, \{\boldsymbol{j}_{\mu_1}\}_{\mu_1 \in [P_1]}, \boldsymbol{v}\right] = \langle \int \prod_{\mu_1 \mu_2} \frac{d\boldsymbol{\chi}_{\mu_1} \hat{\boldsymbol{\chi}}_{\mu_1} \boldsymbol{\chi}_{\mu_2} \hat{\boldsymbol{\chi}}_{\mu_2} d\boldsymbol{\xi} d\hat{\boldsymbol{\xi}}}{(2\pi)^N (2\pi)^N (2\pi)^N} \exp(\sum_{\mu_1} \boldsymbol{j}_{\mu_1} \cdot \boldsymbol{\chi}_{\mu_1} + \sum_{\mu_2} \boldsymbol{j}_{\mu_2} \cdot \boldsymbol{\chi}_{\mu_2} + \boldsymbol{v} \cdot \boldsymbol{\xi}
$$
$$
+ i\hat{\boldsymbol{\chi}}_{\mu_1} \cdot (\boldsymbol{\chi}_{\mu_1} - \frac{1}{\sqrt{D}} \boldsymbol{W}(0) \boldsymbol{x}_{\mu_1}) + i\hat{\boldsymbol{\chi}}_{\mu_2} \cdot (\boldsymbol{\chi}_{\mu_2} - \frac{1}{\sqrt{D}} \boldsymbol{W}(0) \boldsymbol{x}_{\mu_2})) \tag{43}
$$
$$
+ i\hat{\boldsymbol{\xi}} \cdot (\boldsymbol{\xi} - \boldsymbol{w}(0)) \rangle_{\theta_0}
$$

Now we can compute the average over the random noise, namely $\theta_0$, since we know that $W_{ij}, w_{ij} \sim \mathcal{N}(0, 1)$. Having the Gaussian terms plus a linear term deriving from the explicit definition of fields, the integral to be computed is essentially a

Fourier transform of the original Gaussian, which leads to another Gaussian (inverse of the Hubbard-Stratonovich transform):

$$Z\left[\{\boldsymbol{j}_{\mu_1}\}_{\mu_1\in[P_1]}, \{\boldsymbol{j}_{\mu_1}\}_{\mu_1\in[P_1]}, \boldsymbol{v}\right] = \int \prod_{\mu_1\mu_2} \frac{d\boldsymbol{\chi}_{\boldsymbol{\mu_1}}d\hat{\boldsymbol{\chi}}_{\boldsymbol{\mu_1}}d\boldsymbol{\chi}_{\boldsymbol{\mu_2}}d\hat{\boldsymbol{\chi}}_{\boldsymbol{\mu_2}}d\boldsymbol{\xi}d\hat{\boldsymbol{\xi}}}{(2\pi)^N(2\pi)^N(2\pi)^N} \exp\Big(\sum_{\mu_1}(\boldsymbol{j}_{\mu_1}+i\hat{\boldsymbol{\chi}}_{\boldsymbol{\mu_1}})\cdot\boldsymbol{\chi}_{\boldsymbol{\mu_1}}+$$

$$\sum_{\mu_2}(\boldsymbol{j}_{\mu_2}+i\hat{\boldsymbol{\chi}}_{\boldsymbol{\mu_2}})\cdot\boldsymbol{\chi}_{\boldsymbol{\mu_2}} - \frac{1}{2}\big(\sum_{\mu_1,\alpha_1}\hat{\boldsymbol{\chi}}_{\boldsymbol{\mu_1}}\cdot\hat{\boldsymbol{\chi}}_{\boldsymbol{\alpha_1}}K^x_{\mu_1\alpha_1} + \sum_{\mu_1,\alpha_2}\hat{\boldsymbol{\chi}}_{\boldsymbol{\mu_1}}\cdot\hat{\boldsymbol{\chi}}_{\boldsymbol{\alpha_2}}K^x_{\mu_1\alpha_2}$$

$$+ \sum_{\mu_2,\alpha_1}\hat{\boldsymbol{\chi}}_{\boldsymbol{\mu_2}}\cdot\hat{\boldsymbol{\chi}}_{\boldsymbol{\alpha_1}}K^x_{\mu_2\alpha_1} + \sum_{\mu_2,\alpha_2}\hat{\boldsymbol{\chi}}_{\boldsymbol{\mu_2}}\cdot\hat{\boldsymbol{\chi}}_{\boldsymbol{\alpha_2}}K^x_{\mu_2\alpha_2}\big) - \frac{1}{2}|\hat{\boldsymbol{\xi}}|^2 + (\boldsymbol{v}+i\hat{\boldsymbol{\xi}})\cdot\boldsymbol{\xi}\Big) \tag{44}$$

Now we introduce the definitions of the order parameters, namely kernels

$$\Phi_{\mu_1\mu_2}(t,s) = \frac{1}{N}\phi(\boldsymbol{h}_{\mu_1}(t))\cdot\phi(\boldsymbol{h}_{\mu_2}(s)), \quad G_{\mu_1\mu_2}(t,s) = \frac{1}{N}\boldsymbol{g}_{\mu_1}(t)\cdot\boldsymbol{g}_{\mu_2}(s)$$

through Dirac delta definitions, as we did for the random initial fields:

$$1 = \int \frac{dG_{\mu_1\mu_2}(t,s)d\hat{G}_{\mu_1\mu_2}(t,s)}{2\pi i N^{-1}} \exp\Big(\hat{G}_{\mu_1\mu_2}(t,s)\left(NG_{\mu_1\mu_2}(t,s) - \boldsymbol{g}_{\mu_1}(t)\cdot\boldsymbol{g}_{\mu_2}(s)\right)\Big)$$

$$1 = \int \frac{d\Phi_{\mu_1\mu_2}(t,s)d\hat{\Phi}_{\mu_1\mu_2}(t,s)}{2\pi i N^{-1}} \exp\Big(\hat{\Phi}_{\mu_1\mu_2}(t,s)\left(N\Phi_{\mu_1\mu_2}(t,s) - \phi(\boldsymbol{h}_{\mu_1}(t))\cdot\phi(\boldsymbol{h}_{\mu_2}(s))\right)\Big) \tag{45}$$

that put together in the above formula for Z lead to:

$$Z \propto \int \prod_{\mu_1\mu_2 ts} d\Phi_{\mu_1\mu_2}(t,s)d\hat{\Phi}_{\mu_1\mu_2}(t,s)dG_{\mu_1\mu_2}(t,s)d\hat{G}_{\mu_1\mu_2}(t,s)\exp(NS[\Phi,\hat{\Phi},G,\hat{G}]) \tag{46}$$

where $S$ is the action and is given by:

$$S[\Phi_{\mu_1\mu_2},\hat{\Phi}_{\mu_1\mu_2},G_{\mu_1\mu_2},\hat{G}_{\mu_1\mu_2}] = \sum_{\mu_1\mu_2}\int dsdt[\Phi_{\mu_1\mu_2}(t,s)\hat{\Phi}_{\mu_1\mu_2}(t,s) + G_{\mu_1\mu_2}(t,s)\hat{G}_{\mu_1\mu_2}(t,s)]$$

$$+ \frac{1}{N}\sum_{i=1}^{N}\ln(\mathcal{Z}[j_{1_i},j_{2_i},v_i])$$

$$\mathcal{Z}[j,v] = \int \prod_{\mu_1\mu_2} \frac{d\chi_{\mu_1}d\hat{\chi}_{\mu_1}d\chi_{\mu_2}d\hat{\chi}_{\mu_2}d\xi d\hat{\xi}}{(2\pi)^N(2\pi)^N(2\pi)^N} \exp\Big(\sum_{\mu_1}(j_{\mu_1}+i\hat{\chi}_{\mu_1})\cdot\chi_{\mu_1} + \sum_{\mu_2}(j_{\mu_2}+i\hat{\chi}_{\mu_2})\cdot\chi_{\mu_2}$$

$$-\frac{1}{2}\big(\sum_{\mu_1,\alpha_1}\hat{\chi}_{\mu_1}\cdot\hat{\chi}_{\alpha_1}K^x_{\mu_1\alpha_1} + \sum_{\mu_1,\alpha_2}\hat{\chi}_{\mu_1}\cdot\hat{\chi}_{\alpha_2}K^x_{\mu_1\alpha_2} + \sum_{\mu_2,\alpha_1}\hat{\chi}_{\mu_2}\cdot\hat{\chi}_{\alpha_1}K^x_{\mu_2\alpha_1} + \sum_{\mu_2,\alpha_2}\hat{\chi}_{\mu_2}\cdot\hat{\chi}_{\alpha_2}K^x_{\mu_2\alpha_2}\big)$$

$$-\frac{1}{2}\hat{\xi}^2 + (v+i\hat{\xi})\cdot\xi - \hat{\Phi}_{\mu_1\mu_2}(t,s)\phi(h_{\mu_1}(t))\cdot\phi(h_{\mu_2}(s)) - \hat{G}_{\mu_1\mu_2}(t,s)g_{\mu_1}(t)\cdot g_{\mu_2}(s)\Big) \tag{47}$$

We then take the saddle point solution for the integral defining Z, which essentially means finding the values of the order parameters maximizing S. This leads to the following equations:

$$\frac{\delta S}{\delta\Phi_{\mu_1\mu_2}(t,s)} = \hat{\Phi}_{\mu_1\alpha}(t,s) = 0, \frac{\delta S}{\delta\hat{\Phi}_{\mu_1\mu_2}(t,s)} = \Phi_{\mu_1\mu_2}(t,s) - \frac{1}{N}\sum_{i=1}^{N}\langle\phi\left(h_{\mu_1}(t)\right)\phi\left(h_{\mu_2}(s)\right)\rangle_i = 0$$

$$\frac{\delta S}{\delta G_{\mu_1\mu_2}(t,s)} = \hat{G}_{\mu_1\mu_2}(t,s) = 0, \frac{\delta S}{\delta\hat{G}_{\mu_1\mu_2}(t,s)} = G_{\mu_1\mu_2}(t,s) - \frac{1}{N}\sum_{i=1}^{N}\langle g_{\mu_1}(t)g_{\mu_2}(s)\rangle_i = 0 \tag{48}$$

where the average $\langle\rangle_i$ is defined as:

$$\langle O(\chi,\hat{\chi},\xi,\hat{\xi})\rangle_i = \frac{1}{\mathcal{Z}[j_{1_i},j_{2_i},v_i]} \int \prod_{\mu_1\mu_2} d\chi_{\mu_1}d\hat{\chi}_{\mu_1}d\chi_{\mu_2}d\hat{\chi}_{\mu_2}d\xi d\hat{\xi}$$

$$\exp\Big(-\mathcal{H}\Big[\{\chi_{\mu_1},\hat{\chi}_{\mu_1},\chi_{\mu_2},\hat{\chi}_{\mu_2}\},\xi,\hat{\xi},j_{1_i},j_{2_i},v_i\Big]\Big)O(\chi,\hat{\chi},\xi,\hat{\xi})$$

where $\mathcal{H}$ is the logarithm of the $\mathcal{Z}[j_{1_i}, j_{2_i}, v_i]$. Once we plug in the saddle point solutions we can rewrite the remaining terms defining the matrix and vectors:

$$\boldsymbol{A} = \begin{bmatrix} K^x_{\mu_1\alpha_1} & K^x_{\mu_1\alpha_2} \\ K^x_{\mu_2\alpha_1} & K^x_{\mu_2\alpha_2} \end{bmatrix}, \quad \boldsymbol{y} = \begin{bmatrix} \chi_{\mu_1} \\ \chi_{\mu_2} \end{bmatrix}, \quad \boldsymbol{x} = \begin{bmatrix} \hat{\chi}_{\mu_1} \\ \hat{\chi}_{\mu_2} \end{bmatrix}, \quad \boldsymbol{j} = \begin{bmatrix} j_{\mu_1} \\ j_{\mu_2} \end{bmatrix}, \tag{49}$$

so that we can rewrite Z in the following way:

$$\begin{aligned} Z[\boldsymbol{j}, v] &= \int \frac{d\boldsymbol{x}d\boldsymbol{y}d\xi d\hat{\xi}}{(2\pi)^N(2\pi)^N(2\pi)^N} \exp(-\frac{1}{2}\boldsymbol{x}^T\boldsymbol{A}\boldsymbol{x} + \boldsymbol{j}\boldsymbol{y} + i\boldsymbol{x}\boldsymbol{y} - \frac{1}{2}\hat{\xi}^2 + (v + i\hat{\xi})\cdot\xi) \\ &= \int \frac{d\boldsymbol{y}d\xi}{(2\pi)^N(2\pi)^N(2\pi)^N} \exp(-\frac{1}{2}\boldsymbol{y}^T\boldsymbol{C}\boldsymbol{y} + \boldsymbol{j}\boldsymbol{y} - \frac{1}{2}\xi^2 + v\xi) \end{aligned} \tag{50}$$

where we have defined the matrix $C = A^{-1}$. At this point, we keep integrating and obtain:

$$Z[\boldsymbol{j}, v] = \exp(\frac{1}{2}\boldsymbol{j}^T\boldsymbol{A}\boldsymbol{j} + \frac{1}{2}v^2) \tag{51}$$

which is the MGF of the fields $\chi_{\mu_1}, \chi_{\mu_2}, \xi$. If we are interested in the value of $\langle \chi_{\mu_1} \cdot \chi_{\mu_2} \rangle$, we can easily get it from Z by:

$$\begin{aligned} \langle \chi_{\mu_1} \cdot \chi_{\mu_2} \rangle &= \frac{\partial}{\partial \boldsymbol{j}_{\mu_2} \partial \boldsymbol{j}^T_{\mu_1}} Z|_{\boldsymbol{j},v=0} \\ &= \frac{\partial}{\partial \boldsymbol{j}_{\mu_2} \partial \boldsymbol{j}^T_{\mu_1}} \exp(\frac{1}{2}(\boldsymbol{j}^T_{\mu_1} K^x_{\mu_1\mu_1} \boldsymbol{j}_{\mu_1} + \boldsymbol{j}^T_{\mu_1} K^x_{\mu_1\mu_2} \boldsymbol{j}_{\mu_2} + \boldsymbol{j}^T_{\mu_2} K^x_{\mu_2\mu_1} \boldsymbol{j}_{\mu_1} + \boldsymbol{j}^T_{\mu_2} K^x_{\mu_2\mu_2} \boldsymbol{j}_{\mu_2} + v^2))|_{\boldsymbol{j}_{\mu_1}, \boldsymbol{j}_{\mu_2}, v=0} \\ &= K^x_{\mu_1\mu_2} \end{aligned} \tag{52}$$

## D. Comparison of Infinite Width Simulations and Experimental Results

In order to verify the goodness of the theoretical framework developed for the non-stationary learning case, we compared the results from a single hidden layer MLP of width 4096, ReLU activation function, $\mu P$ parameterized, $\gamma_0 = 1$ and learning rate $\eta_0 = 0.25$, with the predictions from the infinite width limit version. The dataset is the same used in the previous experiments, namely the PermutedMNIST restricted to 30 samples, 3 per label. The number of epochs are 500 and the total number of tasks is 4, each generated from a different complete permutation of the images pixels.

The interesting quantities to observe are the ones involving the internal representation of the samples, gradients and the losses evolution, in order to test the accuracy of the theory from both low and high level observables. We show below the distributions of the pre-activations $\{h_i\}_{i \in \{1,2,3,4\}}$, of gradients $\{g_i\}_{i \in \{1,2,3,4\}}$ for all 30 samples:

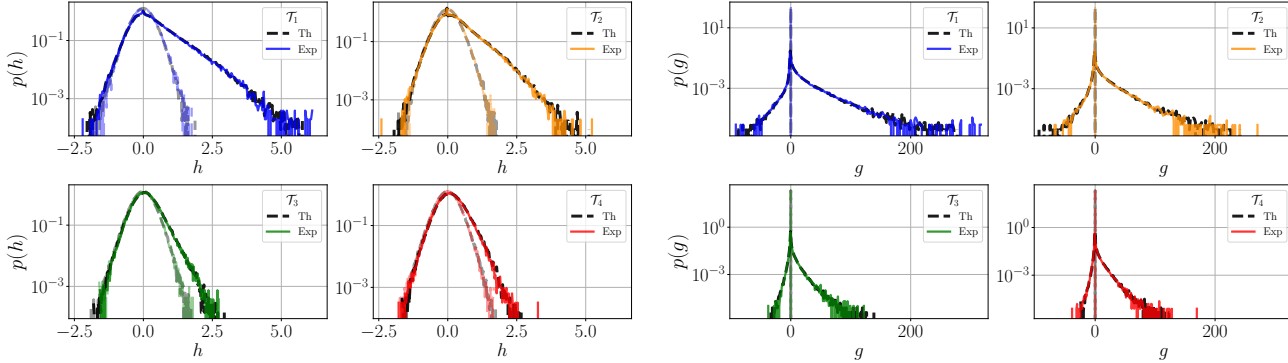

(a) Pre-activations distributions for all 30 samples of the different tasks. The shaded lines are the distributions at $t = 450$, the full lines are the distributions at $t = 550$.

(b) Gradients distributions for all 30 samples of the different tasks. The shaded lines are the distributions at $t = 450$, the full lines are the distributions at $t = 550$.

*Figure 20.*

It is clear in above figure the effect of training on the pre-activations, and consequently gradients, distributions: there is a shift from the initial Gaussian curves towards heavily tailed, non-Gaussian distributions. Having the pre-activations and gradients we can easily obtain the primitive forward and backward kernels $\Phi_{\mathcal{T}_i \mathcal{T}_j}$ and $G_{\mathcal{T}_i \mathcal{T}_j}$. We show below the kernels $\Phi_{\mathcal{T}_1 \mathcal{T}_2}, G_{\mathcal{T}_1 \mathcal{T}_2}, K_{\mathcal{T}_1 \mathcal{T}_2}$ and $\Phi_{\mathcal{T}_3 \mathcal{T}_4}, G_{\mathcal{T}_3 \mathcal{T}_4}, K_{\mathcal{T}_3 \mathcal{T}_4}$ at time steps $t = 450$ and $t = 550$, thus 50 epochs before and after the switch from task $\mathcal{T}_1$ to task $\mathcal{T}_2$:

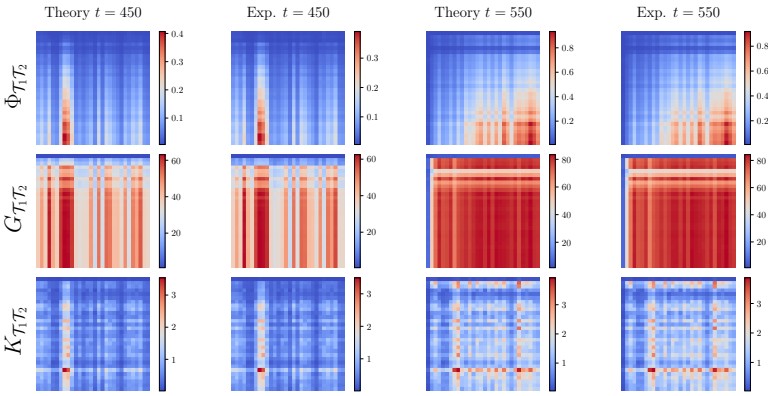

*Figure 21.* All primitive kernels and NTK across tasks $\mathcal{T}_1$ and $\mathcal{T}_2$. The theoretical one are obtained through simulations of the infinite width model, the experimental ones comes from the trained network aforementioned

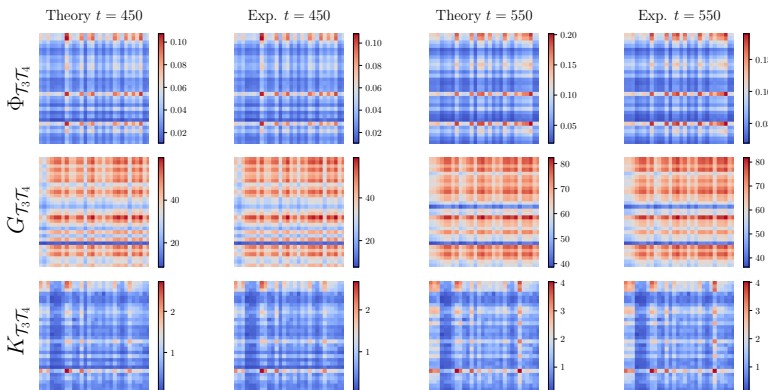

*Figure 22.* All primitive kernels and NTK across tasks $\mathcal{T}_3$ and $\mathcal{T}_4$. The theoretical one are obtained through simulations of the infinite width model, the experimental ones comes from the trained network aforementioned

Last, from equation 30, we can compute the loss evolution of all tasks' losses $\{\mathcal{L}_{\mathcal{T}_i}\}_{i \in \{1,2,3,4\}}$ during all tasks' training:

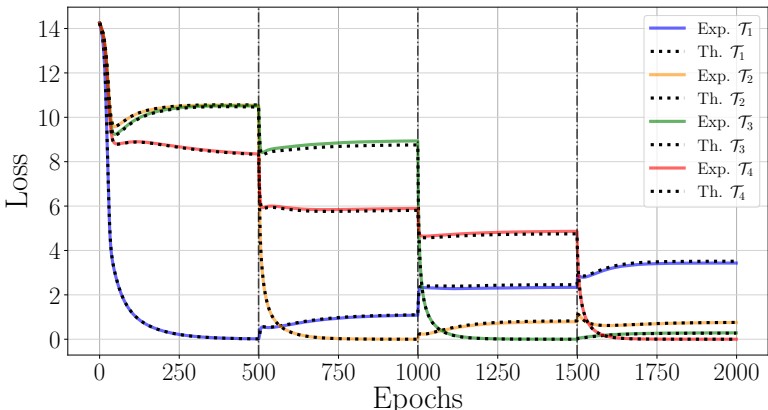

*Figure 23.* Tasks' losses in a four task scenario mentioned above. Vertical dot-dashed lines represent the epochs at which the shift between tasks training happens.

In order to gain a deeper insight into the learning dynamics, a useful measure is the alignment of the last-layer forward Kernel ($\{\Phi^L_{\mathcal{T}_i \mathcal{T}_i}\}_{i \in \{1,2,3,4\}}$) along the task-relevant subspace $YY^\top$, as defined in (Atanasov et al., 2024, App. P) and termed *Kernel-Target Alignment* (KTA). We denote the KTA as $\mathcal{A}(\Phi, YY^T)$. This measure provides a qualitative picture of how and when features develop structures able to cluster samples according to respective labels. As shown in Fig. 24 the KTA increases only during training of the respective task, and the alignment decreases during other tasks' training.

We can see that the infinite width model matches also this measure, serving as a further validation of the closely matching characteristics of the experimental and theoretical setup.

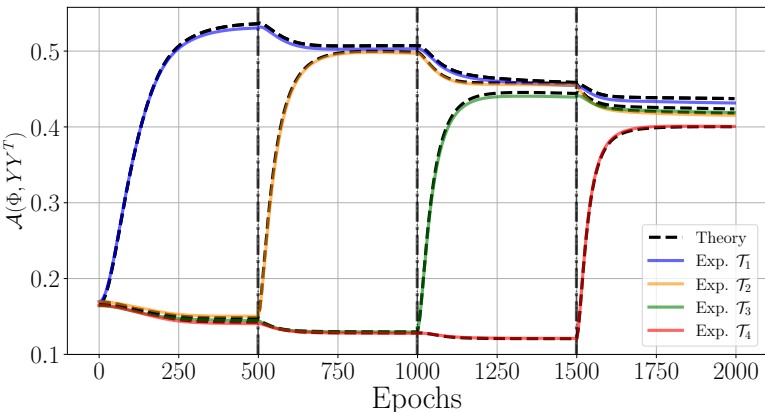

*Figure 24.* Theoretical and experimental tasks' alignment $\mathcal{A}(\Phi, YY^T)$ for all four tasks during all training processes. Vertical dot-dashed lines represent the epochs at which the shift between tasks training happens.

# E. Catastrophic Forgetting Under Perturbation Theory in $\gamma_0$

## E.1. Effect of Feature Learning on CF in the Infinite Width Limit

The dependence on $\gamma_0$ in Equation (12) is implicit and recursively applied over the training steps. To get a better sense of the influence of feature learning on forgetting, we apply perturbation theory, which assumes the following expansion for forgetting:

$$CF(t) = CF^{(0)}(t) + \gamma_0 CF^{(1)}(t) + \gamma_0^2 CF^{(2)}(t) + \mathcal{O}(\gamma_0^3) \tag{53}$$

The aim is to compute the coefficients $CF^{(0)}(t), CF^{(1)}, \ldots$, identifying the respective terms in front of the relative $\gamma_0$ expansion deriving from the other set of variables composing the self consistent system of equations. Notice that here superscripts index the coefficients and not powers (e.g. $CF^{(2)}(t)$ is the second order coefficient of the expansion). By design, the expansion coefficients $\{CF^{(i)}\}$ do not depend on $\gamma_0$. For a two-layer linear network, up to order $\gamma_0^2$, we find the following:

**Proposition E.1.** *Let $f$ be the output of a neural network in Eq. 1 with identity activation function, a single hidden layer, and $\alpha = 0$ (i.e. without skip connections) trained for $T = 2$ tasks $\mathcal{T}_1, \mathcal{T}_2$ for a finite number of steps. Let $t_1$ be the number of training steps on $\mathcal{T}_1$ and $t > t_1$. In the limit $N \to \infty$, the catastrophic forgetting defined in Eq. 11 admits the expansion of Eq. 53. The first three coefficients are:*

$$CF^{(0)}(t) = -\int_{t_1}^t \Delta_{\mathcal{T}_1}^{(0)}(s) K_{\mathcal{T}_1\mathcal{T}_2}^{(0)} \Delta_{\mathcal{T}_2}^{(0)}(s) ds$$

$$CF^{(1)} = 0$$

$$CF^{(2)}(t) = -\int_{t_1}^t \Big( \Delta_{\mathcal{T}_1}^{(0)}(s) K_{\mathcal{T}_1\mathcal{T}_2}^{(2)}(s) \Delta_{\mathcal{T}_2}^{(0)}(s)$$

$$+ \Delta_{\mathcal{T}_1}^{(2)}(s) K_{\mathcal{T}_1\mathcal{T}_2}^{(0)} \Delta_{\mathcal{T}_2}^{(0)}(s) + \Delta_{\mathcal{T}_1}^{(0)}(s) K_{\mathcal{T}_1\mathcal{T}_2}^{(0)} \Delta_{\mathcal{T}_2}^{(2)}(s) \Big) ds,$$

*where $\Delta^{(0)}, K^{(0)}, K^{(2)}, \Delta^{(2)}$ are the perturbative coefficients for the residual $\Delta$ and the NTK $K$. The full set of equations, which includes the expansion of the kernels $\Phi$ and $G$, can be found in Appendix E.*

**Remarks**. The zeroth order term corresponds to the catastrophic forgetting incurred in the NTK/lazy regime, where $\gamma_0 = 0$. This term was analyzed in (Doan et al., 2021; Bennani et al., 2020) for a slightly different definition of forgetting. Here, the zero-order term of the NTK $K_{\mathcal{T}_1\mathcal{T}_2}^0$ (which corresponds to the NTK in the lazy regime) is fixed to initialization, thus we exclude the explicit time dependency. The first order term is zero, as found in Bordelon & Pehlevan (2022) for the features and backward kernels. Compared to them, here we also compute leading order perturbations for $\Delta$, as necessary for our analysis. Also note that this characterization of forgetting is only valid for small values of $\gamma_0$, where the higher order terms $\mathcal{O}(\gamma_0^3)$ can be ignored. Our theory characterizes the effect of *increasing the degree of feature learning* on CF, starting from the lazy training setting – for which CF is already known. Perturbative methods have been applied in other contexts of deep learning theory, particularly in computing finite size corrections to the infinite width limit (Roberts et al., 2022; Hanin & Nica, 2019).

Our analysis offers a general blueprint for calculating the effect of small feature learning perturbations on CF. However, in practice, the expansion coefficients can only be solved in closed form by considering specific learning instances.

### E.1.1. DERIVATION

Following previous works by (Bordelon & Pehlevan, 2022), we inspected the expansion of the fundamental quantities governing the evolution of residuals and the residuals themselves around $\gamma_0 = 0$. This is done using a perturbation theory

approach, thus expanding the fields and the residuals in power series of $\gamma_0$, so that:

$$h_{\mu_1}(t) = \chi_{\mu_1} + \sum_{n=1}^{\infty} \gamma_0^n h_{\mu_1}^{(n)}(t) \tag{54}$$

$$z_{\mu_1}(t) = \xi + \sum_{n=1}^{\infty} \gamma_0^n z_{\mu_1}^{(n)}(t) \tag{55}$$

$$\Delta_{\mu_1}(t) = \sum_{n=0}^{\infty} \gamma_0^n \Delta_{\mu_1}^{(n)} \tag{56}$$

In this study, we will stop the expansion at the second order term, due to practical reasons, but also because the parameter $\gamma_0$ is typically in the range $[0, 1]$ and the *lazy-rich* transition observed in experiments is typically of the order of $0.1$. We begin by considering the update equations for the fields and residuals, and plug on the left side of the equation the expansions in $\gamma_0$:

$$h_{\mu_1}^{(0)} + \gamma_0 h_{\mu_1}^{(1)} + \gamma_0^2 h_{\mu_1}^{(2)} + \ldots = \chi_{\mu_1} + \int_{t_1}^{t} \sum_{\alpha_2} \Delta_{\alpha_2}(s) g_{\alpha_2}^1(s) K_{\mu_1 \alpha_2}^x ds \tag{57}$$

$$z^{(0)} + \gamma_0 z^{(1)} + \gamma_0^2 z^{(2)} + \ldots = \xi + \gamma_0 \int_{t_1}^{t} \sum_{\alpha_2} \Delta_{\alpha_2} \phi(h_{\alpha_2}^1) ds \tag{58}$$

$$\Delta_{\mu_1}^{(0)}(t) + \gamma_0 \Delta_{\mu_1}^{(1)}(t) + \gamma_0^2 \Delta_{\mu_1}^{(2)}(t) + \ldots = Y + \int_{t_1}^{t} K_{\mu_1 \alpha_2}(s) \Delta_{\alpha_2}(s) ds \tag{59}$$

The next step is to identify on the right side of the equations above the corresponding terms in the left-hand side expansions, which is long but trivial. We finally obtain:

$$h_{\mu_1}^{(0)} = \chi_{\mu_1} \tag{60}$$

$$h_{\mu_1}^{(1)} = \int_0^{t_1} \sum_{\alpha_2} \Delta_{\alpha_1}^{(0)} \xi K_{\alpha_1 \mu_1}^x ds + \int_{t_1}^{t} \sum_{\alpha_2} \Delta_{\alpha_2}^{(0)} \xi K_{\alpha_2 \mu_1}^x ds \tag{61}$$

$$h_{\mu_1}^{(2)} = \int_0^{t_1} \sum_{\alpha_2} \Delta_{\alpha_2}^{(0)} z_{\alpha_1}^{(1)} K_{\alpha_1 \mu_1}^x ds + \int_{t_1}^{t} \sum_{\alpha_2} \Delta_{\alpha_2}^{(0)} z_{\alpha_2}^{(1)} K_{\alpha_2 \mu_1}^x ds \tag{62}$$

$$z^{(0)} = \xi \tag{63}$$

$$z^{(1)} = \int_0^t \sum_{\alpha_2} \Delta_{\alpha_2}^{(0)} \chi_{\mu_1} ds \tag{64}$$

$$z^{(2)} = \int_0^t \sum_{\alpha_2} \Delta_{\alpha_2}^{(0)} h_{\alpha_2}^{(1)} ds \tag{65}$$

$$\tag{66}$$

while for the residuals we can identify differential equations involving the evolution of higher order terms in the $\gamma_0$ expansion. From the fields we can obtain the expansion of kernels $\Phi_{\mu_1\alpha_2}, G_{\mu_1\alpha_2}$:

$$\Phi^{(0)}_{\mu_1\alpha_2} = \langle \chi_{\mu_1} \chi_{\alpha_2} \rangle \tag{67}$$

$$\Phi^{(1)}_{\mu_1\alpha_2} = 0 \tag{68}$$

$$\Phi^{(2)}_{\mu_1\alpha_2} = \sum_{\nu_1\beta_1} K^x_{\mu_1\nu_1} K^x_{\alpha_2\beta_1} \int_0^{t_1} dt' \Delta_{\nu_1}(t') \int_0^{t'} dt'' \Delta_{\beta_1}(t'') \tag{69}$$

$$+ \sum_{\nu_2\beta_2} K^x_{\mu_1\nu_2} K^x_{\alpha_2\beta_2} \int_{t_1}^t dt' \Delta_{\nu_2}(t') \int_{t_1}^{t'} dt'' \Delta_{\beta_2}(t'') + sym \tag{70}$$

$$+ \sum_{\nu_1\beta_1} K^x_{\mu_1\nu_1} K^x_{\alpha_2\beta_1} \left[ \int_0^t dt' \Delta_{\nu_1}(t') \right] \left[ \int_0^s ds' \Delta_{\beta_1}(s') \right] \tag{71}$$

$$+ \sum_{\nu_2\beta_2} K^x_{\mu_1\nu_2} K^x_{\alpha_2\beta_2} \left[ \int_0^t dt' \Delta_{\nu_2}(t') \right] \left[ \int_0^s ds' \Delta_{\beta_2}(s') \right] \tag{72}$$

$$G^{(0)}_{\mu_1\alpha_2} = \langle \xi\xi \rangle \tag{73}$$

$$G^{(2)}_{\mu_1\alpha_2} = 0 \tag{74}$$

$$G^{(2)}_{\mu_1\alpha_2} = \sum_{\alpha_1\beta_1} K^x_{\alpha_1\beta_1} \int_0^t dt' \Delta_{\alpha_1}(t') \int_0^{t'} dt'' \Delta_{\beta_1}(t'') \tag{75}$$

$$+ \sum_{\alpha_2\beta_2} K^x_{\alpha_2\beta_2} \int_{t_1}^t dt' \Delta_{\alpha_2}(t') \int_{t_1}^{t'} dt'' \Delta_{\beta_2}(t'') \tag{76}$$

$$+ \sum_{\alpha_1\beta_1} K^x_{\mu_1\alpha_1} \left[ \int_0^t dt' \Delta_{\alpha_1}(t') \right] \left[ \int_0^s ds' \Delta_{\alpha_1}(s') \right] \tag{77}$$

$$+ \sum_{\alpha_2\beta_2} K^x_{\mu_2\alpha_2} \left[ \int_{t_1}^t dt' \Delta_{\alpha_2}(t') \right] \left[ \int_{t_1}^s ds' \Delta_{\alpha_2}(s') \right] \tag{78}$$

$$\tag{79}$$

and the residuals' expansion reads as:

$$\frac{\partial \Delta^{(0)}_{\mu_1}}{\partial t} = U(t_1-t) \sum_{\alpha_1} K^x_{\mu_1\alpha_1} \Delta^{(0)}_{\alpha_1} + U(t-t_1) \sum_{\alpha_2} K^x_{\mu_1\alpha_2} \Delta^{(0)}_{\alpha_2} \tag{80}$$

$$\frac{\partial \Delta^{(1)}_{\mu_1}}{\partial t} = U(t_1-t) \sum_{\alpha_1} K^x_{\mu_1\alpha_1} \Delta^{(1)}_{\alpha_1} + U(t-t_1) \sum_{\alpha_2} K^x_{\mu_1\alpha_2} \Delta^{(1)}_{\alpha_2} \tag{81}$$

$$\frac{\partial \Delta^{(2)}_{\mu_1}}{\partial t} = U(t_1-t) \left( \sum_{\alpha_1} K^x_{\mu_1\alpha_1} \Delta^{(2)}_{\alpha_1} + K^{(2)}_{\mu_1\alpha_1} \Delta^{(0)}_{\alpha_1} \right) + U(t-t_1) \left( \sum_{\alpha_2} K^x_{\mu_1\alpha_2} \Delta^{(2)}_{\alpha_2} + K^{(2)}_{\mu_1\alpha_2} \Delta^{(0)}_{\alpha_2} \right) \tag{82}$$

$$\tag{83}$$

that with the initial condition $\Delta_{\mathcal{T}_1}(0) = \Delta_{\mathcal{T}_2}(0) = Y$ we obtain $\Delta_{\mathcal{T}_1}^{(0)} = \Delta_{\mathcal{T}_2}^{(0)} = Y$ and $\Delta_{\mathcal{T}_1}^{(i)}(t) = \Delta_{\mathcal{T}_2}^{(i)}(t) = 0 \quad \forall i \in [1, \infty)$ and for all times $t$. Giving these $\gamma_0$ expansion coefficients, can compute the CF expansion through 11, obtaining:

$$\frac{dCF(t)}{dt} = -\sum_{\mu_1\mu_2}(\Delta_{\mu_1}^{(0)} + \gamma_0^2\Delta_{\mu_1}^{(2)} + ...)(K_{\mu_1\mu_2}^{(0)} + \gamma_0^2 K_{\mu_1\mu_2}^{(2)})(\Delta_{\mu_2}^{(0)} + \gamma_0^2\Delta_{\mu_2}^{(2)} + ...) \tag{84}$$

$$= -\sum_{\mu_1\mu_2}\Delta_{\mu_1}^{(0)}K_{\mu_1\mu_2}^{(0)}\Delta_{\mu_2}^{(0)} - \gamma_0^2\sum_{\mu_1\mu_2}\left(\Delta_{\mu_1}^{(0)}K_{\mu_1\mu_2}^{(2)}(s)\Delta_{\mu_2}^{(0)}(s) + \Delta_{\mu_1}^{(2)}(s)K_{\mu_1\mu_2}^{(0)}\Delta_{\mu_2}^{(0)}(s) + \Delta_{\mu_1}^{(0)}(s)K_{\mu_1\mu_2}^{(0)}\Delta_{\mu_2}^{(2)}(s)\right) + \dots \tag{85}$$

$$= \frac{dCF^{(0)}}{dt} + \gamma_0^2\frac{dCF^{(2)}}{dt} + \mathcal{O}(\gamma_0^3) \tag{86}$$

$$\tag{87}$$

The zeroth order term represents the NTK limit term and can be solved quite easily in the non-stationary case, keeping in mind the equations above for the residuals' evolutions, that follow exponential decays.

### E.2. CF Perturbation in $\gamma_0$ with Non-Stationarity $\rho$

#### E.2.1. MODELING SIMPLE NON-STATIONARITIES

Calling the input dimension $D$ and the number of data points per task $P$, we adopted the following setup:

- The inputs covariance matrices across *same* tasks are identity matrices, $K_{\mathcal{T}_i\mathcal{T}_i}^x = \mathbb{I}$, with $i \in \{1, \dots, T\}$. This is equivalent to having orthonormal samples.

- The inputs covariance matrices across *different* tasks are proportional to identity matrices, so $K_{\mathcal{T}_i\mathcal{T}_j}^x = \rho\mathbb{I}$, with $i \neq j \quad i, j \in \{1, \dots, T\}$ and $\rho \in [0, 1]$. In this way, we can smoothly control the similarity across tasks, with the totally orthogonal setting of $\rho = 0$ and the same task setting of $\rho = 1$. Different tasks' samples correspond to rotations of other tasks' samples with a certain angle of which cosine is $\rho$. To clarify, imagine the samples' matrix $X_{\mathcal{T}_1} \in \mathbb{R}^{D\times P}$ as:

$$\begin{bmatrix} 1 & 0 & \dots & 0 \\ 0 & 1 & 0 & 0 \\ \vdots & \vdots & \ddots & \vdots \\ 0 & 0 & 0 & 1 \\ 0 & 0 & \dots & 0 \\ \vdots & \vdots & \ddots & \vdots \\ 0 & 0 & \dots & 0 \end{bmatrix} \tag{88}$$

with P columns and D rows, and the identity matrix that lies in the upper part of it. The second task matrix $X_{\mathcal{T}_2}$ can be constructed as:

$$\begin{bmatrix} \rho & 0 & \dots & 0 \\ 0 & \rho & 0 & 0 \\ \vdots & \vdots & \ddots & \vdots \\ 0 & 0 & 0 & \rho \\ a_{P+1} & b_{P+1} & \dots & z_{P+1} \\ \vdots & \vdots & \ddots & \vdots \\ a_D & b_D & \dots & z_D \end{bmatrix} \tag{89}$$

The last step is building an orthonormal base in the lower part of the matrix, which is always possible if we have enough degrees of freedom for the coefficients $\{\{a_i\}_{i=P+1,\dots,D}, \dots, \{z_i\}_{i=P+1,\dots,D}\}$, i.e. $D - P \geq P$. Thus the condition is the one mentioned in the main text of $D \geq 2P$. This corresponds to a rotation in a D-dimensional space of P orthonormal vectors.

The data modeling process above can thus be summarized as:

$$K^x_{\alpha_i \beta_j} = \begin{cases} \rho & \alpha = \beta, i \neq j \\ 1 & \alpha = \beta, i = j \\ 0 & \text{otherwise} \end{cases}$$

### E.2.2. INFINITE-TIME LIMIT OF A LINEAR MLP

In order to simplify the previous expansions, and with the goal of reaching a simple and interpretable formulation of forgetting w.r.t. the parameter $\gamma_0$, we embark on further exploration of the CF perturbation in $\gamma_0$ for infinitely long training ($t \to \infty$) of a linear 1-hidden-layer MLP. To reach a final formulation of forgetting, we need to formulate the expansions of $\Delta_{\mu_1}$ and $\Delta_{\mu_2}$ at the end of training the second task. To get these expressions we first need to derive the expression of the same quantities at the end of training the first task (for an infinitely long time).

We recall that we model the non-stationarity level as $K^x_{\mathcal{T}_1 \mathcal{T}_1}, K^x_{\mathcal{T}_2 \mathcal{T}_2} = \mathbb{I}$ and the across tasks kernel $K^x_{\mathcal{T}_1 \mathcal{T}_2} = \rho \mathbb{I}$.

First, we write down the backward and forward fields following perturbation theory up to the second order:

$$h_{\mu_1}(t) = \chi_{\mu_1} + \gamma_0 \int_0^t ds \sum_{\alpha_1} K^x_{\mu_1 \alpha_1} z_{\alpha_1}(s) \Delta_{\alpha_1}(s)$$

$$= \chi_{\mu_1} + \gamma_0 \int_0^t ds \sum_{\alpha_1} K^x_{\mu_1 \alpha_1} (\xi + \gamma_0 z^{(1)}(s) + \gamma_0 z^{(2)}(s))(\Delta^{(0)}_{\alpha_1}(s) + \gamma_0 \Delta^{(0)}_{\alpha_1}(s) + \gamma_0^2 \Delta^{(2)}_{\alpha_1}(s))$$

$$= \chi_{\mu_1} + \gamma_0 \int_0^t ds \sum_{\alpha_1} K^x_{\mu_1 \alpha_1} \xi \Delta^{(0)}_{\alpha_1}(s) + \gamma_0^2 \int_0^t ds \sum_{\alpha_1} K^x_{\mu_1 \alpha_1} z^{(0)} \Delta^{(0)}_{\alpha_1}(s)$$

$$= \chi_{\mu_1} + \gamma_0 \int_0^t ds \sum_{\alpha_1} K^x_{\mu_1 \alpha_1} \xi \Delta^{(0)}_{\alpha_1}(s) + \gamma_0^2 \int_0^t ds \sum_{\alpha_1} K^x_{\mu_1 \alpha_1} \Delta^{(0)}_{\alpha_1}(s) \int_0^s ds' \chi_1 \Delta^{(0)}_{\alpha_1}(s')$$

$$z(t) = \xi + \gamma_0 \int_0^t ds \sum_{\alpha_1} h_{\alpha_1} \Delta_{\alpha_1}(s)$$

$$= \xi + \gamma_0 \int_0^t ds \sum_{\alpha_1} (\chi_{\alpha_1} + \gamma_0 h^{(1)}_{\alpha_1}(s) + \gamma_0^2 h^{(2)}_{\alpha_1}(s))(\Delta^{(0)}_{\alpha_1}(s) + \gamma_0 \Delta^{(1)}_{\alpha_1}(s) + \gamma_0^2 \Delta^{(2)}_{\alpha_1}(s))$$

$$= \xi + \gamma_0 \int_0^t ds \sum_{\alpha_1} \Delta^{(0)}_{\alpha_1}(s) \chi_{\alpha_1} + \gamma_0^2 \int_0^t ds \sum_{\alpha_1} h^{(1)}_{\alpha_1}(s) \Delta^{(0)}_{\alpha_1}(s)$$

$$= \xi + \gamma_0 \int_0^t ds \sum_{\alpha_1} \Delta^{(0)}_{\alpha_1}(s) \chi_{\alpha_1} + \gamma_0^2 \int_0^t ds \sum_{\alpha_1} \xi \Delta^{(0)}_{\alpha_1}(s) \int_0^s ds' K^x_{\alpha_1 \alpha_1} \Delta^{(0)}_{\alpha_1}(s')$$

This allows us to rewrite equations as:

$$h_{\mu_1}(t) = \chi_{\mu_1} + \sum_{\alpha_1} \gamma_0 K^x_{\mu_1\alpha_1} \xi A_{\alpha_1}(t) + \sum_{\alpha_1} \gamma_0^2 K^x_{\mu_1\alpha_1} \chi_{\mu_1} B_{\alpha_1}(t)$$

$$= \chi_{\mu_1} + \sum_{\alpha_1} \gamma_0 \xi A_{\alpha_1}(t) + \sum_{\alpha_1} \gamma_0^2 \chi_{\mu_1} B_{\alpha_1}(t)$$

$$z(t) = \xi + \sum_{\alpha_1} \gamma_0 \chi_{\mu_1} A_{\alpha_1}(t) + \sum_{\alpha_1} \gamma_0^2 \xi K^x_{\mu_1\alpha_1} B_{\alpha_1}(t)$$

$$= \xi + \sum_{\alpha_1} \gamma_0 \chi_{\mu_1} A_{\alpha_1}(t) + \sum_{\alpha_1} \gamma_0^2 \xi B_{\alpha_1}(t)$$

$$h_{\mu_2}(t) = \chi_{\mu_2} + \sum_{\alpha_1} \gamma_0 K^x_{\mu_1\alpha_2} \xi A_{\alpha_1}(t) + \sum_{\alpha_1} \gamma_0^2 K^x_{\mu_1\alpha_2} \chi_{\mu_1} B_{\alpha_1}(t)$$

$$= \chi_{\mu_2} + \sum_{\alpha_1} \gamma_0 \xi \rho A_{\alpha_1}(t) + \sum_{\alpha_1} \gamma_0^2 \rho \chi_{\mu_1} B_{\alpha_1}(t)$$

where we have defined the function $A_{\mu_1}(t) = \int_0^t ds \Delta^{(0)}_{\mu_1}(s)$ and $B_{\mu_1}(t) = \int_0^t \Delta^{(0)}_{\mu_1}(s) \int_0^s ds' \Delta^{(0)}_{\mu_1}(s')$. Now, having all we need to compute kernels, these latter are obtained through expectations of the inner product between fields, leading to:

$$H_{\alpha_2\mu_1}(t) = \langle h_{\alpha_2}(t) h_{\mu_1}(t)^\top \rangle$$

$$= \rho + \gamma_0^2 \rho \Big( A_{\alpha_1}(t)^2 + 2B_{\alpha_1}(t) \Big)$$

$$H_{\alpha_1\mu_1}(t) = \langle h_{\alpha_1}(t) h_{\mu_1}(t)^\top \rangle$$

$$= 1 + \gamma_0^2 \Big( A_{\alpha_1}(t)^2 + 2B_{\alpha_1}(t) \Big)$$

$$G(t) = \langle z(t) z(t)^\top \rangle$$

$$= 1 + \gamma_0^2 \Big( A_{\alpha_1}(t)^2 + 2B_{\alpha_1}(t) \Big)$$

composing the NTK expansion in $\gamma_0$:

$$K_{\alpha_2\mu_1}(t) = H_{\alpha_2\mu_1}(t) + G(t) K^x_{\alpha_2\mu_1} = \rho + \gamma_0^2 \rho \Big( A_{\alpha_1}(t)^2 + 2B_{\alpha_1}(t) \Big) + \rho \Big( 1 + \gamma_0^2 \Big( A_{\alpha_1}(t)^2 + 2B_{\alpha_1}(t) \Big) \Big)$$

$$= \underbrace{2\rho}_{K^{(0)}_{\alpha_2\mu_1}} + \gamma_0^2 \underbrace{2\rho \Big( A_{\alpha_1}(t)^2 + 2B_{\alpha_1}(t) \Big)}_{K^{(2)}_{\alpha_2\mu_1}}$$

Now we can expand also the residuals in powers of $\gamma_0$, resulting in a set of differential equations:

$$\partial_t \Delta_{\mu_1} = -\sum_{\alpha_1} K_{\mu_1\alpha_1} \Delta_{\alpha_1}$$

$$\partial_t (\Delta^{(0)}_{\mu_1} + \gamma_0 \Delta^{(1)}_{\mu_1} + \gamma_0^2 \Delta^{(2)}_{\mu_1}) = -\sum_{\alpha_1} (K^{(0)}_{\mu_1\alpha_1} + \gamma_0^2 K^{(2)}_{\mu_1\alpha_1})(\Delta^{(0)}_{\alpha_1} + \gamma_0 \Delta^{(1)}_{\alpha_1} + \gamma_0^2 \Delta^{(2)}_{\alpha_1})$$

that can be rearranged as:

$$\begin{cases} \partial_t \Delta_{\mu_1}^{(0)} = -\sum_{\alpha_1} K_{\mu_1\alpha_1}^{(0)} \Delta_{\alpha_1}^{(0)}, & \Delta_{\alpha_1}^{(0)}(0) = y_{\alpha_1} \\ \partial_t \Delta_{\mu_1}^{(1)} = -\sum_{\alpha_1} K_{\mu_1\alpha_1}^0 \Delta_{\alpha_1}^{(1)}, & \Delta_{\alpha_1}^{(1)}(0) = 0 \\ \partial_t \Delta_{\mu_1}^{(2)} = -\sum_{\alpha_1}(K_{\mu_1\alpha_1}^{(0)} \Delta_{\alpha_1}^{(2)} + K_{\mu_1\alpha_1}^{(2)} \Delta_1^{(0)}), & \Delta_{\alpha_1}^2(0) = 0 \end{cases}$$

$$\begin{cases} \partial_t \Delta_{\mu_2}^{(0)} = -\sum_{\alpha_1} K_{\mu_2\alpha_1}^{(0)} \Delta_{\alpha_1}^{(0)}, & \Delta_{\alpha_1}^{(0)}(0) = y_{\alpha_1} \\ \partial_t \Delta_{\mu_2}^{(1)} = -\sum_{\alpha_1} K_{\mu_2\alpha_1}^{(0)} \Delta_{\alpha_1}^{(1)}, & \Delta_{\alpha_1}^{(1)}(0) = 0 \\ \partial_t \Delta_{\mu_2}^{(2)} = -\sum_{\alpha_1}(K_{\mu_2\alpha_1}^{(0)} \Delta_{\alpha_1}^{(2)} + K_{\mu_2\alpha_1}^{(2)} \Delta_1^{(0)}), & \Delta_{\alpha_1}^{(2)}(0) = 0 \end{cases}$$

that allows us to conclude that the first order term of both tasks' residuals is 0, namely $\Delta_{\alpha_1}^{(1)}(t) = \Delta_{\alpha_2}^{(1)}(t) = 0$. We can, then, solve the zeroth order differential equation and obtain the lazy regime solutions:

$$\Delta_{\mu_1}^{(0)}(t) = ye^{-2t}$$
$$\Delta_{\mu_2}^{(0)}(t) = y(1-\rho)(1-e^{-2t})$$

and from these equations we can obtain the analytical form of $A_{\alpha_1}(t)$ and $B_{\alpha_1}(t)$, that allows us to compute the second order differential equation, obtaining:

$$\Delta_{\mu_1}^{(2)}(t) = y^3 \left( e^{-2t}(t - \frac{3}{4}) + e^{-4t} - \frac{1}{4}e^{-6t} \right)$$
$$\Delta_{\mu_2}^{(2)}(t) = \rho y^3 \left( e^{-2t}(\frac{4t-3}{4}) + e^{-4t} - \frac{1}{4}e^{-6t} \right)$$

This allows us to inspect the infinite-time limit of task 2 residuals $\Delta_{\mu_2}(t) = \Delta_{\mu_2}^{(0)}(t) + \gamma_0^2 \Delta_{\mu_2}^{2)}(t) + ...$ after training on task 1, that gives us the following results:

$$\lim_{t\to\infty} \Delta_{\mu_2}^{(0)}(t) = y(1-\rho)$$
$$\lim_{t\to\infty} \Delta_{\mu_2}^{(2)}(t) = 0$$
$$\lim_{t\to\infty} \Delta_{\mu_2}(t) = y(1-\rho)$$

so that the final loss value expansion till the second order reads as:

$$\lim_{t\to\infty} \frac{1}{2}(\Delta_{\mu_2}(t))^2 = \frac{y^2}{2}(1-\rho)^2 \tag{90}$$

From this equation, we can conclude that the infinite-time limit of task 2 loss after infinitely long training on task 1, is independent of $\gamma_0$, and this is validated empirically in Fig. 25.

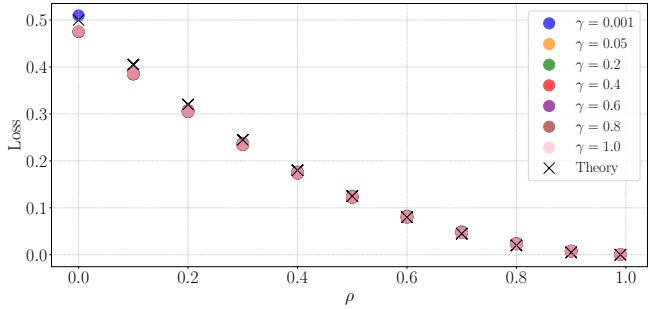

Figure 25. Last value of the loss on Task $\mathcal{T}_2$ after (long) training on Task $\mathcal{T}_1$, on a single hidden layer NN, linear activation, and Gaussian data in order to tune the task similarity and obtain the theoretical setup. Black crosses represent the theoretical values at different levels of similarity, following equation 90. At low $\rho$ the theoretical constraints on the setup become weaker due to the intrinsic noise level of Gaussian data.

Now, to reach the expression for forgetting, we would need to express the residuals after training on the second task. However, such a lengthy formulation is out of scope for this work, and is thus left for future development. Nevertheless, what we can easily achieve from the formulation so far, is the infinite-time limit expression for the lazy limit ($\gamma_0 = 0$, i.e. the expression for $CF^{(0)}$).

In order to do so, we write down the dynamics for residuals of both tasks during first task training, which read as:

$$\Delta_{\mu_1} = ye^{-2t}$$
$$\Delta_{\mu_2} = y(1-\rho)(1 - e^{-2t})$$

that will converge to the following limiting behaviors:

$$\lim_{t\to\infty} \Delta_{\mu_1} = 0$$
$$\lim_{t\to\infty} \Delta_{\mu_2} = y(1-\rho)$$

This represents the "new initial conditions" for the second task differential equations of the lazy regime, which are:

$$\partial_t \Delta_{\mu_1}^{(0)} = -\sum_{\alpha_2} K_{\mu_1\alpha_2}^{(0)} \Delta_{\alpha_2}^{(0)}$$
$$\partial_t \Delta_{\mu_2}^{(0)} = -\sum_{\alpha_2} K_{\mu_2\alpha_2}^{(0)} \Delta_{\alpha_2}^{(0)},$$

Solving the above ODE we obtain the following equations for residuals, where we simplify the notation by writing time $t - t_1$ as $t$, considering the effective training time from the beginning of the second task training. This allows us to eliminate the factors that vanish when plugging in the assumption of an infinite training time for the first task ($t_1 \to \infty$). Thus we obtain:

$$\Delta_{\mu_1}^{(0)} = y\rho(1-\rho)(e^{-2t} - 1)$$
$$\Delta_{\mu_2}^{(0)} = y(1-\rho)e^{-2t}$$

and computing the limiting values of both losses is now straightforward since $\mathcal{L}_{\alpha_i} = \frac{1}{2}\Delta_{\alpha_i}^2$:

$$\lim_{t\to\infty} \mathcal{L}_{\mu_1}^{(0)}(t) = \frac{y^2}{2}\rho^2(1-\rho)^2$$
$$\lim_{t\to\infty} \mathcal{L}_{\mu_2}^{(0)}(t) = 0$$

Since CF in terms of losses is defined as $CF = \mathcal{L}(t) - \mathcal{L}(t_1)$, under these simplified conditions, $\lim_{t\to\infty} \mathcal{L}(t_1) = 0$, the infinite-time behavior of Catastrophic Forgetting is captured by:

$$\lim_{t\to\infty} CF^{(0)}(t) = \frac{y^2}{2}\rho^2(1-\rho)^2$$

with a quadratic dependence on similarity level $\rho$, maximized at exactly $\rho = 0.5$, as well known for linear models (Goldfarb et al., 2024).

