# OpenReview forum: "The Importance of Being Lazy: Scaling Limits of Continual Learning"
_ICML.cc/2025/Conference — ICML 2025 poster_

### Official Review · Reviewer_DJW3 · 2025-02-27

**Overall Recommendation:** 3

**Summary:**

This paper explores the relationship between scaling regimes and catastrophic forgetting using the lens of dynamical mean field theory (DMFT). The authors demonstrate theoretically that in feature learning regimes, catastrophic forgetting is more likely. In particular, there is a sharp transition between the lazy and rich regimes with respect to forgetting, which the authors term as edge of laziness (EoL). They show that in continual learning settings, there is an optimal choice of laziness level, which is transferable across model capacities. The authors further note that as forgetting becomes less important (as tasks are more similar), the optimal level of richness increases (optimal laziness decreases).

**Claims And Evidence:**

Overall, the claims made in the paper are generally well supported by theory and experiments.

One area of potential concern is with the terminology "edge of laziness." The experimental results in Figure 3 could be interpreted either as a discrete transition between a lazy and rich regime, or a sharp, but still continuous transition. If the latter is the case, then I believe "edge of laziness" may not be the correct term to describe the results. Can the authors justify why the former is the case? Is there a phase transition at this point that the authors can show theoretically?

**Essential References Not Discussed:**

As far as I am aware, the relevant literature is discussed.

**Experimental Designs Or Analyses:**

Experimental setup as described in the captions and throughout the main text appears sounds.

**Methods And Evaluation Criteria:**

Evaluations are conducted on Permuted-MNIST and Split-CIFAR, which are standard benchmarks.

**Other Comments Or Suggestions:**

It looks like the labels in Figure 6 are cut off. I would also recommend increasing the size of the Figure (perhaps some of the whitespace could be removed).

**Other Strengths And Weaknesses:**

Overall, the paper is well-written and presented and makes a novel contribution to the space of continual learning theory.

A key weakness of the theoretical results is that the DMFT techniques used by the authors applies to one hidden layer neural networks. Experimentally, the authors consider deeper networks though, which somewhat alleviates this concern.

**Questions For Authors:**

As mentioned above, can the authors theoretically demonstrate a phase transition between the lazy and rich regime (for edge of laziness)?

Also, can the authors theoretically show a connection between the edge of laziness and the optimal stability-plasticity tradeoff? For what kinds of tasks is it true that the optimal $\gamma_0$ exists at the edge of laziness?

**Relation To Broader Scientific Literature:**

As the authors note, previous works have already considered applying scaling limit results to continual learning. The key difference with this work is that they consider large networks in the rich regime under finite data, a previously unexplored setting for continual learning.

**Theoretical Claims:**

The theoretical claims in the main text appear correct. They seem to rely on standard arguments in DMFT.

---

> ### Author Rebuttal · Authors · 2025-04-01
>
> *Thank you for taking the time to read and review our paper. We are glad to hear that you found our paper well-written and that it offers a novel contribution to the space of continual learning.*
>
> ---
>
> ## Edge of Laziness and Phase Transition
>
> Thank you for your thoughtful comment, and we are sorry for the confusion caused by the terminology of Edge of Laziness.
>
> Firstly, we would like to take the opportunity to clarify our position on the observed phase transition. In our results, it is clear that a phase transition happens, however, it does not always appear as a sharp, discontinuous transition between the two regimes, but more like a non-linear continuous function of $\gamma_0$ in the area where the transition occurs. For this reason, we do not make any claims about the nature of the transition, being it *discrete* or *sharp but still continuous*.
>
> Secondly, our choice of terminology, "Edge of Laziness," was not intended to rigorously characterize the phase transition. Instead, it was intended to be evocative of a boundary between an effectively lazy region of the $\gamma_0$-space.
>
> Thirdly, we are currently unable to demonstrate the nature of the phase transition theoretically, a result that we believe to be highly non-obvious. Nevertheless, we are actively working towards this result, and we were recently able to characterize the infinite-time solution of CF in the rich regime of the linear case. In this setting, we observe a highly non-trivial relation between the task similarity $\rho$ and $\gamma_0$, where the second-order coefficient of CF is a sixth-grade polynomial in $\rho$. The complexity of this relationship - even in the linear case - points to the non-triviality of the behavior and thus to the difficulty of getting a general characterization of the phase transition theoretically. Nevertheless, we will continue working to extend this result to the non-linear setting for future work.
>
> Following your comment, and determined to avoid any confusion or misunderstanding with future readers, we have rephrased the parts of the paper that introduce and discuss the concept of the edge of laziness. We decided to refer to it as *lazy-rich transition*, avoiding the word "edge". We are also in the process of changing the title to reflect this change of terminology accordingly. We are grateful for the opportunity to improve the clarity of our claims and paper.
>
> ## Other Comments
>
> - **DMFT with one hidden layer NN**
>
>     We plan to extend the DMFT to deeper networks in future work. In the meantime, however, we would like to stress the striking overlap between the results in the shallow and deep networks, suggesting that the one-layer derivation is already able to represent well the behaviors of deep networks.
> - **Edge of Laziness, tradeoff, and optimal $\gamma_0$**
>
>     Thanks for the interesting question, which allows us to touch upon two distinct phenomena that influence the overall optimal level of richness.
>
>     Firstly, we have observed that the task similarity has a direct and powerful impact on the feature evolution itself, meaning that a higher similarity effectively reduces the amount of feature learning for fixed $\gamma_0$ (Fig. 5a). This suggests that the task similarity (or in general the degree of (non-)stationarity of the data) non-trivially interacts with the regimes of the network. Including this perspective into the DMFT formalism is non-trivial, and we have just started exploring this avenue (see Appendix E3, F). However, we believe this direction would be very interesting and valuable for future work.
>
>     Secondly, the optimal $\gamma_0^\star$ reflects the plasticity-stability tradeoff, and therefore, $\gamma_0^\star$ can shift from the $\gamma_0$ at which the *lazy-rich transition* happens. Generally, we find that when the tasks are close to stationary, the optimal $\gamma_0^\star$ is always 1, i.e. the maximum of the range tested. In other words, when the data is stationary, higher plasticity helps reach better performance. By contrast, as the amount of non-stationarity is increased, stability is necessary to avoid losing performance on the old data and the optimal $\gamma_0^\star$ is consistently lower than 1. From this point of view, the $\gamma_0$ at which the *lazy-rich transition* happens represents the minimal $\gamma_0^\star$ as any value below it will not increase stability (because the network is effectively lazy) and it will not increase plasticity.
>
>     While we do observe the plasticity-stability tradeoff even in the theoretical experiments at infinite width (Fig. 4a), we have not specifically investigated it theoretically.
>
> - **Fig. 6 labels**
>
>     Thanks for pointing out the cut-off labels of Fig. 6, and for suggesting an increased size of the Figure. We will correct these in the final version of the paper.
>
> ---
>
> *We hope that we have adequately addressed your questions. If you have further questions or comments, we remain at your disposal.*

---

### Official Review · Reviewer_6Eo8 · 2025-03-06

**Overall Recommendation:** 3

**Summary:**

This paper studies how neural network parameterization (at the extremes, NTP and $\mu$P) shapes the effect of network width on catastrophic forgetting.

## Update after rebuttal

My assessment of the paper remains unchanged. I think this is an interesting contribution, but I am still skeptical of the added value provided by the DMFT analysis.

**Claims And Evidence:**

The claims are well-supported.

**Essential References Not Discussed:**

The authors do a generally good job of reviewing related prior art, but there are few missing references.

- Petrini et al. ("Learning sparse features can lead to overfitting in neural networks", NeurIPS 2022) studied how very rich feature learning can lead to overfitting to spurious features. This bears a close conceptual relation to the results here on severe catastrophic forgetting in very rich networks. Moreover, it offers a previous example setting where more feature learning is not better.

- Vayas et al. ("Feature-Learning Networks Are Consistent Across Widths At Realistic Scales", NeurIPS 2023) show that beyond some minimal width, $\mu$P-parameterized networks of different widths behave very similarly, which is related to the authors' findings in Figure 2.

**Experimental Designs Or Analyses:**

The experiments are generally well-designed, and the authors provide a good selection of additional figures in the Appendices.

**Methods And Evaluation Criteria:**

The methods are appropriate.

**Other Comments Or Suggestions:**

- In figure lables, the authors should just write 1-CKA instead of "Features evolution, 1-CKA".

- Mentions of Bordelon et al (2023) for depth scaling in ResNets should cite also the contemporaneous work of Yang et al.

- Please make sure to state the dataset and number of samples used in figure captions (e.g. this is missing in Figure 4).

- I think the claim in the Discussion that "our findings complete the picture on feature learning in modern NNs" is too broad, particularly given previous work on settings where feature learning harms generalization.

**Other Strengths And Weaknesses:**

- I do not think the title does a good job of conveying the main contributions of the paper. I would suggest something that makes the main conclusion obvious. To give a very rough example, perhaps something like "Rich feature learning can accentuate catastrophic forgetting".

- One substantial concern I have is with the role DMFT plays in the paper, as I think the space devoted to it in the main text might better be used to present additional experimental results (for instance, the experiments on training time could be promoted to the main text). To be very clear, I am by training a statistical physicist, so I think it's nice that the authors have the DMFT description. However, it is not enough to write down the self-consistent equations; there must be some conceptual meat extracted from them. My concerns are (1) that the DMFT results do not substantially strengthen the claims based on experiment, and (2) that the technical novelty here relative to the cited work of Bordelon and Pehlevan is minimal. What do we gain conceptually from the perturbative approximation?

- The authors do not adequately discuss how some of their findings (particularly the relatively sharp increase in CKA with $\gamma\_0$) relate to those of Atanasov et al. 2024 ("The Optimization Landscape of SGD Across the Feature Learning Strength"), which documents similar phenomenology. This is also relevant to their choice of scaling of learning rate with $\gamma\_0$.

- I think the paper would benefit from some futher analysis of representational changes across tasks. For instance, the authors could plot as a function of training time the kernel-target alignment (as in Atanasov et al. 2024) for each task. This would help clarify precisely what structure is learned and forgotten. Moreover, it would help relate this work to the abovementioned work of Petrini et al. 2022.

- The paper is missing some methodological details on how the authors solved the DMFT equations. In line 771, the authors state that they "implemented simple computational physics discrete-time dynamics", but this is too vague to be useful. Presumably they just modified the solver from Bordelon & Pehlevan, but this should be stated.

**Questions For Authors:**

I am a bit puzzled by the use of "capacity" at a few points in this paper, e.g. in line 352 where it is stated that increasing $\gamma\_0$ increases capacity. I would associate capacity more with expressivity than with learning-related features. Can you clarify this?

**Relation To Broader Scientific Literature:**

This paper bridges two bodies of work in machine learning: that on optimal network parameterizations, and that on catastrophic forgetting. It is well-situated within the literature, and should be of broad interest to the ICML audience.

**Theoretical Claims:**

The theoretical claims are straightforward extensions of the DMFT results of Bordelon & Pehlevan (2021), and appear sound.

---

> ### Author Rebuttal · Authors · 2025-04-01
>
> *We are grateful for your time reading our paper, and thank you for your thoughtful review.*
>
> ---
>
> ## The role of DMFT in the paper
>
> We agree that the results of the DMFT are somewhat abstract, however we believe this is an inherent limitation of this theory: it is hard to simplify the results and reach interpretable results without sacrificing its completeness. Committed to this direction for future work, we are currently working on integrating minimal assumptions that could help us simplify the results without trivializing them.
>
> Nevertheless, we would like to address your opinions that "(1) the DMFT results do not substantially strengthen the claims based on experiments", and that "(2) the technical novelty here is limited".
>
> 1. The DMFT equations allow us to actually reach the infinite-width limit, something that would have been impossible otherwise. Although it is unusual to lay the theory at the service of the experimental evidence, we believe that the infinite-width curves crucially strengthen our finite-width insights of the experiments.
>
> 2. We agree that the DMFT techniques are a known approach to analyze the network behavior. Nevertheless, we believe that our extension to the Continual Learning scenario is notable and highly valuable for the community, as we
>
>      - introduce the NTK across tasks as a new entity of the DMFT formalism
>      - are the first work to analyze feature learning in CL
>      - propose a perturbative approximation which is different from previous approaches (e.g., Bordelon and Pehlevan), as we also expand the residuals in powers of $\gamma_0$, and we apply it to the CL scenario. This allows us to study the complex dependence between $\gamma_0$ and the task similarity $\rho$, obtaining a closed-form solution for the coefficients of CF, up to the second order, in the linear case. Seeking the "conceptual meat", we were recently able to obtain further insights on the role of $\gamma_0$ in the linear case, where we find that in the infinite-time solution, the second-order coefficient of CF is always positive, certifying that larger $\gamma_0$ yields higher forgetting. Moreover, we find that this relationship has a non-trivial dependence on $\rho$ through a sixth-order polynomial, where the maximum CF is obtained at richness-dependent levels of $\rho$. You can find the preliminary plot [here](https://ibb.co/9mQhGZg1).
>
> ## Discussion of related work
> - **Petrini et al., 2022.**, **Vayas et al., 2023.**, **Yang et al., 2023**. Thanks, they should definitely be included in the related works and cited accordingly.
> - **Atanasov et al., 2025.** We would like to point out that this work is concurrent and will be presented at ICLR 2025. Although the paper is related to ours, we feel it is unfair to list as a weakness the missing discussion of such a recent work.
>
>     Nevertheless, their findings are interesting and might be a good starting point for future work. As you suggest, the observed *lazy-rich transition* might be related to our choice of LR scaling (LR scales quadratically with $\gamma_0$): they observe that this scaling is optimal for the lazy regime, but not anymore for the rich and ultra-rich regimes (Fig. 1b), where instead sub-quadratic LR scaling is optimal. Our transition might therefore be due to a change of the effective LR, shifting from the optimal LR towards a too-large LR and thus approaching divergence. Note that this is just an intuition that requires further study.
>
> ## Other comments
> - **Title:** following also the review of DJW3, we have decided to avoid referring to Edge of Laziness, and will therefore opt for a more informative title, as you suggest.
>
> - **Representational Changes:** We have quickly implemented this interesting experiment, and you can find the results for the Permuted-MNIST, restricted to samples 0 and 1, [here](https://ibb.co/Zzwp7QTB). The drop in the alignment after the tasks switch represents an intriguing insight that serves as an interesting starting point for future analysis. We will add this experiment to the Appendix.
>
> - **Implementation of DMFT:** thanks for pointing out the missing details. We use a modified version of the one by Bordelon \& Pehlevan in the L=1 case, extended to support multiple tasks as per our training setting. We will clarify this in the final version.
>
> - **Network Capacity:** we refer to *network capacity* as the network size (i.e. the width), irrespective of $\gamma_0$. In l. 352 we state that "the increased capacity (i.e. size) of the network does not benefit performance".
>
> - **Figure lables:** we agree that the labels are slightly notation-heavy, but we believe that the CKA measure might be obscure to readers not familiar with the kernel literature; the labels provide a useful guidance to such a reader.
>
> - **Claim in the discussion:** we agree and will adapt this claim.
>
> ---
>
> *We hope that we have adequately addressed your questions. If you have further questions or comments, we remain at your disposal.*

---

### Official Review · Reviewer_CVSx · 2025-03-09

**Overall Recommendation:** 3

**Summary:**

This paper investigates the effect of model scale and the degree of feature learning in continual learning. It identifies a transition called Edge of Laziness influenced by task similarity, where the model exits an effectively lazy regime with low forgetting to enter a rich regime with significant forgetting. Technically, it extends the DMFT theory to non-stationary learning. Infinite-width simulations and real-world experiments support its claims.

**Claims And Evidence:**

Yes.

**Essential References Not Discussed:**

NA.

**Experimental Designs Or Analyses:**

Yes. The experiments are conducted on different datasets and support their claims and theoretical results well. For the MNIST dataset, the infinite-width simulations motivated by the DMFT further verify the real-world experiments in finite width.

**Methods And Evaluation Criteria:**

Yes.

**Other Comments Or Suggestions:**

I maintain my score after the rebuttal.

**Other Strengths And Weaknesses:**

Strengths:
1. The paper is written clearly.
2. The paper designs a systematic study on the impact of model scale and the degree of feature learning in continual learning, and the experiments support their claims well.
3. As far as I know, this paper takes a first step to extend the DMFT to continual learning.

**Questions For Authors:**

1. Do the claims in this paper hold for the Transformer architecture trained with Adam optimizer?
2. Following question 1, can the DMFT be extended to an adaptive optimizer like that in the TP 4b paper?
3. In practice, we may train a neural network with $O(width)$ steps by the scaling law. Can we extend the experiments and the DMFT theory to that case?
4. In this paper, DMFT theory is used to conduct simulations to verify the findings in finite-width experiments, which is expensive. Can we predict the existence of phase transition theoretically?
5. In my opinion, the $\gamma_0$ is just an output multiplier hyperparameter in existing mup papers (e.g., TP 5), so it should transfer across different widths. Am I right?

**Relation To Broader Scientific Literature:**

NA.

**Theoretical Claims:**

I did not check the proof carefully, but it seems good, and the experiments support the theoretical results.

---

> ### Author Rebuttal · Authors · 2025-04-01
>
> *Thank you for taking the time to read and review our paper. We are glad to hear that you found our paper clearly written, and that you consider our experiments well-designed, providing a systematic study of the topic and supporting well both claims and theoretical results.*
>
> ---
>
> 1. >**Do the claims in this paper hold for the Transformer architecture trained with Adam optimizer?**
>
>     Despite the success of transformers in DL, their adoption in CL is still limited and fairly understudied [1,2,3]. If transformers in CL are already rare, their optimization with adaptive optimizers (e.g., Adam) is even more so, whereas vanilla SGD is the standard optimizer even when training transformers [4,5]. For these reasons, in our work we decided to study the ResNet architecture with SGD. Nevertheless, we agree that exploring our claims even for transformers and for adaptive optimizers is an interesting avenue for future research.
>
> 2. > **Can the DMFT be extended to an adaptive optimizer like that in the TP 4b paper?**
>
>     The DMFT could potentially be extended from [6]. However, in this work, we were more interested in understanding the dependence of the dynamics on $\gamma_0$, which requires lengthy derivations even for a two-layer linear network trained under gradient flow.
>
> 3. > **In practice, we may train a neural network with $O(width)$ steps by the scaling law. Can we extend the experiments and the DMFT theory to that case?**
>
>     The DMFT relies on the assumptions of the number of datapoints and timesteps of order O(1), beyond which the theory may break down. In this study, an extension to width-dependent training times is out of scope. However, we point to the experiments presented in Appendix B.1, in which we investigate the effect of training time in the NTP. We find that longer training time systematically yields high feature evolution even in the NTP, and therefore it leads to CF.
>
> 4. > **Can we predict the existence of a phase transition theoretically?**
>
>     We are currently unable to predict the phase transition theoretically. Nevertheless, we are actively working towards this result, and we were recently able to characterize the infinite-time solution of CF in the rich regime of the linear case. In this setting, we observe a highly non-trivial relation between the task similarity $\rho$ and $\gamma_0$, where the second-order coefficient of CF is a sixth-grade polynomial in $\rho$. The complexity of this relationship - even in the linear case - points to the non-triviality of the behavior and thus to the difficulty of getting a general characterization of the phase transition theoretically. Nevertheless, we will continue working to extend this result to the non-linear setting for future work.
>
> 5. > **In my opinion, the $\gamma_0$ is just an output multiplier hyperparameter in existing mup papers (e.g., TP 5), so it should transfer across different widths. Am I right?**
>
>     Thanks for the interesting question, which allows us to clarify the role of $\gamma_0$ as well as the novelty of our findings.
>
>     Firstly, $\gamma_0$ is not only an output multiplier, instead it also quadratically modulates the LR (cfr Tab 1 in our paper). This implies that one cannot infer transfer properties of $\gamma_0$ from those of the LR, as the two are non-trivially related.
>
>     Secondly, and crucially, the transfer properties of LR (and other hyperparameters) have only been shown - but not proven analytically - in the stationary setting, and not in the non-stationary and CL scenario. We therefore deem it non-trivial and surprising to observe that, with $\mu$P, $\gamma_0$ shows these transfer properties even in the non-stationary scenario.
>
>     In hindsight, given that width scaling does not affect forgetting under mean field scaling, it is intuitive to expect that $\gamma_0$ should transfer. However, if width scaling were to affect the dynamics (e.g., by lowering forgetting due to increased capacity), then one could in principle expect $\gamma_0$ to scale differently with width. We hope this clarifies the subtlety of these results and the coherence of our findings.
>
> ---
>
> *We hope we have adequately addressed your questions and reservations that hindered your recommendation for acceptance. We remain at your disposal in case you might have further comments or doubts.*
>
> ---
>
> ### References
> - [1] Ramasesh et al., "Anatomy of Catastrophic Forgetting: Hidden Representations and Task Semantics", ICLR 2021
> - [2] Mirzadeh et al., "Wide Neural Networks Forget Less Catastrophically", ICML 2022
> - [3] Lu, et al. "Revisiting neural networks for continual learning: An architectural perspective." arXiv preprint arXiv:2404.14829 (2024).
> - [4] Ramasesh et al., "Effect of scale on catastrophic forgetting in neural networks", ICLR 2021
> - [5] Mirzadeh et al., "Architecture matters in continual learning." arXiv preprint arXiv:2202.00275 (2022).
> - [6] Bordelon et al., "Infinite Limits of Multi-head Transformer Dynamics.", Neurips 2024

---

### Official Review · Reviewer_hpnB · 2025-03-13

**Overall Recommendation:** 4

**Summary:**

The authors present a theoretical and experimental analysis of the effect of the neural network parametrisation on Catastrophic Forgetting.
The study extends previous works which focused on the lazy regime only.
The authors identify a spectrum of training regimes from the lazy regime to the feature learning regime. This spectrum is parametrised with a single parameter gamma.
The authors show that depending on the parametrisation, the width scaling laws is different for Catastrophic Forgetting.
Also, the authors identify an optimal tradeoff between accuracy and forgetting, as a function of the parametrisation.
The tradeoff occurs because forgetting increases the more the feature rich the training regime, while the accuracy increases.
Finally, the authors define and study the EoL, which varies as a function of the stationarity of the tasks.
The more stationary the tasks, the higher the EoL.

**Claims And Evidence:**

The main claims of the paper are :
1- The effect of width scaling on CF depends on the network parametrization. The evidence to support it is :
- Experimental : Measuring CF as a function of the width for the Split-CIFAR-10 and Permuted MNIST tasks.
2- The existence of the Edge of Laziness (EoL), a region which separates to regimes. The lazy regime where the features don't change much and CF is low, the rich regime where the features changes significantly more as well as CF.
- Figure 3, a, b, c : The variation of CFr as a function of gamma_0
- Figure 4, a : The variation of the average loss as a function of gamma_0
3- The optimal gamma_star for the plasticity stability tradeoff is almost identical regardless of the network width :
- Figure 4, b : The pareto front between CF and Learning error
4- High feature learning is only beneficial with highly similar tasks
- Figure 5 C : The average error as a function of the task similarity and gamma_0 on Permuted MNIST

**Essential References Not Discussed:**

I am not aware of any essential references not discussed :)

**Experimental Designs Or Analyses:**

I checked the soundness of all the experiment designed presented in the main paper, but I didn't check the ones in the Appendix.

The authors considered the Permuted MNIST, CIFAR 10 and Split-TinyImageNet tasks for their analysis.
The first two tasks show clearly that the law is satisfied and the third one shows a different behaviour in the rich regime which the authors explain with forward transfer.
One question I have is if there is a specific reason not to consider the CIFAR-100 task, as it lies between the two datasets and would help determine the boundary in terms of data distribution where the pre-training effect would apply or not.

**Methods And Evaluation Criteria:**

The proposed evaluation methods are sensible. The authors use the tasks Permuted MNIST, CIFAR 10 and Split-TinyImageNet for their experimental validation.
The last observation about the Pre-Training effect in Section 6.2 is very interesting and intriguing. The authors show that on Split-TinyImageNet the law is different compared to the other two tasks. I think this experiment is very important to highlight the validity of the law on training setups and datasets closer to real world data.

**Other Comments Or Suggestions:**

Some typos :
- In the contributions section : I think NTP and muP are not defined beforehand, they are defined in the next page.
- Figure 6 - d : The title is clipped

- L163 : Could you briefly explain the cubic complexity without going much in detail ?

**Other Strengths And Weaknesses:**

The paper was really interesting to read and it challenged several intuitions I had. I found the explanations very clear and I particularly appreciated the experimental evidence provided to support and illustrate the claims in practical settings.

Also, I think the contribution is significant because the analysis links between several prior findings and further provides new insights about the impact of parametrization on a scale.

I didn't note any major weaknesses, in the other sections I noted some clarification questions.

**Questions For Authors:**

(The questions below are clarification questions and wouldn't impact the final score)

- In  Figure 1 and Figure 4, the law of the smallest width (64) is significantly more nosy than the other larger widths. Could you share an intuition about why it might be the case ?
- In Figure 6 : Given that ImageNet has a hierarchy of class similarities, how do you control for this similarity when splitting the tasks and classes ?
- Also about the pre-training effect, wouldn't it be expected for it to occur in CIFAR-10 as well. Is it sensible to measure it with the forward transfer metric and compare for TinyImageNet and CIFAR-10 ?

**Relation To Broader Scientific Literature:**

This paper relates to the broader literature in the following ways :
- [1] and [2] provide a theory of CF in the lazy regime, formulating theoretical bounds and a closed form of CF in the NTK regime.
- [3] and [4] observe that increasing the width of the neural networks reduces CF and study the impact of the model architecture on CF.
- [5] study theoretically the effect of overparametrization on CF for linear models.

The paper also relates to the rich feature learning literature, I am not familiar with this research area.

- [1] Doan, Thang Van et al. “A Theoretical Analysis of Catastrophic Forgetting through the NTK Overlap Matrix.” International Conference on Artificial Intelligence and Statistics (2020).
- [2] Bennani, Mehdi et al. (2020). Generalisation Guarantees for Continual Learning with Orthogonal Gradient Descent. ArXiv, abs/2006.11942.
- [3] Mirzadeh, S., Chaudhry, A., Hu, H., Pascanu, R., Gorur, D., & Farajtabar, M. (2021). Wide Neural Networks Forget Less Catastrophically. International Conference on Machine Learning.
- [4] Mirzadeh, S., Chaudhry, A., Yin, D., Nguyen, T., Pascanu, R., Gorur, D., & Farajtabar, M. (2022). Architecture Matters in Continual Learning. ArXiv, abs/2202.00275.
- [5] Evron, Itay, Daniel Goldfarb, Nir Weinberger, Daniel Soudry and Paul Hand. “The Joint Effect of Task Similarity and Overparameterization on Catastrophic Forgetting - An Analytical Model.” ArXiv abs/2401.12617 (2024): n. pag.

**Theoretical Claims:**

- There seems to be a notation inconsistency between the definition of the NTK across tasks (Eq 2) and its use in Eq 12. The definition only has a single time index, while Eq 12 has two time indexes. Also, I suggest clarifying the time index in the definition of Eq 2, in the weights matrix. Currently the definition doesn't explicitly highlight the time / task index.
- General question : Out of curiosity, is it tractable to derive the expression of forgetting at t=infinity from the PDE in Eq 12 ?

I skimmed through the proof but haven't checked it in detail.

---

> ### Author Rebuttal · Authors · 2025-04-01
>
> *We sincerely thank you for taking the time to review our paper and for the thoughtful feedback. We are glad to hear that you found the work really interesting and stimulating. We are also grateful to hear that you appreciate our contribution to the literature.*
>
> ---
>
> ## The Pretraining Effect
>
> We are glad you found the pretraining effect particularly interesting and intriguing; we share your enthusiasm for this phenomenon. We agree that these observations offer an important perspective on our results, in particular as the setting approaches training scenarios closer to the real world. In this regard, we think you might be interested in inspecting also the interplay between the pretraining effect and width scaling, as reported in Figures 17-19 of the appendix.
>
> Before answering your specific questions, we would like to clarify that the design choices for our experiments targeted the main goals of the paper. However, we agree with you that looking into the pre-training effect more thoroughly is certainly very interesting.
>
> ### Split-CIFAR
> > Is there a specific reason not to consider the CIFAR100 task?
>
> We find no specific reason for not considering CIFAR100 in our experiments. We have chosen CIFAR10 as a representative dataset for our experiments for computational convenience, and then directly extended our analysis to Split-TinyImagenet, aiming to provide stronger empirical evidence on a more challenging dataset. Nevertheless, we agree that CIFAR100 might be an interesting in-between dataset and we thank you for pointing it out.
>
> > Wouldn't the pretraining effect be expected for it to occur in CIFAR10 as well?
>
> In Split-TinyImageNet, we observe the pretraining effect happening for a larger number of classes per task, namely when the tasks have enough data diversity, and thus a greater overlap between task distributions. Our intuition is therefore that the reason why this behavior is not observed in CIFAR10, is that the number of classes is too small to allow for a significant overlap between task distributions.
>
> ### Other Questions on the Pretraining Effect
> > Is it sensible to measure the pretraining effect with the forward transfer metric?
>
> We haven't considered measuring the pretraining effect with the forward transfer metric, instead, we have focused on disentangling the feature evolution of the first and later tasks. However, we agree that the forward transfer metric could be a valuable addition to our analysis and we will look into integrating it.
>
> > How do you control the hierarchy of class similarities in Split-TinyImageNet?
>
> We naively split sequentially the classes of TinyImageNet, without accounting for the semantics of the classes. However, when we vary the number of classes per task, we ensure to keep fixed the classes-to-task assignment, to have a fair comparison across runs with different nr. of classes per task.
>
> Exploring the effect of the semantical hierarchy is, however, a very interesting avenue. For example, one interesting aspect one could explore in future work is whether one can reproduce our "pretraining effect" even with a fixed number of classes per task, and only by modifying the variety of the data from a semantical perspective.
>
> ---
>
> ## Other Comments
> - **Tractability of forgetting expression at $t \to \infty$**:
>
>     The short answer is yes, albeit the formula is not present in the current version of the paper. Indeed, we were recently able to extend our perturbation theory approach for the infinite-time solution in the two-layer linear network setting, where we have a non-trivial relation between $\gamma_0$ and similarity $\rho$.
>
> - **Cubic complexity of infinite-width simulations**:
>
>     The fields h and z are composed of sums over P data points, and discretized integrals over T time steps, i.e., h and g are $\mathcal{O}(PT)$. The NTK is a sum of $\Phi$ and $G$, which are the inner product of respective fields and thus $\mathcal{O}(P^2T^2)$. Finally, the output is the matrix-vector product of the NTK and $\Delta$ of dimension P, and then integrated over T time steps following the PDE of Eq. 3. This leads to $\mathcal{O}(P^3T^3)$.
>
> - **Noise of low-width networks**:
>
>    The training process is stochastic (over initialisation), and quantities like the NTK rely on the self-averaging properties characteristic of the $N\to\infty$ limit. Concretely, this means that in the limit, the network's dynamics become deterministic and instead, at finite widths, these observables are a partial/incomplete snapshot of the underlying process, introducing more stochasticity.
>
> - **Typos and notation inconsistencies**
>
>     Thanks for pointing these out. We will correct these in the final version of the paper.
>
> ---
>
>  *We hope that we have adequately addressed your questions. If you have further questions or comments, we remain at your disposal.*

---

### Decision · Program_Chairs · 2025-05-01

**Decision:**

Accept (poster)

**Comment:**

The submission presents a theoretical and experimental investigation into the effects of neural network parameterization on catastrophic forgetting, extending previous work that focused on the lazy regime (i.e. low plasticity). The authors identify a spectrum of training regimes characterized by a parameter, $\gamma_0$, revealing that variations in parameterization lead to different scaling laws related to forgetting. A notable finding is the *Edge of Laziness*, marking a transition influenced by task similarity in which forgetting shifts from low to high. The paper combines Dynamic Mean Field Theory (DMFT) with empirical validation on standard CL tasks such as Permuted MNIST, CIFAR-10, and TinyImageNet.

Reviewers unanimously appreciate the clarity of writing and the potential significance of the contribution, noting that the analysis bridges prior findings and offers new insights into the influence of parameterization on catastrophic forgetting. Some reviewers comment that the terminology "edge of laziness" lacks clarity in terms of the real contribution, and the extent to which DMFT enhances the experimental claims. There were also several clarifications and corrections requested -- especially the toning-down of the claims made in the discussion and the adjustment of the title to better articulate the content and contributions of the work.

Overall, the reviewers found the work intriguing, well-supported by both theoretical and experimental data, and a valuable contribution to the understanding of the dynamics of CL. The authors are encouraged to incorporate the clarifications presented in rebuttal as there are several that help articulate the subtleties of the theoretical predictions and empirical verification.